# A proximal ADMM for multiblock problems with block anti-upper triangular constraints

Zhanwang Deng [1]    Yuqiu Su [2]    Wen Huang [2]

## Abstract

In this paper, we present the convergence analysis of the proximal Alternating Direction Method of Multipliers (ADMM) for problems with block anti-upper triangular constraints. While the linear constraints can be treated separately, most analyses of ADMM and its variants predominantly regard the linear constraints as one. Hence, they rely on assumptions related to the entire constraint matrix, such as the full column rank assumption. However, some problems with block anti-upper triangular constraints that can be solved by ADMM do not satisfy these assumptions. To fill this gap, a new assumption is proposed and used to guarantee the global convergence of the proximal ADMM for nonconvex problems. In the strongly convex setting, we also prove the global convergence of the proximal ADMM and establish the linear convergence under four different scenarios. This work extends the theoretical understanding of the multi-block ADMM to problems with block anti-upper triangular constraints.

This work was supported by the National Natural Science Foundation of China (No. 12371311), the Natural Science Foundation of Fujian Province (No. 2023J06004), the Fundamental Research Funds for the Central Universities (No. 20720240151), and Xiaomi Young Talents Program. [1]Academy for Advanced Interdisciplinary Studies, Peking University, Beijing, China [2]School of Mathematical Sciences, Xiamen University, Xiamen, China. Correspondence to: Wen Huang <wen.huang@xmu.edu.cn>.

*Proceedings of the $43^{rd}$ International Conference on Machine Learning*, Seoul, South Korea. PMLR 306, 2026. Copyright 2026 by the author(s).

## 1. Introduction

In this paper, we consider a proximal ADMM algorithm for solving the multi-block problems in the form of:

$$\min_{\mathbf{X}_i \in \mathbb{R}^{m_i \times p_i}} \quad f_1(\mathbf{X}_1) + \cdots f_n(\mathbf{X}_n), \tag{1}$$

$$\text{s.t.} \begin{bmatrix} \mathcal{A}_{1,1} \cdots & & \cdots & \mathcal{A}_{1,n-1} & \mathcal{A}_{1,n} \\ \mathcal{A}_{2,1} \cdots & & \cdots & \mathcal{A}_{2,n-1} & 0 \\ \vdots & \vdots & \ddots & & \vdots \\ \mathcal{A}_{m,1} \cdots & \mathcal{A}_{m,n-m+1} & \cdots & & 0 \end{bmatrix} \begin{bmatrix} \mathbf{X}_1 \\ \mathbf{X}_2 \\ \vdots \\ \mathbf{X}_n \end{bmatrix} = \begin{bmatrix} \mathbf{b}_1 \\ \mathbf{b}_2 \\ \vdots \\ \mathbf{b}_m \end{bmatrix},$$

where $f_i : \mathbb{R}^{m_i \times p_i} \to (-\infty, \infty]$ is a proper closed function for $i = 1, \cdots, n$. $\mathcal{A}_{i,j} : \mathbb{R}^{m_j \times p_j} \to \mathbb{R}^{q_i}, i = 1, \cdots m, j = 1, \cdots, n-i+1$ are linear operators, and $\mathbf{b}_i \in \mathbb{R}^{q_i}$, for $i = 1, \cdots, m$. It is assumed in this paper that $\text{range}(\mathcal{A}_{i,j}) \subseteq \text{range}(\mathcal{A}_{i,n-i+1})$ for $i = 1, \cdots, m$ and $j = 1, \cdots, n-i$[1]. Problem (1) finds widespread applications across various domains. In what follows, we provide several instances where it is applicable.

**Example 1: Consensus problem.** The consensus problem is a classical problem in distributed optimization, where the objective is to coordinate multiple decision variables such that they eventually reach consensus while minimizing a certain objective function. The problem can be formulated as

$$\min_{\mathbf{X}_0, \mathbf{X}_1, \cdots, \mathbf{X}_n} \quad \sum_{i=1}^{n} f(\mathbf{X}_i) + h(\mathbf{X}_0), \tag{2}$$
$$\text{s.t.} \quad \mathbf{X}_0 = \mathbf{X}_i, \quad i = 1, \cdots, n.$$

One can easily verify that the linear constraint in (2) satisfies the block anti-upper triangular setting and hence is a special case of (1). In practical applications, this problem arises in various domains, including multi-agent cooperative control (Olfati-Saber et al., 2007; Ren & Beard, 2008), distributed machine learning (Yang et al., 2019; Boyd et al., 2011), and sensor network data fusion (Kar & Moura, 2008; Olfati-Saber, 2007).

---

[1]In the strongly convex case, some scenarios do not need this assumption, see Table 1 for details.

**Example 2: Block-angular problem.** The block-angular problem (Castro, 2016) can be reformulated as

$$\min_{\mathbf{X}_i} \quad \sum_{i=1}^{n+1} \frac{1}{2} \langle \mathbf{X}_i, \mathcal{Q}_i(\mathbf{X}_i) \rangle_{\mathbb{R}^{m_i \times p_i}} + \langle \mathbf{C}_i, \mathbf{X}_i \rangle_{\mathbb{R}^{m_i \times p_i}},$$

$$\text{s.t.} \begin{bmatrix} \mathcal{A}_{1,1} & \cdots & \mathcal{A}_{1,n} & \mathcal{A}_{1,n+1} \\ 0 & \cdots & 0 & \mathcal{A}_{2,n+1} \\ 0 & \ddots & 0 & \vdots \\ 0 & 0 & \mathcal{A}_{n,n} & \mathcal{A}_{n,n+1} \end{bmatrix} \begin{bmatrix} \mathbf{X}_1 \\ \mathbf{X}_2 \\ \vdots \\ \mathbf{X}_{n+1} \end{bmatrix} = \begin{bmatrix} \mathbf{b}_1 \\ \mathbf{b}_2 \\ \vdots \\ \mathbf{b}_n \end{bmatrix},$$

where $\mathcal{A}_{i,i}$ are of full rank for $i = 1, \cdots, n$, $\mathcal{Q}_i \succeq 0, i = 1, \cdots, n$ are element-wise operators, $\mathbf{C}_i \in \mathbb{R}^{m_i \times p_i}$, and $\langle \cdot, \cdot \rangle_{\mathbb{R}^{m_i \times p_i}}$ denotes the inner product of the Euclidean space $\mathbb{R}^{m_i \times p_i}$.[2] By swapping the order of the variables, it can be verified that the linear constraint in block-angular problem aligns with the block anti-upper triangular setting as that in (1). In practical applications, quadratic and linear objective functions with a primal block-angular structure appear in contexts, such as multicommodity flow problems (Assad, 1978) and statistical disclosure control (Hundepool et al., 2012).

**Example 3: Low-rank and sparse image processing model.** A sparse and low-rank based model (Deng et al., 2025) for image processing is represented by:

$$\min_{\mathbf{X}, \mathbf{W}} \quad \sum_i \phi(\sigma_i(\sqrt{\mathbf{X}^{\mathrm{T}} \mathbf{X} + \varepsilon^2})) + \lambda p(\mathbf{W}),$$

$$\text{s.t.} \quad \mathbf{X} = \mathbf{W}, \quad \mathcal{P}_\Omega(\mathcal{A}(\mathbf{W})) = \mathcal{P}_\Omega(\mathbf{Y}), \tag{3}$$

where $\phi$ is a surrogate function that substitutes the rank function, $\varepsilon$ is a smoothing parameter, $\mathcal{A}$ is a linear operator, $\mathcal{P}_\Omega$ denotes the projection operator on a given region $\Omega$, $p(\mathbf{W})$ is a Lipschitz continuously differentiable function to promote sparsity such as the Huber function or MCP function (Lu et al., 2014). The linear constraint in (3) is block anti-upper triangular, coinciding with (1). Model (3) has demonstrated considerable performance on image denoising and image inpainting problems (Deng et al., 2025).

**Example 4: Robust regression problem.** In (Chen et al., 2025), the nonconvex generalized elastic net regression (Majumdar & Ward, 2014; Tran et al., 2025) is formulated as the robust regression problem

$$\min_{\mathbf{X}_1, \mathbf{X}_2, \mathbf{X}_3} g(\mathbf{X}_1) + \alpha_1 \left\{ \alpha_2 P(\mathbf{X}_2) + \frac{1}{2}(1 - \alpha_2)\|\mathbf{X}_3\|_2^2 \right\},$$

$$\text{s.t.} \quad \mathcal{H}(\mathbf{X}_1) = \mathbf{X}_2, \quad \mathcal{H}(\mathbf{X}_1) = \mathbf{X}_3, \tag{4}$$

where $\mathbf{X}_1$ denotes the regression coefficients, $\mathbf{X}_2$ and $\mathbf{X}_3$ are slack variables, $g(\mathbf{X}_1)$ denotes the regression term, $P(\cdot)$ denotes the regularizer term, and $\mathcal{H}$ is a transformation matrix. Model (4) can also be used for variable selection (Fan & Li, 2001).

Note that Problem (1) (as well as Problems (2) to (4)) can be formulated as:

$$\min_{\mathbf{X}_i} \quad f_1(\mathbf{X}_1) + \cdots + f_n(\mathbf{X}_n),$$

$$\text{s.t.} \quad \mathcal{A}_1(\mathbf{X}_1) + \cdots + \mathcal{A}_n(\mathbf{X}_n) = \mathbf{b}, \tag{5}$$

by rewriting $\mathcal{A}_i = \begin{bmatrix} \mathcal{A}_{1,i}^{\mathrm{T}} & \cdots & \mathcal{A}_{m,i}^{\mathrm{T}} \end{bmatrix}^{\mathrm{T}}$ and $\mathbf{b} = \begin{bmatrix} \mathbf{b}_1^{\mathrm{T}} & \cdots & \mathbf{b}_m^{\mathrm{T}} \end{bmatrix}^{\mathrm{T}}$. To the best of our knowledge, the commonly used methods, ADMM and its variants, require a "*range assumption*", i.e.,

$$\text{range}(\mathcal{A}_i) \subseteq \text{range}(\mathcal{A}_n)$$

for $i = 1, \ldots, n - 1$, for the global convergence analyses if $f_i, i = 1, \cdots, n$ is nonconvex, see Table 3 in appendix for details. However, the range assumption fails for Problems (2) to (4). To address this limitation, we propose a weaker assumption, i.e., $\text{range}(\mathcal{A}_{i,j}) \subseteq \text{range}(\mathcal{A}_{i,n-i+1})$ for $i = 1, \cdots, m$ and $j = 1, \cdots, n - i$ (see A(2) in Assumption 1 for details), which holds for Problem (1) (as well as Problems (2) to (4)). In the strongly convex setting, we discuss the necessity of the weaker assumption (A(2)) in the convergence analysis. We provide a non-exhaustive review of the convergence analysis for ADMM and its variants under nonconvex and strongly convex settings in the following.

### 1.1. Related works

The classical ADMM is first introduced by Gabay, Mercier (Gabay & Mercier, 1976) and Glowinski, Fortin (Glowinski & Marroco, 1975)[3]. Due to its ease of implementation and effectiveness, it has been widely adopted in machine learning (Chang et al., 2014), distributed optimization (Boyd et al., 2011), and image processing (Liang et al., 2012; Chen et al., 2025). ADMM and its variants have proven to be effective for solving a wide range of problems (Wen et al., 2010; Sun et al., 2015; Makhdoumi & Ozdaglar, 2017). There have been several survey papers from different points of view (Han, 2022; Boyd et al., 2011) illustrating the theory and applications of ADMM. We briefly review the convergence analysis of ADMM for nonconvex and strongly convex problems with a focus on the range assumption.

---

[2] For notational convenience, we will often omit the subscript in the inner product hereafter, with the understanding that each $\langle \cdot, \cdot \rangle$ corresponds to the inner product in its respective space.

[3] In this paper, the classical ADMM refers to the ADMM without proximal term.

**Convergence of ADMM for nonconvex problems.** The existing global convergence for nonconvex Problem (5) either requires the range assumption (Yashtini, 2022; Li & Pong, 2015; Boţ & Nguyen, 2020; Liu et al., 2019; Wang et al., 2019; Hong et al., 2016) or other assumptions that imply it, such as full row rank (Wang et al., 2018) and $\mathcal{A}_n = \mathcal{I}$ (Hong et al., 2016; Jiang et al., 2019; Yang et al., 2017; Li & Pong, 2015). Therefore, essentially, they also satisfy the range assumption. Meanwhile, these existing works cover various ADMM-type algorithms, including classical ADMM, proximal ADMM, linearized ADMM, Bregman ADMM, and proximal linearized ADMM. Therefore, the existing range assumption plays a key role in the convergence analysis of ADMM-class algorithms for nonconvex problems. We present a nonexhaustive summary of typical works on nonconvex ADMM in Table 3. Compared with these works, we only assume $\mathrm{range}(\mathcal{A}_{i,j}) \subseteq \mathrm{range}(\mathcal{A}_{i,n-i+1})$ for $i = 1, \cdots, m$ and $j = 1, \cdots, n-i$, which is a weaker assumption for Problem (1).

**Convergence of ADMM using strong convexity.** In (Lin et al., 2015a), a multi-block classic ADMM is presented, and the linear convergence rate is established. However, it does not consider the ADMM with proximal term, and the penalty parameter $\beta$ needs to lie in a specific interval. In (Lin et al., 2018), a three-block ADMM is proven to converge globally with any $\beta > 0$ if the condition number of the strongly convex function is in [1,1.0798). The algorithm is used to solve a class of regularized least squares decomposition problems. In (Deng & Yin, 2016), a two-block type of ADMM with and without the proximal term is considered. The global and linear convergence results are given. These works (Deng & Yin, 2016; Lin et al., 2018; 2015a) all focus on strongly convex problems. A recent work (Chen et al., 2025) does not consider a strongly convex problem but uses essentially the strongly convex property in the convergence analysis. Specifically, the work (Chen et al., 2025) considers Problem (5) with $n = 3$ under the assumption that $f_1$ is weakly convex and $f_2$ and $f_3$ are strongly convex. The optimization problem can be transformed into an unconstrained strongly convex optimization problem if the weakly convex modulus $\mu_1 > 0$ (Chen et al., 2025, Lemma 3.4). Note that in the strongly convex setting, the range assumption in the existing works is not always assumed. In fact, it is also not necessary in the convex setting (Chen et al., 2013; Lin et al., 2015b; 2016). A summary of the theoretical results in existing ADMM for strongly convex problems is given in Table 4.

## 1.2. Contributions

- To the best of our knowledge, we are the first to consider a proximal ADMM for multiblock problems with block anti-upper triangular constraints. A novel assumption for problems with the block anti-upper triangular constraints is proposed. The assumption (i.e., A(2) in Assumption 1) holds for many applications in which the classical "range assumption" does not hold.
- For nonconvex problems, the global convergence analysis is given under the newly proposed assumption. To the best of our knowledge, it is the first global convergence analysis of proximal ADMM for multi-block nonconvex nonsmooth problems without the classic "range assumption".
- For strongly convex problems, the global and linear convergence under different scenarios and cases are established under reasonable assumptions.
- In the numerical experiments, we have demonstrated that Algorithm 1 is efficient for solving the generalized elastic net regression problem and the block angular problem in the nonconvex and strongly convex setting, respectively, which aligns with our theoretical findings. Furthermore, it also demonstrates robustness and efficiency compared with the classical ADMM.

## 1.3. Notation

Throughout this paper, scalars, vectors, and matrices are respectively denoted by lowercase letters $x$, boldface lowercase letters $\mathbf{x}$, and boldface capital letters $\mathbf{X}$. For a matrix $\mathbf{X} \in \mathbb{R}^{m \times n}$, $\mathbf{X}_{ij}$ denotes the $i$-th row, $j$-th column element of $\mathbf{X}$. The Frobenius norm of $\mathbf{X}$ is defined as $\|\mathbf{X}\|_{\mathrm{F}} := \sqrt{\sum_{i=1}^{n} \sum_{j=1}^{m} |\mathbf{X}_{ij}|^2}$. The inner product of two matrices $\mathbf{G}_1, \mathbf{G}_2 \in \mathbb{R}^{m \times n}$ is denoted as $\langle \mathbf{G}_1, \mathbf{G}_2 \rangle := \mathrm{tr}(\mathbf{G}_1^{\mathrm{T}} \mathbf{G}_2)$. The column space of a given matrix $\mathbf{X}$ is denoted by $\mathrm{range}(\mathbf{X})$. Given a symmetric positive semidefinite operator $\mathcal{W} : \mathbb{R}^{m \times n} \to \mathbb{R}^{m \times n}$, the weighted norm is defined as $\|\mathbf{X}\|_{\mathcal{W}} := \sqrt{\langle \mathcal{W}(\mathbf{X}), \mathbf{X} \rangle}$. The operator norm is defined as $\|\mathcal{W}\| := \sup_{\mathbf{X}} \frac{\|\mathcal{W}(\mathbf{X})\|_{\mathrm{F}}}{\|\mathbf{X}\|_{\mathrm{F}}}$. For an operator $\mathcal{W}$, we use $\mathcal{W}^{\mathrm{T}}$ to denote the its adjoint. When $\mathcal{W}$ is a matrix, $\mathcal{W}^{\mathrm{T}}$ is its transpose. Let $\mathcal{I} : \mathbb{R}^{m \times n} \to \mathbb{R}^{m \times n}$ denote the identity operator. If $n = 1$, then $\mathcal{I}$ is the $m$ by $m$ identity matrix. We use $\mathrm{dom}(f)$ to denote the domain of the function $f$ and $\frac{0}{0}$ is defined as $\infty$ for the simplicity of notation. We also use $[n]$ to denote the set $\{1, \cdots, n\}$.

### 1.4. Organization

The rest of the paper is organized as follows. The proximal ADMM algorithm and the convergence analysis under nonconvex and strongly convex settings are presented in Section 2. Numerical experiments are conducted in Section 3 to verify our theoretical results. Finally, we conclude this paper in Section 4.

## 2. A proximal ADMM for problems with block anti-upper triangular constraints

The derivation of the ADMM begins with the augmented Lagrangian function. In particular, the augmented Lagrangian function of Problem (1) is given by $\mathcal{L}(\overline{\mathbf{X}}, \overline{\mathbf{\Lambda}}) = \sum_{i=1}^{n} f_i(\mathbf{X}_i) + \sum_{j=1}^{m} \frac{\beta_j}{2} \|\mathcal{A}_{j,1}\mathbf{X}_j + \cdots + \mathcal{A}_{j,n-j+1}\mathbf{X}_{n-j+1} - \mathbf{b}_j + \frac{\mathbf{\Lambda}_j}{\beta_j}\|_{\mathrm{F}}^2 - \sum_{i=1}^{m} \frac{1}{2\beta_j}\|\mathbf{\Lambda}_i\|_{\mathrm{F}}^2$, where $\overline{\mathbf{X}} = [\mathbf{X}_1, \cdots, \mathbf{X}_n] \in \mathbb{R}^{m_1 \times p_1} \times \cdots \times \mathbb{R}^{m_n \times p_n}$ denotes the variable, $\overline{\mathbf{\Lambda}} = [\mathbf{\Lambda}_1, \cdots, \mathbf{\Lambda}_m] \in \mathbb{R}^{q_1} \times \cdots \mathbb{R}^{q_n}$ denotes the Lagrangian multipliers, and $\beta_j > 0, \forall j$ denote the penalty parameters. Let $\overline{\mathbf{X}}^{(k)} = [\mathbf{X}_1^{(k)}, \cdots, \mathbf{X}_n^{(k)}]$ and $\overline{\mathbf{\Lambda}}^{(k)} = [\mathbf{\Lambda}_1^{(k)}, \cdots, \mathbf{\Lambda}_m^{(k)}]$ denote the $k$-th iteration point for $\overline{\mathbf{X}}$ and $\overline{\mathbf{\Lambda}}$ respectively. The proposed proximal ADMM for solving (1) is summarized in Algorithm 1. In Step 4 of Algorithm 1, $\mathbf{X}_i^{(k+1)}$ is a minimizer of the objective $\mathcal{L}(\overline{\mathbf{X}}, \overline{\mathbf{\Lambda}}) + \frac{1}{2} \left\| \mathbf{X}_i - \mathbf{X}_i^{(k)} \right\|_{\mathcal{Q}_i^{(k)}}^2$, where $\mathbf{X}_j = \mathbf{X}_j^{(k+1)}$ if $1 \leq j \leq i-1$ and $\mathbf{X}_j = \mathbf{X}_j^{(k)}$ if $i+1 \leq j \leq n$. Step 7 of Algorithm 1 is used to update the Lagrange multipliers. All the proofs of the theorems and Lemmas are presented in Appendix A.

### 2.1. Convergence analysis under nonconvex setting

We first introduce some notations for the subsequent analysis. The algorithm-dependent notations $q_i = \sup_{k \geq 0} \|\mathcal{Q}_i^{(k)}\|$ for $i = 1, \cdots, n$; $q_i^- = \inf_{k \geq 0} \|\mathcal{Q}_i^{(k)}\|$ for $i = 1, \cdots, n$; $\Delta \mathbf{U}^{(k)} = \mathbf{U}^{(k)} - \mathbf{U}^{(k-1)}$ for any sequence $\{\mathbf{U}^{(k)}\}$ generated by Algorithm 1. The problem-dependent notations $c_A$ denotes the inverse of the smallest positive eigenvalue of $\mathcal{A}_{i,j}^{\mathrm{T}}\mathcal{A}_{i,n-i+1}, i = 1, \cdots, m$; $d_A = \max_{j=1,\cdots,m} \sqrt{\sum_{i=1}^{n-j+1} \|\mathcal{A}_{j,i}^{\mathrm{T}}\mathcal{A}_{j,n-j+1}\|}$. The global convergence for nonconvex problems relies on Assumption 1.

**Assumption 1.**

*A(1) The lower semi-continuous functions $f_i, i = 1, \cdots n - m$ are coercive and the functions $f_i, i = n - m + 1, \cdots, n$ are $L_{f_i}$ Lipschitz differentiable respectively and bounded from below. A function $h$ is called $L_h$*

*Lipschitz differentiable if it holds that $\|\nabla h(\mathbf{W}_1) - \nabla h(\mathbf{W}_2)\|_{\mathrm{F}} \leq L_h \|\mathbf{W}_1 - \mathbf{W}_2\|_{\mathrm{F}}, \quad \forall \mathbf{W}_1, \mathbf{W}_2$. It has been shown in (Beck, 2017)[Lemma 5.7] that it implies $h(\mathbf{W}_1) \leq h(\mathbf{W}_2) + \langle \nabla h(\mathbf{W}_2), \mathbf{W}_1 - \mathbf{W}_2 \rangle + \frac{L_h}{2}\|\mathbf{W}_1 - \mathbf{W}_2\|_{\mathrm{F}}^2, \quad \forall \mathbf{W}_1, \mathbf{W}_2$.*

*A(2) $\mathrm{range}(\mathcal{A}_{i,j}) \subseteq \mathrm{range}(\mathcal{A}_{i,n-i+1}), i = [m]; j = [n-i]$.*

*A(3) $\mathcal{A}_{i,n-i+1}$ is injective, $\quad i = 1, \cdots, m$.*

*A(4) The sequences $\{\mathcal{Q}_i^{(k)}\}$ and the penalty parameters $\{\beta_i\}$ satisfy that (i) $q_i < \infty, i = 1, \cdots, n$; (ii) there exist constants $r < 1$ and $a_i > 0, i = 1, \cdots, n$ such that*

$$\frac{\beta_{n-i+1}}{2}\mathcal{A}_{n-i+1,i}^{\mathrm{T}}\mathcal{A}_{n-i+1,i} - r(\theta_{0,i} + \theta_{1,i})\mathcal{I}$$
$$+ (q_i^- - L_{f_i})\mathcal{I} \succeq a_i\mathcal{I}, \quad i = n - m + 1, \cdots, n, \tag{6}$$

$$q_i^-\mathcal{I} \succeq a_i\mathcal{I}, \quad i = 1, \cdots, n - m,$$

*where*

$$\theta_{0,l} = \begin{cases} \frac{2(n+1-l)}{\beta_{n+1-l}}c_{n-l+1,n-l+1}\eta_{0,l}^2 \\ \quad + \sum_{i=n-l+2}^{m} \frac{2i}{\beta_i}c_A^2 \left( \sum_{p=n-l+2}^{i} \bar{\beta}_{n-l+1}c_{p,i}d_A^2 \right. \\ \quad \left. + c_{n-l+1,i}\eta_{0,l} \right)^2, \text{ for } l = n - m + 2, \cdots, n; \\ \frac{m}{\beta_m}\eta_{0,l}^2, \quad \text{ for } l = n - m + 1, \end{cases}$$

$$\theta_{1,l} = \begin{cases} \frac{2(n+1-l)}{\beta_{n+1-l}}c_{n-l+1,n-l+1}\eta_{1,l}^2 \\ \quad + \sum_{i=n-l+2}^{m} \frac{2i}{\beta_i}c_A^2 \left( \sum_{p=n-l+2}^{i} \bar{\beta}_{n-l+1}c_{p,i}d_A^2 \right. \\ \quad \left. + c_{n-l+1,i}\eta_{1,l} \right)^2, \text{ for } l = n - m + 2, \cdots, n; \\ \frac{m}{\beta_m}\eta_{1,l}^2, \quad \text{ for } l = n - m + 1, \end{cases}$$

$$\tag{7}$$

*$\eta_{1,l} = L_{f_l} + q_l$, $L_{f_l}$ is the Lipschitz constant of $\nabla f_l$, $\eta_{0,l} = q_l$, $\bar{\beta}_i = \max_{j<i} \beta_j$, and*

$$c_{k,i} = \begin{cases} 1, & \text{if } k = i, \\ \sum_{j=k}^{i-1} b_{i,j}\, c_{k,j}, & \text{if } 1 \leq k < i. \end{cases}$$

*$b_{i,j} = c_A d_A$ which depends only on $c_A$ and $d_A$ (see Lemmas 5 and 6 for details).*
*(iii) $\theta_{3,i} < \frac{1}{2L_{f_i}}$, for $i = n - m + 1, \cdots, n$, where $\theta_{3,i} = \sum_{i=n-l+1}^{m} \frac{2(n+1-i)}{\beta_{n+1-i}}c_{n-l+1,i}^2 c_A^2 \eta_{0,l}^2$.*

A(1) of Assumption 1 has been widely-used for the convergence analysis of nonconvex ADMM (Yashtini, 2022; Li & Pong, 2015) for the case $m = 1$. A(2) is the proposed assumption that is weaker than the range assumption and holds for the examples in Section 1. Note that if $\mathcal{A}_{i,n-i+1}$ is a matrix, then A(3) is equivalent to that $\mathcal{A}_{i,n-i+1}$ has full

---

**Algorithm 1** Proximal ADMM algorithm (pADMM)

---

1: **Input:** Initial iterate $\overline{\mathbf{X}}^{(0)} \in \mathbb{R}^{m_1 \times p_1} \times \cdots \times \mathbb{R}^{m_n \times p_n}$; initial multipliers $\overline{\boldsymbol{\Lambda}}^{(0)} \in \mathbb{R}^{q_1 \times \cdots \times q_m}$; and a sequence of positive semidefinite linear operators $\mathcal{Q}_i^{(k)} : \mathbb{R}^{m_i \times p_i} \to \mathbb{R}^{m_i \times p_i}$, $i = 1, \cdots, n$.

2: **for** $k = 1, 2, \ldots$ **do**

3:     **for** $i = 1, \cdots n$ **do**

4:         $\mathbf{X}_i^{(k+1)} \in \arg\min_{\mathbf{X}_i} \mathcal{L}\left(\left[\mathbf{X}_1^{(k+1)}, \ldots, \mathbf{X}_i, \mathbf{X}_{i+1}^{(k)}, \ldots, \mathbf{X}_n^{(k)}\right], \overline{\boldsymbol{\Lambda}}^{(k)}\right) + \frac{1}{2}\left\|\mathbf{X}_i - \mathbf{X}_i^{(k)}\right\|_{\mathcal{Q}_i^{(k)}}^2$;     **(Update $\overline{\mathbf{X}}^{(k+1)}$)**

5:     **end for**

6:     **for** $i = 1, \cdots m$ **do**

7:         $\boldsymbol{\Lambda}_i^{(k+1)} = \boldsymbol{\Lambda}_i^{(k)} + \beta_i \left(\sum_{j=1}^{n-i+1} \mathcal{A}_{i,j}(\mathbf{X}_j^{(k+1)}) - \mathbf{b}_i\right)$;             **(Update $\overline{\boldsymbol{\Lambda}}^{(k+1)}$)**

8:     **end for**

9: **end for**

---

column rank. The tightness of A(1) and A(3) in Assumption 1 for problems with $m = 1$ is presented in (Wang et al., 2019). For A(4), we have the following lemma.

**Lemma 1.** *If A(3) in Assumption 1 holds, there exist $\beta_i, i \in [m]$ such that (ii) and (iii) of A(4) in Assumption 1 hold.*

### 2.1.1. GLOBAL CONVERGENCE

To guarantee the global convergence under the nonconvex setting, it is shown in Lemma 4 in Appendix A that $\sum_{i=1}^{m} \frac{1}{\beta_i} \|\Delta \boldsymbol{\Lambda}_i^{(k+1)}\|_{\mathrm{F}}^2$ in can be bounded by $\|\Delta \mathbf{X}_i^{(k)}\|$ and $\|\Delta \mathbf{X}_i^{(k+1)}\|, i = n - m + 1, \cdots, n$. It is the main difficulty in the construction of the merit function. Due to the block anti-upper triangular constraints, our proof technique is of great difference compared with the works that consider Problem (5), see Lemma 6 in Appendix for details.

Let $\mathcal{R}(\overline{\mathbf{X}}, \overline{\boldsymbol{\Lambda}}, \overline{\mathbf{X}}') := \mathcal{L}(\overline{\mathbf{X}}, \overline{\boldsymbol{\Lambda}}) + \sum_{i=n-m+1}^{n} r\theta_{0,i} \|\mathbf{X}_i - \mathbf{X}_i'\|_{\mathrm{F}}^2$. The merit function $\mathcal{R}^{(k)}$ is defined by $\mathcal{R}^{(k)} := \mathcal{R}(\overline{\mathbf{X}}^{(k)}, \overline{\boldsymbol{\Lambda}}^{(k)}, \overline{\mathbf{X}}^{(k-1)}) = \mathcal{L}(\overline{\mathbf{X}}^{(k)}, \overline{\boldsymbol{\Lambda}}^{(k)}) + \sum_{i=n-m+1}^{n} r\theta_{0,i} \|\Delta \mathbf{X}_i^{(k)}\|_{\mathrm{F}}^2$. If Assumption 1 holds, it can be shown that the merit function is decreasing. Specifically, inequality (8) holds:

$$\mathcal{R}^{(k+1)} + a\left(\|\Delta \overline{\mathbf{X}}^{(k+1)}\|_{\mathrm{F}}^2 + \|\Delta \overline{\boldsymbol{\Lambda}}^{(k+1)}\|_{\mathrm{F}}^2\right) \leq \mathcal{R}^{(k+1)}$$
$$+ \sum_{i=1}^{n-m} \frac{q_i}{2} \|\Delta \mathbf{X}_i^{(k+1)}\|_{\mathrm{F}}^2 + \sum_{i=1}^{m} \frac{r-1}{\beta_i} \|\Delta \boldsymbol{\Lambda}_i^{(k+1)}\|_{\mathrm{F}}^2$$
$$+ \sum_{i=n-m+1}^{n} \|\Delta \mathbf{X}_i^{(k+1)}\|_{\mathcal{B}_i^{(k)} - r(\theta_{1,i} + \theta_{0,i})\mathcal{I}}^2 \leq \mathcal{R}^{(k)} \leq \mathcal{R}^{(1)},$$
$$(8)$$

where $a = \min_{i=1,\cdots,n,j=1,\cdots,m} \{a_i, \frac{r-1}{\beta_j}\}$, $a_i$ is defined in Assumption 1, the first inequality follows from A(4) in Assumption 1, the second inequality follows from Lemma 6 in appendix, and the third inequality is due to induction of $\mathcal{R}^{(k)} \leq \mathcal{R}^{(k+1)}$ for any $k \geq 0$. After obtaining the

sufficient decrease property of $\mathcal{R}^{(k)}$, it follows that $\mathcal{R}^{(k)}$ is monotonically decreasing and has a lower bound. We aim to bound the iteration sequence $\{(\overline{\mathbf{X}}^{(k)}, \overline{\boldsymbol{\Lambda}}^{(k)})\}_{k \geq 0}$, where the main difference compared with the existing works is to bound $\{\overline{\boldsymbol{\Lambda}}^{(k)}\}$ using A(3) in Assumption 1.

**Theorem 1** (Bounded sequence of $\{(\overline{\mathbf{X}}^{(k)}, \overline{\boldsymbol{\Lambda}}^{(k)})\}_{k \geq 0}$)**.** *If Assumption 1 holds, the sequence $\{(\overline{\mathbf{X}}^{(k)}, \overline{\boldsymbol{\Lambda}}^{(k)})\}_{k \geq 0}$ generated by Algorithm 1 is bounded.*

According to Theorem 1, $(\overline{\mathbf{X}}^{(k)}, \overline{\boldsymbol{\Lambda}}^{(k)})$ has limit point due to the Weierstrass theorem. Theorem 2 presents the property of the limit point set.

**Theorem 2.** *Let $\Pi$ denote the limit point set of the sequence $\{(\overline{\mathbf{X}}^{(k)}, \overline{\boldsymbol{\Lambda}}^{(k)})\}$. It follows that $\Pi$ is nonempty, $\mathrm{dist}((\overline{\mathbf{X}}^{(k)}, \overline{\boldsymbol{\Lambda}}^{(k)}), \Pi) = 0$, and $\Pi \subseteq \mathrm{crit}\,\mathcal{L}$ where $\mathrm{crit}\,\mathcal{L}$ denotes the critical point of $\mathcal{L}$, i.e. $0 \in \partial\mathcal{L}(\overline{\mathbf{X}}, \overline{\boldsymbol{\Lambda}})$ for $(\overline{\mathbf{X}}, \overline{\boldsymbol{\Lambda}}) \in \Pi$.*

According to Theorem 1, Lemmas 9 and 8 in the appendix, the following convergence result of the whole sequence, established under the Kurdyka-Łojasiewicz property, follows from (Yashtini, 2022, Theorem 2).

**Theorem 3** (Global Convergence)**.** *Suppose that Assumption 1 holds and $\mathcal{R}$ satisfies the KL property on the limit point set $\Gamma$, i.e., for every $\mathbf{V}_* = (\overline{\mathbf{X}}_*, \overline{\boldsymbol{\Lambda}}_*, \overline{\mathbf{X}}_*) \in \Gamma$, there exists $\varepsilon > 0$ and desingularizing function $\psi : [0, \eta] \to [0, \infty)$, for some $\eta \in [0, \infty)$ such that for all $\mathbf{V} = (\overline{\mathbf{X}}, \overline{\boldsymbol{\Lambda}}, \overline{\mathbf{X}}')$ in the set: $\mathcal{S} := \{\mathbf{V} : \mathrm{dist}(\mathbf{V}, \Gamma) < \varepsilon \text{ and } \mathcal{R}(\mathbf{V}^*) < \mathcal{R}(\mathbf{V}) < \mathcal{R}(\mathbf{V}^*) + \eta\}$, the inequality*

$$\psi'(\mathcal{R}(\mathbf{V}) - \mathcal{R}(\mathbf{V}^*))\mathrm{dist}(0, \partial\mathcal{R}(\mathbf{V})) \geq 1$$

*holds. Then $\{(\overline{\mathbf{X}}^{(k)}, \overline{\boldsymbol{\Lambda}}^{(k)})\}_{k \geq 0}$ satisfies the finite length property: $\sum_{k=0}^{\infty} \|\Delta \overline{\mathbf{X}}^{(k)}\|_{\mathrm{F}} + \|\Delta \overline{\boldsymbol{\Lambda}}^{(k)}\|_{\mathrm{F}} < \infty$, and conse-*

*quently converges to a stationary point of* (1)*, i.e.*

$$\sum_{l=1}^{\min\{n-i+1,m\}} \mathcal{A}_{l,i}^{\mathrm{T}} \overline{\mathbf{\Lambda}}_{*,i} \in \partial f_i(\mathbf{X}_{*,i}), \quad i = 1, \cdots, n,$$

$$\mathcal{A}_{j,1}\mathbf{X}_{*,1} + \cdots + \mathcal{A}_{j,n-j+1}\mathbf{X}_{*,n} = \mathbf{b}_j, \quad j = 1, \cdots, m. \tag{9}$$

## 2.2. Convergence analysis under strongly convex setting

In this subsection, we present the linear convergence of Algorithm 1 under the strongly convex setting. The assumptions for four scenarios and the convergence results for four cases are summarized in Tables 1 and 2, respectively. It is noted that $\mathcal{Q}_i^{(k)} = \mathcal{Q}_i$ for $i = 1, \cdots, n$ in the subsequent analysis. We make the following standard assumptions that have been used in (Deng & Yin, 2016; Lin et al., 2015a).

**Assumption 2.** *The saddle point set $\Omega$ for* (1) *is nonempty, i.e., there exists $(\overline{\mathbf{X}}_*, \overline{\mathbf{\Lambda}}_*)$ such that* (9) *holds.*

**Assumption 3.** $f_i, i = 1, \cdots, n$ *are all convex. For all* $\mathbf{X}_1, \mathbf{X}_2 \in \mathrm{dom}(f_i), \mathbf{Y}_1 \in \partial f_i(\mathbf{X}_1), \mathbf{Y}_2 \in \partial f_i(\mathbf{X}_2)$, *we define the following modules $\mu_i$ for the subsequent analysis:*

$$\langle \mathbf{Y}_1 - \mathbf{Y}_2, \mathbf{X}_1 - \mathbf{X}_2 \rangle \geq \mu_{f_i} \|\mathbf{X}_1 - \mathbf{X}_2\|_{\mathrm{F}}^2.$$

### 2.2.1. GLOBAL CONVERGENCE

We first define a merit function $r^{(k)}$ and establish its sufficient decrease property.

**Lemma 2** (Sufficient decrease). *Let $(\overline{\mathbf{X}}_{\mathrm{op}}, \overline{\mathbf{\Lambda}}_{\mathrm{op}}) = (\mathbf{X}_{\mathrm{op},1}, \cdots, \mathbf{X}_{\mathrm{op},n}, \mathbf{\Lambda}_{\mathrm{op},1}, \cdots, \mathbf{\Lambda}_{\mathrm{op},m}) \in \Omega$, the following inequality holds:*

$$r^{(k)} - r^{(k+1)} \geq \sum_{i=1}^{n} \left( \kappa_{f_i} - \frac{\beta_{\max} a_i}{2} e_A^2 \right) \|\mathbf{X}_i^{(k+1)} - \mathbf{X}_{\mathrm{op},i}\|_{\mathrm{F}}^2$$

$$+ \frac{\beta_i}{2} \sum_{j=1}^{m} u_{1,j,k} + \frac{1}{2} \sum_{i=1}^{n} \|\mathbf{X}_i^{(k)} - \mathbf{X}_i^{(k+1)}\|_{\mathcal{Q}_i}^2, \tag{10}$$

*where*

$$a_i = \begin{cases} m(i-1)(\frac{(2n-i)-(m-1)}{2})e_A^2, & \text{if } m \leq n-i+1, \\ \frac{n}{2}(n-i+1)(i-1)e_A^2, & \text{if } m > n-i+1, \end{cases}$$

$e_A = \max_{i,j} \|\mathcal{A}_{i,j}\|, i = [m], j = [n-i+1], u_{i,j,k} = \left\| \sum_{l=1}^{i-1} \mathcal{A}_{j,l}\mathbf{X}_{\mathrm{op},l} + \mathcal{A}_{j,i}\mathbf{X}_i^{(k+1)} + \sum_{l=i+1}^{n-j+1} \mathcal{A}_{j,l}\mathbf{X}_l^{(k)} - \mathbf{b}_j \right\|_{\mathrm{F}}^2$ *and*

$$r^{(k)} = \sum_{j=1}^{m} \frac{\beta_i}{2} \sum_{i=1}^{n-j+1} \left\| \sum_{l=i+1}^{n-j+1} \mathcal{A}_{j,l}\mathbf{X}_l^{(k)} - \mathcal{A}_{j,l}\mathbf{X}_{\mathrm{op},l} \right\|_{\mathrm{F}}^2$$

$$+ \sum_{i=1}^{m} \frac{1}{\beta_i} \|\mathbf{\Lambda}_i^{(k)} - \mathbf{\Lambda}_{\mathrm{op},i}\|_{\mathrm{F}}^2 + \frac{1}{2} \sum_{i=1}^{n} \|\mathbf{X}_i^{(k)} - \mathbf{X}_{\mathrm{op},i}\|_{\mathcal{Q}_i}^2. \tag{11}$$

The proof of Lemma 2 follows a line similar to that in (Lin et al., 2015a), but the details are substantially different because we consider the more general case $m > 1$, while (Lin et al., 2015a) only studies $m = 1$. In the special case $m = 1$, our analysis can be viewed as an extension of (Lin et al., 2015a) to both the proximal and non-proximal settings. Moreover, when $m = 1$, we have $a_i = \frac{(2n-i)(i-1)}{2}$, which agrees with Equation (2.17) of (Lin et al., 2015a). The coefficient of $a_n$ in our derivation, however, is different, because the summation in (41) runs from $i = 1$ to $n - j + 1$ rather than from $i = 1$ to $n - j$. Theorem 4 presents the global convergence of Algorithm 1 under different scenarios.

**Theorem 4** (Global convergence). *Suppose $\beta_{\max}$ satisfies*

$$\beta_{\max} < \min_{i=2,\cdots,n-1} \left\{ \frac{2\kappa_{f_i}}{a_i} \right\}. \tag{12}$$

*In Scenario 1, cases 1 and 3, there exists $(\overline{\mathbf{X}}_*, \overline{\mathbf{\Lambda}}_*) \in \Omega$ such that $(\mathcal{A}_1\mathbf{X}_1^{(k)}, \mathbf{X}_2^{(k)}, \cdots, \mathbf{X}_n^{(k)}, \overline{\mathbf{\Lambda}}^{(k)})$ converges to $(\mathcal{A}_1\mathbf{X}_{*,1}, \mathbf{X}_{*,2}, \cdots, \mathbf{X}_{*,n}, \overline{\mathbf{\Lambda}}_*)$, regardless of the choice of $\mathcal{Q}_i, i = 1, \cdots, n$. In Scenarios 1,2, 3, and 4, it holds that $(\overline{\mathbf{X}}^{(k)}, \overline{\mathbf{\Lambda}}^{(k)})$ generated by Algorithm 1 converges to some $(\overline{\mathbf{X}}_*, \overline{\mathbf{\Lambda}}_*) \in \Omega$, regardless of the choice of $\mathcal{Q}_i, i = 1, \cdots, n$.*

### 2.2.2. Q-LINEAR CONVERGENCE

Define $\tilde{r}^{(k)} = \frac{1}{2} \sum_{j=1}^{m} \beta_j \sum_{i=1}^{n-j+1} \| \sum_{l=i+1}^{n-j+1} \mathcal{A}_{j,l}(\mathbf{X}_l^{(k)} - \mathbf{X}_{*,l})\|_{\mathrm{F}}^2 + r^{(k)}$ and it follows form Lemma 2 that

$$\tilde{r}^{(k)} - \tilde{r}^{(k+1)} \geq \sum_{i=2}^{n} \left[ (\kappa_{f_i} - \beta_{\max}a_i) \|\mathbf{X}_i^{(k+1)} - \mathbf{X}_{*,i}\|_{\mathrm{F}}^2 \right]$$

$$+ \sum_{j=1}^{m} \frac{\beta_j}{2} \sum_{i=1}^{n-j+1} \| \sum_{l=i+1}^{n-j+1} \mathcal{A}_{j,l}\mathbf{X}_l^{(k)} - \mathcal{A}_{j,l}\mathbf{X}_{*,l}\|_{\mathrm{F}}^2 \tag{13}$$

$$+ \sum_{i=1}^{m} \frac{\beta_i}{2} u_{1,i,k} + \frac{1}{2} \sum_{i=1}^{n} \|\mathbf{X}_i^{(k+1)} - \mathbf{X}_i^{(k)}\|_{\mathcal{Q}_i}^2.$$

Based on this merit function, the next theorem establishes the Q-linear convergence result summarized in Table 2.

**Theorem 5** (Linear convergence). *Suppose the conditions in Table 1 hold. For four cases, if $\beta_{\max} < \min_{i=2,\cdots,n-1} \left\{ \frac{\kappa_{f_i}}{a_i} \right\}$, then for $\tilde{r}^{(k)} = \sum_{j=1}^{m} \beta_j \sum_{i=1}^{n-j+1} \| \sum_{l=i+1}^{n-j+1} \mathcal{A}_{j,l}(\mathbf{X}_l^{(k)} - \mathbf{X}_{*,l})\|_{\mathrm{F}}^2 + \sum_{i=1}^{m} \frac{1}{\beta_i} \|\mathbf{\Lambda}_i^{(k)} - \mathbf{\Lambda}_{*,i}\|_{\mathrm{F}}^2 + \frac{1}{2} \sum_{i=1}^{n} \|\mathbf{X}_i^{(k)} - \mathbf{X}_{*,i}\|_{\mathcal{Q}_i}^2$*

$$\tilde{r}^{(k)} \geq (1 + \delta)\tilde{r}^{(k+1)}. \tag{14}$$

Hence, $\overline{\mathbf{H}}^{(k)}$ defined in Table 2 converges Q-linearly to $(\sum_{j=2}^{n} \mathcal{A}_j\mathbf{X}_{j,*}, \ldots, \sum_{j=n-1}^{n} \mathcal{A}_j, \sum_{j=n}^{n} \mathcal{A}_j\mathbf{X}_{j,*}, \overline{\mathbf{\Lambda}}_*)$. In

*Table 1.* Assumptions for four scenarios under strongly convex setting.

| Scenario | Strongly convex | Lipschitz continuous | Operator assumption | Additional assumptions |
|---|---|---|---|---|
| 1 | $f_{\min\{2,n-m+1\}},\cdots,f_n$ | $\nabla f_i\ (i=n-m+1,\cdots,n)$ | Assumption A(2) | $\mathcal{A}_1$ is injective. |
| 2 | $f_1,f_2,\cdots,f_n$ | $\nabla f_i\ (i=n-m+1,\cdots,n)$ | Assumption A(2) | |
| 3 | $f_2,\cdots,f_n$ | $\nabla f_i\ (i=1,\cdots,n)$ | - | $\mathcal{A}_1$ is injective. |
| 4 | $f_1,f_2,\cdots,f_n$ | $\nabla f_i\ (i=1,\cdots,n)$ | - | - |

*Table 2.* Convergence results for four cases of the proximal term (see Theorems 5 and 6).

| Case | $\mathcal{Q}_i\ (i=1,\cdots,n-m)$ | $\mathcal{Q}_i\ (i=n-m+1,\cdots,n)$ | Scenario 1-4 ▷ | |
|---|---|---|---|---|
| | | | Q-linear convergence | R-linear convergence◇ |
| 1 | $\mathcal{Q}_i=0$ | $\mathcal{Q}_i=0$ | $(\overline{\mathbf{H}}^{(k)}\,\clubsuit,\overline{\mathbf{\Lambda}}^{(k)})$ | $(\mathbf{X}_n^{(k)},\overline{\mathbf{T}}^{(k)},\overline{\mathbf{\Lambda}}^{(k)})$ |
| 2 | $\mathcal{Q}_i=0$ | $\mathcal{Q}_i\succ0$ | $(\mathbf{X}_i^{(k)}(i=n-m+1,\cdots,n),\overline{\mathbf{H}}^{(k)},\overline{\mathbf{\Lambda}}^{(k)})$ | $(\mathbf{X}_i^{(k)}(i=n-m+1,\cdots,n),\overline{\mathbf{T}}^{(k)},\overline{\mathbf{\Lambda}}^{(k)})$ |
| 3 | $\mathcal{Q}_i\succ0$ | $\mathcal{Q}_i=0$ | $(\mathbf{X}_i^{(k)}\ (i=1,\cdots,n-m),\overline{\mathbf{H}}^{(k)},\overline{\mathbf{T}}^{(k)},\overline{\mathbf{\Lambda}}^{(k)})$ | $(\mathbf{X}_i^{(k)}\ (i=1,\cdots,n-m),\overline{\mathbf{T}}^{(k)},\overline{\mathbf{\Lambda}}^{(k)})$ |
| 4 | $\mathcal{Q}_i\succ0$ | $\mathcal{Q}_i\succ0$ | $(\mathbf{X}_i^{(k)}(i=1,\cdots,n),\overline{\mathbf{H}}^{(k)},\overline{\mathbf{\Lambda}}^{(k)})$ | $(\mathbf{X}_i^{(k)}(i=1,\cdots,n),\overline{\mathbf{T}}^{(k)},\overline{\mathbf{\Lambda}}^{(k)})$ |

♣ $\overline{\mathbf{H}}^{(k)}$ and $\overline{\mathbf{T}}^{(k)}$ are auxiliary variables derived from $\overline{\mathbf{X}}^{(k)}$. $\overline{\mathbf{H}}^{(k)}=(\mathbf{H}_2^{(k)},\cdots,\mathbf{H}_n^{(k)})$, where $\mathbf{H}_i^{(k)}=\sum_{j=i}^n\mathcal{A}_j\mathbf{X}_j^{(k)}$.
$\overline{\mathbf{T}}^{(k)}=(\mathbf{T}_2^{(k)},\cdots,\mathbf{T}_n^{(k)})$, where $\mathbf{T}_i^{(k)}=\mathcal{A}_j\mathbf{X}_j^{(k)}$.
◇ Any part of a Q-linearly convergent sequence converges R-linearly.
▷ Our results are extensions of the results in (Deng & Yin, 2016, Table 2) which consider the case $m=1$ and $n=2$.

Theorem 5, the main difficulty and difference compared with other works is to bound $\sum_{i=1}^m\frac{1}{\beta_i}\|\mathbf{\Lambda}_i^{(k+1)}-\mathbf{\Lambda}_{*,i}\|_{\mathrm{F}}^2$ under different scenarios, see Lemma 11 in appendix for details.

### 2.2.3. R-LINEAR CONVERGENCE

By the definition of R-linear convergence, any part of a Q-linearly convergent sequence converges R-linearly. The following theorem presents the R-linear convergence of the iteration point sequences $\mathbf{X}_i^{(k)}, i=1,\cdots,n$ and $\mathbf{\Lambda}_j^{(k)}, j=1,\cdots,m$ of Algorithm 1.

**Theorem 6.** *Under the same conditions in Theorem 5, $\mathbf{X}_n^{(k)}$, $\overline{\mathbf{\Lambda}}^{(k)}$, and $\mathcal{A}_i\mathbf{X}_i^{(k)}, i=1,\ldots,n-1$, converge R-linearly. Moreover, if $\mathcal{A}_i, i=1,2,\ldots,n-1$ are further assumed to be injective, then $\mathbf{X}_i^{(k)}, i=1,2,\ldots,n-1$ converge R-linearly, respectively.*

## 3. Numerical experiments

In this section, we conduct numerical experiments to verify the convergence analysis of Algorithm 1 under nonconvex and strongly convex settings, respectively, and to demonstrate its robustness and efficiency. We first introduce the tested problems, performance measures, stopping criteria, and parameter settings. We then report the numerical results for the nonconvex robust regression problem and the strongly convex block-angular problem, respectively, and compare pADMM with several ADMM-type baselines.

### 3.1. Problem description and parameter setting

In this subsection, we describe the optimization models used in the experiments and summarize the parameter settings.

For the nonconvex setting, the problem we consider is (Chen et al., 2025)

$$\min_{\mathbf{x},\mathbf{u},\mathbf{v}\in\mathbb{R}^p}\sum_{i=1}^n g(t_i)+\alpha_2\{\alpha_1\Phi_{\lambda,\gamma}(\mathbf{u})+\frac{1}{2}(1-\alpha_2)\|\mathbf{v}\|_2^2\},$$

$$\text{s.t.}\begin{bmatrix}-H & 0_p & I_p\\-H & I_p & 0_p\end{bmatrix}\begin{bmatrix}\mathbf{x}\\\mathbf{v}\\\mathbf{u}\end{bmatrix}=\begin{bmatrix}\mathbf{0}\\\mathbf{0}\end{bmatrix},\quad (15)$$

where $t_i=(A\mathbf{x}-\mathbf{b})_i$, $A=(a_1,a_2,...,a_n)^{\mathrm{T}}\in\mathbb{R}^{n\times p}$ contains $n$ samples; $\mathbf{b}\in\mathbb{R}^n$ denotes the observation, $H\in\mathbb{R}^{p\times p}$ is an invertible transformation matrix, $g$ is a loss function, $\Phi_{\lambda,\gamma}(\mathbf{x})=\sum_{i=1}^p\phi_{\lambda,\gamma}(x_i)$, $\phi_{\lambda,\gamma}(\cdot)$ is a nonconvex function, and the specific form $\phi$ will be given later in Section 3.2. We set $g(t)=\frac{t^2}{1+t^2}$ the German-McClure (Geman & McCulre, 1987) robust loss function, which is non-convex. Moreover, we set $H=I_p$, $\alpha_1=0.3$ and $\alpha_2=0.2$. The primal residual $L_1^{(k)}:=\|\mathbf{u}^{(k)}-H\mathbf{x}^{(k)}\|_2+\|\mathbf{v}^{(k)}-H\mathbf{x}^{(k)}\|_2$ and $L_2^{(k)}:=\|\mathbf{x}^{(k+1)}-\mathbf{x}^{(k)}\|_2+\|\mathbf{v}^{(k+1)}-\mathbf{v}^{(k)}\|_2+\|\mathbf{u}^{(k+1)}-\mathbf{u}^{(k)}\|_2$ are used to evaluate the performance of the tested algorithms. The stopping criterion is $L_1^{(k)}\leq10^{-6}$ or $|L_1^{(k)}-L_1^{(k-1)}|\leq10^{-6}$.

For the strongly convex setting, we consider Problem (16) motivated by the block angular problem (Lam et al., 2021a), which has strongly convex quadratic objective functions and

anti-block-angular constraints. The problem is formulated as

$$\min_{\mathbf{x}_i} \quad \sum_{i=0}^{n} \frac{1}{2} \langle \mathbf{x}_i, Q_i(\mathbf{x}_i) \rangle + \langle \mathbf{c}_i, \mathbf{x}_i \rangle, \tag{16}$$

$$\text{s.t.} \begin{bmatrix} A_{1,0} & A_{1,1} & A_{1,2} & \cdots & A_{1,n-1} & A_{1,n} \\ A_{2,0} & 0 & 0 & \cdots & A_{2,n-1} & 0 \\ \vdots & \vdots & \vdots & \ddots & \vdots & \vdots \\ A_{n,0} & A_{n,1} & 0 & \cdots & 0 & 0 \end{bmatrix} \begin{bmatrix} \mathbf{x}_0 \\ \mathbf{x}_1 \\ \vdots \\ \mathbf{x}_n \end{bmatrix} = \begin{bmatrix} \mathbf{b}_1 \\ \mathbf{b}_2 \\ \vdots \\ \mathbf{b}_n \end{bmatrix},$$

where for each $i = 0, 1, ..., n$ and $j = 2, 3, ..., n$, $Q_i \in \mathbb{R}^{p_i \times p_i}$ is a positive semidefinite matrix; $A_{1,i} \in \mathbb{R}^{q_1 \times p_i}$ : and $A_{j,n-j+1} \in \mathbb{R}^{q_j \times p_{n-j+1}}$ : are given linear matrices; $\mathbf{c}_i \in \mathbb{R}^{p_i}$ and $\mathbf{b}_i \in \mathbb{R}^{q_i}$ are given data. We define the residual error and relative error as $r_1^{(k)} = \| \sum_{i=0}^{n} A_{1,i}(\mathbf{x}_i^{(k)}) - \mathbf{b}_1 \| + \sum_{i=2}^{n} \|A_{i,0}(x_1^{(k)}) + A_{i,n-i+1}(\mathbf{x}_{n-i+1}^{(k)}) - \mathbf{b}_i \|$ and $r_2^{(k)} = \frac{\|\overline{\mathbf{X}}^{(k)} - \overline{\mathbf{X}}_*\|_{\mathrm{F}}}{\|\overline{\mathbf{X}}_*\|_{\mathrm{F}}}$, respectively, where $\overline{\mathbf{X}}_*$ denotes a high-accuracy reference solution for the problem. The stopping criterion is $r_1^{(k)} \leq \epsilon$ or $|r_1^{(k)} - r_1^{(k-1)}| \leq \epsilon$, $\epsilon = 10^{-6}$. We use the last iterate produced by Algorithm 1 with stopping tolerance $\epsilon = 10^{-8}$ as the reference solution $\overline{\mathbf{X}}_*$. The initial iterate $(\overline{\mathbf{\Lambda}}^{(0)}, \overline{\mathbf{X}}^{(0)})$ is set to the all-ones point for both the nonconvex and strongly convex experiments.

Note that it has been shown in (Attouch et al., 2010) that a semialgebraic function has the KL property and that all the objective functions in Examples 2 to 6 and Problems (15) and (16) in numerical experiments are semialgebraic functions. Therefore, they all have the KL property.

### 3.2. Nonconvex robust regression

In this section, we evaluate the performance of Algorithm 1 for solving Problem (15). We set $\phi_{\lambda,\gamma}$ to be the minimax concave penalty (MCP) and smoothly clipped absolute deviation (SCAD) functions (Zhang, 2010). The corresponding algorithms are denoted as pADMM-MCP and pADMM-SCAD, respectively. The MCP function is defined by $\phi_{\lambda,\gamma}(x) = \begin{cases} \lambda|x| - \frac{x^2}{2\lambda}, & \text{if } |x| \leq \gamma\lambda, \\ \frac{\gamma\lambda^2}{2}, & \text{otherwise,} \end{cases}$. The SCAD function is defined by $\phi_{\lambda,\gamma}(x) = \begin{cases} \lambda|x|, & \text{if } |x| \leq \lambda, \\ \frac{-x^2 + 2\gamma\lambda|x| - \lambda^2}{2(\gamma-1)}, & \text{if } \lambda < |x| < \gamma\lambda, \\ \frac{(\gamma+1)\lambda^2}{2}, & \text{if } |x| > \gamma\lambda, \end{cases}$. We solve the subproblem $\mathbf{x}$ via gradient descent, until the gradient norm is less than $10^{-8}$. The subproblems corresponding to $\mathbf{u}$ and $\mathbf{v}$ have closed-form solutions for MCP and SCAD regularizations (see (Chen et al., 2025) for details). For algorithm pADMM-MCP, we choose $\gamma = 0.1, \lambda = 0.2$. For algorithm pADMM-SCAD, we choose $\gamma = 5, \lambda = 0.2$. We compare Algorithm 1 with the classical ADMM

in (Chen et al., 2025) and test the algorithms on the leukemia (Golub et al., 1999) and Alzheimer's disease MRI dataset (Tran et al., 2025). We set $n = 72, p = 7129$ and $n = 200, p = 4200$ for the two datasets, respectively. In the figures, we also include linearized ADMM (Yashtini, 2022) and semiproximal ADMM(Lam et al., 2021b) as additional ADMM-type reference baselines. The test results are shown in Figure 1. In all panels, the curves compare pADMM and classical ADMM under MCP and SCAD regularizations, together with linearized ADMM and semiproximal ADMM as additional ADMM-type reference baselines. The results indicate that pADMM decreases both the residual error and relative error faster than the ADMM-type baselines in CPU time. For more results, please see Tables 5 and 6 in Appendix A.7.

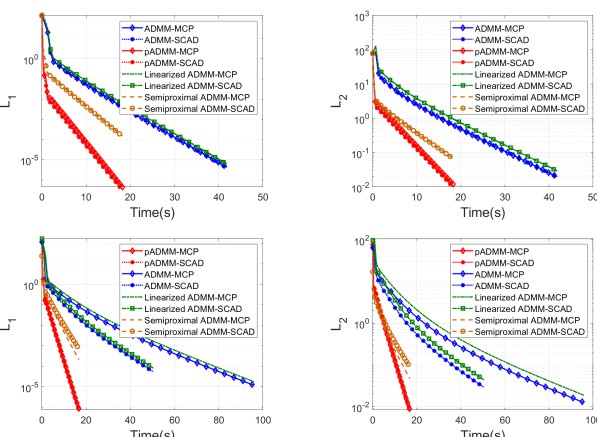

*Figure 1.* Comparison of ADMM-type algorithms for the nonconvex regression experiments.

We further report the convergence behavior of pADMM under different proximal parameter choices for the leukemia dataset in Figure 2 which supports the robustness of the proposed proximal update in the nonconvex regression setting.

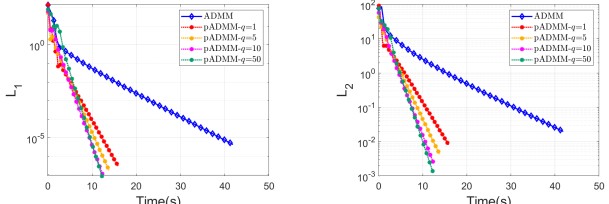

*Figure 2.* Sensitivity to proximal parameters of pADMM on the leukemia dataset.

### 3.3. Strongly convex case

In this section, we test the compared algorithms for Problem (16) using both synthetic and real-world datasets.

### 3.3.1. SYNTHETIC DATA

For the synthetic data, we test the algorithms on two examples. The first example is $n = 30, p_i = q_1 = 50$, and $q_j = 100$, which is denoted by BA1, and the second example is $n = p_i = q_1 = 100$, and $q_j = 400$, which is denoted by BA2, where $i = 0, 1, ..., n$ and $j = 2, 3, ..., n$. The test cases were randomly generated using MATLAB, with the specific syntax: tmp = rand(pi,pi), $Q_i$ = tmp*tmp', $A_{i,j}$ = rand(qi,pi),$C_i$ = rand(pi,1), $\mathbf{b}_1 = \sum_{i=0}^{n} A_{1,i} a$, $\mathbf{b}_j = A_{i,0} a + A_{i,n-i+1} a, a = \text{ones}(p_1, 1), i = [n], j = 2, 3, .., n$. The first row of Figure 3 reports the synthetic-data results which demonstrate that pADMM converges faster than the compared ADMM-type baselines. More numerical results can be found in Table 8 of Appendix A.7.

### 3.3.2. REAL-WORLD DATA

We test (16) on tripart1 and tripart2, which are part of the NMCNF dataset[4]. For the problem taken from tripart1, $n = 16, p_i = q_1 = 192$, and $q_j = 2096$; for problem taken from tripart2, $n = 16, p_i = q_1 = 768$, and $q_j = 8432$, where $i = 0, 1, ..., n, j = 2, 3, ..., n$. The matrices $Q_i, i = 0, \cdots, n$ in (16) are set to be $0.1I$ as has been done in (Castro, 2015). It is shown in Figure 3 that pADMM achieves the target accuracy more efficiently than the compared ADMM-type baselines. The sensitivity of pADMM to different proximal parameter choices is reported in Figure 4. For more results, see Table 7 in Appendix A.7. The results show that the convergence behavior remains stable for the tested parameter choices in the strongly convex real-world setting.

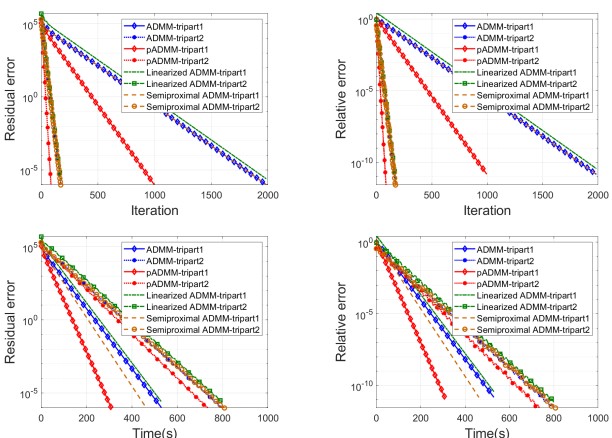

*Figure 3.* Comparison of ADMM-type algorithms for the strongly convex experiments.

---

[4]The tested instances are available from http://www-eio.upc.edu/j̃castro/mmcnf_data.html.

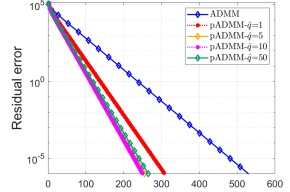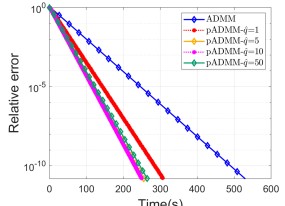

*Figure 4.* Sensitivity to proximal parameters of pADMM on the NMCNF tripart instances.

## 4. Conclusion

In this work, we establish the convergence analysis for the proximal ADMM applied to multi-block problems with block anti-upper triangular constraints. For nonconvex settings, we prove global convergence under a novel assumption that relaxes the traditional range assumption. In the strongly convex case, we establish both global convergence and linear convergence rate. The novel assumption proposed in the work extends the theoretical foundation of algorithms for problems with block anti-upper triangular constraints.

## Impact statement

This paper presents work whose goal is to advance the field of machine learning, particularly in theoretical optimization. Our contribution is the rigorous convergence analysis of proximal ADMM for multi-block problems with structured constraints. As a theoretical study that extends foundational assumptions and convergence guarantees, its direct societal and ethical implications are not immediately apparent. The broader impacts of this work align with those inherent to advancing the mathematical underpinnings of optimization algorithms, and we do not foresee any specific consequences that require highlighting beyond this statement.

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

# A. Appendix

## A.1. Tables of related works

*Table 3.* Some related works of ADMM-type methods for nonconvex problems.

| Algorithm | Optimization problem | Assumptions |
|---|---|---|
| Proximal ADMM (this paper) | $\min_{\mathbf{x}_i} \sum_{i=1}^{n} f_i(\mathbf{x}_i),$ s.t. $\sum_{j=1}^{\min\{n-i+1,m\}} A_{i,j}\mathbf{x}_i = 0, i = 1, \cdots, m.$ | • Assumption 1. |
| Proximal linearized-ADMM (Yashtini, 2022) | $\min_{\mathbf{x},\mathbf{y}} \quad g(\mathbf{x}, \mathbf{y}) + f(\mathbf{x}) + h(\mathbf{y}),$ s.t. $A\mathbf{x} + B\mathbf{y} = 0.$ | • $g, h$ Lipschitz differentiable.
• $g + f + h$ lower bounded, coercive.
• $B$ full column rank.
• $\text{Im}(A) \subseteq \text{Im}(B)$. |
| Proximal ADMM (Boţ & Nguyen, 2020) | $\min_{\mathbf{x},\mathbf{y}} \quad g(\mathbf{x}, \mathbf{y}) + f(\mathbf{x}) + h(\mathbf{y}),$ s.t. $A\mathbf{x} + B\mathbf{y} = 0.$ | • $g, h$ Lipschitz differentiable.
• $g + f + h$ lower bounded, coercive.
• $B$ full column rank.
• $\text{Im}(A) \subseteq \text{Im}(B)$. |
| Linearized ADMM (Liu et al., 2019) | $\min_{\mathbf{x},\mathbf{y}} \quad g(\mathbf{x}, \mathbf{y}) + f(\mathbf{x}) + h(\mathbf{y}),$ s.t. $A\mathbf{x} + B\mathbf{y} = 0.$ | • $g, h$ Lipschitz differentiable.
• $g + f + h$ lower bounded, coercive.
• $B$ full column rank.
• $\text{Im}(A) \subseteq \text{Im}(B)$. |
| ADMM (Wang et al., 2019) | $\min_{\mathbf{x},\mathbf{y}} \quad g(\mathbf{x}) + \sum_i f_i(\mathbf{x}_i) + h(\mathbf{y}),$ s.t. $A\mathbf{x} + B\mathbf{y} = 0.$ | • $g, h$ Lipschitz differentiable.
• $f_0$ lower semi-continuous, $f_i$ prox-regular.
• $\text{Im}(A) \subseteq \text{Im}(B)$.
• Subproblems have Lipschitz continuous solutions. |
| ADMM (Wang et al., 2019) | $\min_{\mathbf{x},\mathbf{y}} \quad \phi(\mathbf{x}, \mathbf{y}),$ s.t. $A\mathbf{x} + B\mathbf{y} = 0.$ | • $\phi$ is Lipschitz differentiable and coercive.
• $\text{Im}(A) \subseteq \text{Im}(B)$.
• Subproblems have Lipschitz continuous solutions. |
| Flexible ADMM (Hong et al., 2016) | $\min_{\mathbf{x}_k,\mathbf{x}_0} \sum_k g_k(\mathbf{x}_k) + l(\mathbf{x}_0),$ s.t. $\sum_k A_k\mathbf{x}_k = \mathbf{x}_0, \mathbf{x}_k \in \mathbf{x}_k.$ | • $l$ Lipschitz differentiable; $g_k$ Lipschitz or convex.
• $f$ is lower bounded.
• $A_k$ are all full column rank. |
| Bregman ADMM (Wang et al., 2018) | $\min_{\mathbf{x},\mathbf{y},\mathbf{z}} \quad f(\mathbf{x}) + g(\mathbf{y}) + h(\mathbf{z}),$ s.t. $A\mathbf{x} + B\mathbf{y} + C\mathbf{z} = 0.$ | • $f, g$ are proper and lower semi-continuous.
• $h$ is smooth, Lipschitz continuous; $f + g$ is coercive.
• $f + g + h$ subanalytic; $f, g$ are bounded.
• $h(\mathbf{z}) - \beta_0\|\nabla h(\mathbf{z})\|_F^2$ is bounded.
• $C$ has full row rank. |
| Prox-gradient ADMM (Jiang et al., 2019) | $\min_{\mathbf{x}_i} \quad f(\mathbf{x}_1, \cdots, \mathbf{x}_n) + \sum_i r_i(\mathbf{x}_i),$ s.t. $\sum_i A_i\mathbf{x}_i = b, \mathbf{x}_i \in X_i.$ | • $f$ is Lipschitz differentiable; $f$ and $r_i$ are lower bounded. Either $r_i$ is Lipschitz continuous and $X_i$ is compact, or $r_i$ is lower semi-continuous and $X_i = \mathbb{R}^{n_i}$.
• $A_n = I$. (not needed for proximal majorized ADMM) |
| ADMM (Yang et al., 2017) | $\min_{\mathbf{L},\mathbf{S},\mathbf{Z}} \quad \Psi(\mathbf{L}) + \Phi(\mathbf{S}) + \frac{1}{2}\|\mathbf{D} - \mathcal{A}(\mathbf{Z})\|_F^2,$ s.t. $\mathcal{B}(\mathbf{L}) + \mathcal{C}(\mathbf{S}) = \mathbf{Z}.$ | • $\Psi, \Phi$ are proper, closed and nonnegative.
• $\Psi$ is convex and continuous in the domain.
• $\liminf \Psi + \Phi > \Theta(\mathbf{L}^{(1)}, \mathbf{S}^{(1)}, \mathbf{Z}^{(1)}, \mathbf{\Lambda}^{(1)})$, where $\mathbf{L}^{(1)}, \mathbf{S}^{(1)}, \mathbf{Z}^{(1)}, \mathbf{\Lambda}^{(1)}$ denote the first iterate, and $\Theta$ is the predefined potential function.
• $\mathcal{B}^T\mathcal{B} \succ 0, \mathcal{C}^T\mathcal{C} \succ 0$. |
| Proximal ADMM (Li & Pong, 2015) | $\min_{\mathbf{x},\mathbf{y}} \quad h(\mathbf{x}) + p(\mathbf{y}),$ s.t. $\mathbf{y} = \mathcal{M}(\mathbf{x}).$ | • $p$ is proper, closed and proximal.
• $h$ is twice continuously differentiable with a bounded Hessian.
• $\mathcal{M}^T\mathcal{M} \succeq \sigma I$. |

## A.2. Proof of Lemma 1

*Proof.* Consider $(iii)$. Note that the $\beta_j$ in the definition of $\theta_{3,j}, j = 1, \cdots, m$ are in the denominator, it follows that there exist $\tilde{\beta}_j, j = 1, \cdots, m$ such that $(iii)$ in A(4) holds if $\beta_j > \tilde{\beta}_j, j = 1, \cdots, m$.

Consider $(ii)$. We first let $\beta_i > M$ for a given constant $M > 0$ and $i = 1 \cdots, m$.

*Table 4.* Some related works of ADMM-type methods using strong convexity.

| Algorithm | Optimization problem | Assumptions |
|---|---|---|
| Proximal ADMM (this paper) | $\min_{\mathbf{x}_i} \sum_{i=1}^n f_i(\mathbf{x}_i),$ s.t. $\sum_{j=1}^{\min\{n-i+1,m\}} A_{i,j}\mathbf{x}_i = 0, i = 1, \cdots, m.$ | • Assumptions 2 and 3, Tables 1 and 2. |
| ADMM (Lin et al., 2015a) | $\min_{\mathbf{x}_i} \sum_{i=1}^n f_i(\mathbf{x}_i),$ s.t. $\sum_{i=1}^n A_i\mathbf{x}_i = 0.$ | • Assumptions 2 and 3. 
 • Strongly convex: $f_2, \ldots, f_N$ 
 • $f_N$ is Lipchitz differentiable, $A_N$ has full row rank or $f_N$ is strongly convex, $f_1, \ldots, f_N$ are Lipchitz differentiable, or $f_1, \ldots, f_N$ are Lipchitz differentiable, $A_1$ has full column rank. |
| Proximal ADMM (Deng & Yin, 2016) | $\min_{\mathbf{x},\mathbf{y}} \quad f(\mathbf{x}) + g(\mathbf{y}),$ s.t. $A\mathbf{x} + B\mathbf{y} = \mathbf{b}.$ | • Assumptions 2 and 3. 
 • Tables 1 and 2 for $m = 1$ and $n = 2$. |
| ADMM (Lin et al., 2018) | $\min_{\mathbf{x},\mathbf{y}} \quad f_1(\mathbf{x}_1) + f_2(\mathbf{x}_2) + f_3(\mathbf{x}_3),$ s.t. $A_1\mathbf{x}_1 + A_2\mathbf{x}_2 + A_3\mathbf{x}_3 = \mathbf{b}.$ | • Assumptions 2 and 3. 
 • $f_1$ and $f_2$ are lower semi-continuous and coercive. 
 • $f_3$ is lower bounded and strongly convex. $f_3(\mathbf{x})$ is Lipchitz differentiable. 
 • $A_1$ and $A_2$ have full column rank. 
 • The condition number of $f_3$ is in [1,1.0798). |
| ADMM (Chen et al., 2025) | $\min_{\mathbf{x}_1,\mathbf{x}_2,\mathbf{x}_3} \quad \theta_1(\mathbf{x}_1) + \theta_2(\mathbf{x}_2) + \theta_3(\mathbf{x}_3),$ s.t. $A_1\mathbf{x}_1 + A_2\mathbf{x}_2 + A_3\mathbf{x}_3 = \mathbf{b}.$ | • Assumptions 2 and 3. The sublevel set of the objective function is nonempty. 
 • $\theta_1$ is weakly convex with modules $\mu_1$, $\theta_2, \theta_3$ are strongly convex with modules $\mu_2$ and $\mu_3$, respectively; 
 • $A_1$ has full column rank. 
 • $\mu_1, \mu_2,$ and $\mu_3$ satisfy some given equalities. |

- If $i = n$ in (6), then

$$\theta_{0,n} = \frac{2}{\beta_1}c_{1,1}\eta_{0,n}^2 + \sum_{i=2}^m \frac{2i}{\beta_i}c_A^2\left(c_{1,i}\eta_{0,n}\right)^2 \le \frac{2}{\beta_1}c_{1,1}\eta_{0,n}^2 + \sum_{i=2}^m \frac{2i}{M}c_A^2\left(c_{1,i}\eta_{0,n}\right)^2,$$

$$\theta_{1,n} = \frac{2}{\beta_1}c_{1,1}\eta_{1,n}^2 + \sum_{i=2}^m \frac{2i}{\beta_i}c_A^2\left(c_{n-l+1,i}\eta_{1,l}\right)^2 \le \frac{2}{\beta_1}c_{1,1}\eta_{1,n}^2 + \sum_{i=2}^m \frac{2i}{M}c_A^2\left(c_{n-l+1,i}\eta_{1,l}\right)^2,$$

where we use $\bar{\beta}_1 = 0$. Combining the above inequalities with A(3) that Inequality (6) holds with $i = n$ for sufficiently large $\beta_1$;

- Suppose the penality parameters $\beta_1, \cdots, \beta_k$ are found such that (6) holds for $i = n - k + 1, \ldots, n$ with undetermined $\beta_{k+1}, \cdots, \beta_m$. Consider $i = n - k$. We have

$$\theta_{0,n-k} = \frac{2(k+1)}{\beta_{k+1}}c_{k+1,k+1}\eta_{0,n-k}^2 + \sum_{i=k+2}^m \frac{2i}{\beta_i}c_A^2\left(\sum_{p=k+2}^i \bar{\beta}_{k+1}c_{p,i}d_A^2 + c_{k+1,i}\eta_{0,n-k}\right)^2$$

$$\le \frac{2(k+1)}{\beta_{k+1}}c_{k+1,k+1}\eta_{0,n-k}^2 + \sum_{i=k+2}^m \frac{2i}{M}c_A^2\left(\sum_{p=k+2}^i \bar{\beta}_{k+1}c_{p,i}d_A^2 + c_{k+1,i}\eta_{0,n-k}\right)^2,$$

$$\theta_{1,n-k} = \frac{2(k+1)}{\beta_{k+1}}c_{k+1,k+1}\eta_{1,n-k}^2 + \sum_{i=k+2}^m \frac{2i}{\beta_i}c_A^2\left(\sum_{p=k+2}^i \bar{\beta}_{k+1}c_{p,i}d_A^2 + c_{k+1,i}\eta_{1,n-k}\right)^2$$

$$\le \frac{2(k+1)}{\beta_{k+1}}c_{k+1,k+1}\eta_{1,n-k}^2 + \sum_{i=k+2}^m \frac{2i}{M}c_A^2\left(\sum_{p=k+2}^i \bar{\beta}_{k+1}c_{p,i}d_A^2 + c_{k+1,i}\eta_{1,n-k}\right)^2,$$

where $\bar{\beta}_{k+1}$ is determined since $\beta_1, \cdots, \beta_k$ are found. Therefore, one can choose a sufficiently large $\beta_{k+1}$ such that (6) holds for $i = n - k$ due to A(3). The conclusion holds by induction.

$\square$

Lemma 3 shows that the augmented Lagrangian function $\mathcal{L}$ has a sufficient decrease property when updating $\overline{\mathbf{X}}$. It is widely used in many convergence analyses for Problem (5), see e.g., (Yashtini, 2022; Deng & Yin, 2016; Li & Pong, 2015). However, such a result has not been given for Problem (1). This lemma is used in Lemma 4 to bound $\mathcal{L}(\overline{\mathbf{X}}^{(k)}, \overline{\mathbf{\Lambda}}^{(k)}) - \mathcal{L}(\overline{\mathbf{X}}^{(k+1)}, \overline{\mathbf{\Lambda}}^{(k+1)})$.

**Lemma 3** (Sufficient decrease of $\mathcal{L}$ for $\overline{\mathbf{X}}$ update). *Suppose A(1) in Assumption 1 holds. The sequence $\{(\overline{\mathbf{X}}^{(k)}, \overline{\mathbf{\Lambda}}^{(k)})\}_{k \geq 0}$ generated by Algorithm 1 satisfies:*

$$\mathcal{L}(\overline{\mathbf{X}}^{(k)}, \overline{\mathbf{\Lambda}}^{(k)}) - \mathcal{L}(\overline{\mathbf{X}}^{(k+1)}, \overline{\mathbf{\Lambda}}^{(k)}) \geq \sum_{i=1}^{n-m} \frac{q_i^-}{2} \|\Delta\mathbf{X}_i^{(k+1)}\|_{\mathrm{F}}^2 + \sum_{i=n-m+1}^{n} \|\Delta\mathbf{X}_i^{(k+1)}\|_{\mathcal{B}_i^{(k)}}^2,$$

*where $\mathcal{B}_i^{(k)} = (\mathcal{Q}_i^{(k)} - L_{f_i})\mathcal{I} + \frac{\beta_{n-i+1}}{2} \mathcal{A}_{n-i+1,i}^{\mathrm{T}} \mathcal{A}_{n-i+1,i}$.*

*Proof.* Define $P_i = \sum_{j=1}^{m} \frac{\beta_j}{2} \|\mathcal{A}_{j,1}(\mathbf{X}_1^{(k+1)}) + \cdots + \mathcal{A}_{j,i}(\mathbf{X}_{i-1}^{(k+1)}) + \mathcal{A}_{j,i}(\mathbf{X}_i^{(k)}) + \cdots + \mathcal{A}_{j,n-j+1}\mathbf{X}_{n-j+1}^{(k)} - \mathbf{b}_j + \mathbf{\Lambda}_j^{(k)}/\beta_j\|_{\mathrm{F}}^2$ for $i = 1, \cdots, n-m$. It follows from the $\mathbf{X}_i^{(k+1)}$ update in Algorithm 1 that $f(\mathbf{X}_i^{(k+1)}) + P_{i+1} + \frac{1}{2}\|\Delta\mathbf{X}_i^{(k+1)}\|_{\mathcal{Q}_i^{(k)}}^2 \leq f(\mathbf{X}_i^{(k)}) + P_i$, for $i = 1, \cdots, n-m$. Hence, we have

$$\begin{aligned}
\mathcal{L}&(\overline{\mathbf{X}}^{(k)}, \overline{\mathbf{\Lambda}}^{(k)}) - \mathcal{L}(\mathbf{X}_1^{(k+1)}, \cdots \mathbf{X}_{n-m}^{(k+1)}, \mathbf{X}_{n-m+1}^{(k)}, \cdots, \mathbf{X}_n^{(k)}, \overline{\mathbf{\Lambda}}^{(k)}) \\
&= \sum_{l=1}^{n-m} f_l(\mathbf{X}_l^{(k)}) - f_l(\mathbf{X}_l^{(k+1)}) + \sum_{l=1}^{n-m} P_l - P_{l+1} \geq \sum_{i=1}^{n-m} \frac{1}{2}\|\Delta\mathbf{X}_i^{(k+1)}\|_{\mathcal{Q}_i^{(k)}}^2.
\end{aligned} \tag{17}$$

For $i = n-m+1, \cdots, n$, it follows from the optimality condition of $\mathbf{X}_i^{(k+1)}$ that

$$-\sum_{j=1}^{n-i+1} \beta_j \mathcal{A}_{j,i}^{\mathrm{T}}(\sum_{l=1}^{i} \mathcal{A}_{j,l}(\mathbf{X}_l^{(k+1)}) + \sum_{l=i+1}^{n-j+1} \mathcal{A}_{j,l}(\mathbf{X}_l^{(k)}) - \mathbf{b}_j + \mathbf{\Lambda}_j^{(k)}/\beta_j) - (\mathcal{Q}_i^{(k)}\Delta\mathbf{X}_i^{(k+1)}) = \nabla f_i(\mathbf{X}_i^{(k+1)}). \tag{18}$$

Hence, we have

$$\mathcal{L}(\mathbf{X}_1^{(k+1)}, \cdots \mathbf{X}_{n-m}^{(k+1)}, \mathbf{X}_{n-m+1}^{(k)}, \cdots, \mathbf{X}_n^{(k)}, \overline{\mathbf{\Lambda}}^{(k)}) - \mathcal{L}(\overline{\mathbf{X}}^{(k+1)}, \overline{\mathbf{\Lambda}}^{(k)})$$

$$= \sum_{i=n-m+1}^{n} \mathcal{L}(\mathbf{X}_1^{(k+1)}, \cdots \mathbf{X}_{i-1}^{(k+1)}, \mathbf{X}_i^{(k)}, \cdots, \mathbf{X}_n^{(k)}, \overline{\mathbf{\Lambda}}^{(k)}) - \mathcal{L}(\mathbf{X}_1^{(k+1)}, \cdots \mathbf{X}_i^{(k+1)}, \mathbf{X}_{i+1}^{(k)}, \cdots, \mathbf{X}_n^{(k)}, \overline{\mathbf{\Lambda}}^{(k)})$$

$$= \sum_{i=n-m+1}^{n} f_i(\mathbf{X}_i^{(k)}) - f_i(\mathbf{X}_i^{(k+1)}) + \sum_{j=1}^{n-i+1} \frac{\beta_j}{2}\|\sum_{l=1}^{i} \mathcal{A}_{j,i}(\mathbf{X}_i^{(k+1)}) + \sum_{l=i+1}^{n-j+1} \mathcal{A}_{j,i}(\mathbf{X}_i^{(k)}) - \mathbf{b}_j + \frac{\mathbf{\Lambda}_j^{(k)}}{\beta_i}\|_{\mathrm{F}}^2$$

$$- \sum_{i=n-m+1}^{n} \sum_{j=1}^{n-i+1} \frac{\beta_j}{2}\|\sum_{l=1}^{i+1} \mathcal{A}_{j,i}(\mathbf{X}_i^{(k+1)}) + \sum_{l=i+2}^{n-j+1} \mathcal{A}_{j,i}(\mathbf{X}_i^{(k)}) - \mathbf{b}_j + \frac{\mathbf{\Lambda}_j^{(k)}}{\beta_i}\|_{\mathrm{F}}^2$$

$$= \sum_{i=n-m+1}^{n} f_i(\mathbf{X}_i^{(k)}) - f_i(\mathbf{X}_i^{(k+1)}) + \sum_{j=1}^{n-i+1} \frac{\beta_j}{2}\|\mathcal{A}_{j,i}(\Delta\mathbf{X}_i^{(k+1)})\|_{\mathrm{F}}^2$$

$$- \sum_{i=n-m+1}^{n} \sum_{j=1}^{n-i+1} \beta_j \langle \mathcal{A}_{j,i}^{\mathrm{T}}(\sum_{l=1}^{i} \mathcal{A}_{j,l}(\mathbf{X}_l^{(k+1)}) + \sum_{l=i+1}^{n-j+1} \mathcal{A}_{j,l}(\mathbf{X}_l^{(k)}) - \mathbf{b}_j + \frac{\mathbf{\Lambda}_j^{(k)}}{\beta_i}), \Delta\mathbf{X}_i^{(k+1)} \rangle + \frac{1}{2}\|\Delta\mathbf{X}_i^{(k+1)}\|_{\mathcal{Q}_i^{(k)}}^2$$

$$\overset{(18)}{\geq} \sum_{i=n-m+1}^{n} f_i(\mathbf{X}_i^{(k)}) - f_i(\mathbf{X}_i^{(k+1)}) - \langle \nabla f_i(\mathbf{X}_i^{(k+1)}), \mathbf{X}^{(k+1)} - \mathbf{X}^{(k)} \rangle + \frac{1}{2}\|\Delta\mathbf{X}_i^{(k+1)}\|_{\mathcal{Q}_i^{(k)}}^2$$

$$+ \sum_{i=n-m+1}^{n} \sum_{j=1}^{n-i+1} \frac{\beta_j}{2}\|\mathcal{A}_{j,i}\Delta\mathbf{X}_i^{(k+1)}\|_{\mathrm{F}}^2$$

$$\overset{A(3)}{\geq} \frac{1}{2}\sum_{i=n-m+1}^{n} \|\Delta\mathbf{X}_i^{(k+1)}\|_{\mathcal{B}_i^{(k)}}^2. \tag{19}$$

The conclusion follows from Inequalities (17) and (19). □

### A.3. Proof of Theorem 1

**Lemma 4.** *Suppose A(1) in Assumption 1 holds. The sequence $\{(\overline{\mathbf{X}}^{(k)}, \overline{\mathbf{\Lambda}}^{(k)})\}_{k \geq 0}$ satisfies*

$$\mathcal{L}(\overline{\mathbf{X}}^{(k+1)}, \overline{\mathbf{\Lambda}}^{(k+1)}) \leq \mathcal{L}(\overline{\mathbf{X}}^{(k)}, \overline{\mathbf{\Lambda}}^{(k)}) - \sum_{i=1}^{n-m} \frac{q_i}{2} \|\Delta\mathbf{X}_i^{(k+1)}\|_{\mathrm{F}}^2 - \sum_{i=n-m+1}^{n} \|\Delta\mathbf{X}_i^{(k+1)}\|_{\mathcal{B}_i^{(k)}}^2 + \sum_{i=1}^{m} \frac{1}{\beta_i} \|\Delta\mathbf{\Lambda}_i^{(k+1)}\|_{\mathrm{F}}^2.$$

*Proof.* According to the update of $\mathbf{\Lambda}_i^{(k+1)}$ in Algorithm 1 and Lemma 3, we have

$$\mathcal{L}(\overline{\mathbf{X}}^{(k+1)}, \overline{\mathbf{\Lambda}}^{(k+1)}) = \mathcal{L}(\overline{\mathbf{X}}^{(k+1)}, \overline{\mathbf{\Lambda}}^{(k)}) + \sum_{i=1}^{m} \frac{1}{\beta_i} \|\Delta\mathbf{\Lambda}_i^{(k+1)}\|_{\mathrm{F}}^2$$

$$\leq \mathcal{L}(\overline{\mathbf{X}}^{(k)}, \overline{\mathbf{\Lambda}}^{(k)}) - \sum_{i=n-m+1}^{n} \|\Delta\mathbf{X}_i^{(k+1)}\|_{\mathcal{B}_i^{(k)}}^2 - \sum_{i=1}^{n-m} \frac{q_i}{2} \|\Delta\mathbf{X}_i^{(k+1)}\|_{\mathrm{F}}^2 + \sum_{i=1}^{m} \frac{1}{\beta_i} \|\Delta\mathbf{\Lambda}_i^{(k+1)}\|_{\mathrm{F}}^2,$$

which completes the proof. □

Lemma 5 presents a recursive relationship that is used in Lemma 6 to bound the error $\|\Delta\mathbf{\Lambda}_i^{(k+1)}\|, i = 1, \cdots, m$. Note that Lemma 5 does not exist in the existing works (Yashtini, 2022; Deng & Yin, 2016; Li & Pong, 2015) for the analysis of Problem (5).

**Lemma 5.** *Given the recursive inequality $L_i \leq a_i F_i + \sum_{j=1}^{i-1} b_{i,j} L_j, i = 1, 2, \ldots, n$, where $a_i > 0$ and $b_{i,j} \geq 0$ are constants, it holds that $L_i \leq \sum_{k=1}^{i} a_k c_{k,i} F_k$, for $i = 1, \ldots, n$, where the coefficients $c_{k,i}$ are defined as*

$$c_{k,i} = \begin{cases} 1, & \text{if } k = i, \\ \sum_{j=k}^{i-1} b_{i,j}\, c_{k,j}, & \text{if } 1 \leq k < i. \end{cases} \tag{20}$$

*Proof.* We proceed by induction on $i$. For the case $i = 1$, it follows from the recursive inequality that $L_1 \leq a_1 F_1 + \sum_{j=1}^{0} b_{1,j} L_j = a_1 F_1 = a_1 c_{1,1} F_1$. Hence, the statement holds for $i = 1$.

Assume that the statement is true for all indices $j < i$, that is, for each $j = 1, \ldots, i-1$, inequality $L_j \leq \sum_{k=1}^{j} a_k c_{k,j} F_k$ holds. Applying the recursive inequality for $L_i$ and the induction hypothesis, we obtain

$$L_i \leq a_i F_i + \sum_{j=1}^{i-1} b_{i,j} L_j \leq a_i F_i + \sum_{j=1}^{i-1} b_{i,j} \left( \sum_{k=1}^{j} a_k c_{k,j} F_k \right).$$

Interchanging the order of summation gives $L_i \leq a_i F_i + \sum_{k=1}^{i-1} a_k F_k \left( \sum_{j=k}^{i-1} b_{i,j}\, c_{k,j} \right)$.

Now we examine the inner sum. By the recursive definition of $c_{k,i}$, for $1 \leq k < i$ we have $c_{k,i} = \sum_{j=k}^{i-1} b_{i,j}\, c_{k,j}$. For $k = i$, we have $c_{i,i} = 1$ by definition. Therefore,

$$L_i \leq a_i F_i + \sum_{k=1}^{i-1} a_k F_k\, c_{k,i} = \sum_{k=1}^{i} a_k c_{k,i} F_k,$$

which completes the inductive step.

□

Lemma 4 presents the sufficient decrease property of $\mathcal{L}(\overline{\mathbf{X}}, \overline{\mathbf{\Lambda}})$ under one step update of $\overline{\mathbf{X}}^{(k+1)}$ and $\overline{\mathbf{\Lambda}}^{(k+1)}$.

**Lemma 6.** *Suppose A(2),A(3) in Assumption 1 hold, it follows that*

$$\mathcal{L}(\overline{\mathbf{X}}^{(k+1)}, \overline{\mathbf{\Lambda}}^{(k+1)}) + \sum_{i=n-m+1}^{n} \|\Delta\mathbf{X}_i^{(k+1)}\|_{\mathcal{B}_i^{(k)}-r\theta_{1,i}\mathcal{I}}^2 + \sum_{i=1}^{n-m} q_i\|\Delta\mathbf{X}_i^{(k+1)}\|_{\mathrm{F}}^2 + \sum_{i=1}^{m} \frac{r-1}{\beta_i}\|\Delta\mathbf{\Lambda}_i^{(k+1)}\|_{\mathrm{F}}^2$$

$$\leq \mathcal{L}(\overline{\mathbf{X}}^{(k)}, \overline{\mathbf{\Lambda}}^{(k)}) + \sum_{i=n-m+1}^{n} r\theta_{0,i}\|\Delta\mathbf{X}_i^{(k)}\|_{\mathrm{F}}^2, \tag{21}$$

*where $r > 1, \theta_{0,i}, \theta_{1,i}$ are defined in* (7).

*Proof.* The optimality condition of $\mathbf{X}_i, i = n - m + 1, \cdots, n$ yields that

$$-\sum_{j=1}^{n-i+1} \beta_j\mathcal{A}_{j,i}^{\mathrm{T}}(\sum_{l=1}^{i}\mathcal{A}_{j,l}(\mathbf{X}_l^{(k+1)}) + \sum_{l=i+1}^{n-j+1}\mathcal{A}_{j,l}(\mathbf{X}_l^{(k)}) - \mathbf{b}_j + \mathbf{\Lambda}_j^{(k)}/\beta_j) - (\mathcal{Q}_i^{(k)}\Delta\mathbf{X}_i^{(k+1)}) = \nabla f_i(\mathbf{X}_i^{(k+1)}). \tag{22}$$

Define the matrix

$$\mathbf{G}_i^{(k+1)} := \nabla f_{n-i+1}(\mathbf{X}_{n-i+1}^{(k+1)}) + (\mathcal{Q}_{n-i+1}^{(k)}\Delta\mathbf{X}_{n-i+1}^{(k+1)}) + \sum_{j=1}^{i-1}(\mathcal{A}_{j,n-i+1}^{\mathrm{T}}\mathbf{\Lambda}_j^{(k+1)})$$

$$- \sum_{l=n-i+2}^{n}\sum_{j=1}^{n-l+1} \beta_j\mathcal{A}_{j,n-i+1}^{\mathrm{T}}\mathcal{A}_{j,l}\Delta\mathbf{X}_l^{(k+1)}, \text{ for } i = 1, \cdots, m. \tag{23}$$

We first prove that the sequence $\{(\overline{\mathbf{X}}^{(k)}, \overline{\mathbf{\Lambda}}^{(k)})\}_{k\geq 0}$ generated by Algorithm 1 satisfies

$$\sum_{i=1}^{m}\frac{1}{\beta_i}\|\Delta\mathbf{\Lambda}_i^{(k+1)}\|_{\mathrm{F}}^2 \leq \sum_{i=n-m+1}^{n}\theta_{0,i}\|\Delta\mathbf{X}_i^{(k)}\|_{\mathrm{F}}^2 + \sum_{i=n-m+1}^{n}\theta_{1,i}\|\Delta\mathbf{X}_i^{(k+1)}\|_{\mathrm{F}}^2.$$

Furthermore,

It follows that $\Delta\mathbf{G}_i^{(k+1)} = \nabla f_{n-i+1}(\mathbf{X}_{n-i+1}^{(k+1)}) - \nabla f_{n-i+1}(\mathbf{X}_{n-i+1}^{(k)}) + \mathcal{Q}_{n-i+1}^{(k)}(\Delta\mathbf{X}_{n-i+1}^{(k)}) - \mathcal{Q}_{n-i+1}^{(k-1)}(\Delta\mathbf{X}_{n-i+1}^{(k-1)}) + \sum_{j=1}^{i-1}(\mathcal{A}_{j,n-i+1}^{\mathrm{T}}\Delta\mathbf{\Lambda}_j^{(k+1)}) - \sum_{l=n-i+2}^{n}\sum_{j=1}^{n-l+1}\beta_j\mathcal{A}_{j,n-i+1}^{\mathrm{T}}\mathcal{A}_{j,l}\Delta\mathbf{X}_l^{(k+1)} + \sum_{l=n-i+2}^{n}\sum_{j=1}^{n-l+1}\beta_j\mathcal{A}_{j,n-i+1}^{\mathrm{T}}\mathcal{A}_{j,l}\Delta\mathbf{X}_l^{(k)}$. By A(2) in Assumption 1, $\nabla f_i(\mathbf{X}_i)$ is $L_{f_i}$ Lipschitz continuous, $i = n - m + 1, \cdots, n$. By the optimality condition of $\mathbf{X}_{n-i+1}$ subproblem, we have

$$\mathbf{G}_i^{(k+1)} + \mathcal{A}_{i,n-i+1}^{\mathrm{T}}\mathbf{\Lambda}_i^{(k+1)} = 0. \tag{24}$$

It follows from the triangle inequality that for $i = 1, \cdots, m$:

$$\frac{1}{c_A}\|\Delta\mathbf{\Lambda}_i^{(k+1)}\|_{\mathrm{F}} \leq \|\mathcal{A}_{i,n-i+1}^{\mathrm{T}}\Delta\mathbf{\Lambda}_i^{(k+1)}\|_{\mathrm{F}} = \|\Delta\mathbf{G}_i^{(k+1)}\|_{\mathrm{F}}$$

$$\leq (L_{f_{n-i+1}} + q_{n-i+1})\|\Delta\mathbf{X}_{n-i+1}^{(k+1)}\|_{\mathrm{F}} + q_{n-i+1}\|\Delta\mathbf{X}_{n-i+1}^{(k)}\|_{\mathrm{F}} + \sum_{l=n-i+2}^{n}\|\sum_{j=1}^{n-l+1}\beta_j\mathcal{A}_{j,i}^{\mathrm{T}}\mathcal{A}_{j,n-i+1}\Delta\mathbf{X}_l^{(k+1)}\|_{\mathrm{F}}$$

$$+ \sum_{l=n-i+2}^{n}\|\sum_{j=1}^{n-l+1}\beta_j\mathcal{A}_{j,i}^{\mathrm{T}}\mathcal{A}_{j,n-i+1}\Delta\mathbf{X}_l^{(k)}\|_{\mathrm{F}} + \sum_{j=1}^{i-1}\|\mathcal{A}_{j,n-i+1}^{\mathrm{T}}\Delta\mathbf{\Lambda}_j^{(k+1)}\|_{\mathrm{F}}$$

$$\leq (L_{f_{n-i+1}} + q_{n-i+1})\|\Delta\mathbf{X}_{n-i+1}^{(k+1)}\|_{\mathrm{F}} + q_{n-i+1}\|\Delta\mathbf{X}_{n-i+1}^{(k)}\|_{\mathrm{F}} + \sum_{l=n-i+2}^{n}\bar{\beta}_{n-l+1}d_A^2\|\Delta\mathbf{X}_l^{(k+1)}\|_{\mathrm{F}}$$

$$+ \sum_{l=n-i+2}^{n}\bar{\beta}_{n-l+1}d_A^2\|\Delta\mathbf{X}_l^{(k)}\|_{\mathrm{F}} + \sum_{j=1}^{i-1}d_A\|\Delta\mathbf{\Lambda}_j^{(k+1)}\|_{\mathrm{F}}.$$

According to Lemma 5, for $i = 1, 2, \ldots, m$, we have:

$$\|\Delta\boldsymbol{\Lambda}_i^{(k+1)}\|_{\mathrm{F}} \leq \sum_{p=1}^{i} c_{p,i} c_A (\eta_{1,n-p+1}\|\Delta\mathbf{X}_{n-p+1}^{(k+1)}\|_{\mathrm{F}} + \eta_{0,n-p+1}\|\Delta\mathbf{X}_{n-p+1}^{(k)}\|_{\mathrm{F}})$$

$$+ \sum_{p=1}^{i} \bar{\beta}_{n-l+1} c_{p,i} \left( \sum_{l=n-p+2}^{n} c_A d_A^2 \|\Delta\mathbf{X}_l^{(k)}\|_{\mathrm{F}} + \sum_{l=n-p+2}^{n} c_A d_A^2 \|\Delta\mathbf{X}_l^{(k+1)}\|_{\mathrm{F}} \right)$$

$$\leq \sum_{l=n-i+1}^{n} c_A c_{n-l+1,i} \left( \eta_{1,l}\|\Delta\mathbf{X}_l^{(k+1)}\|_{\mathrm{F}} + \eta_{0,l}\|\Delta\mathbf{X}_l^{(k)}\|_{\mathrm{F}} \right)$$

$$+ \sum_{l=n-i+2}^{n} \sum_{p=n-l+2}^{i} \bar{\beta}_{n-l+1} c_{p,i} c_A d_A^2 \left( \|\Delta\mathbf{X}_l^{(k)}\|_{\mathrm{F}} + \|\Delta\mathbf{X}_l^{(k+1)}\|_{\mathrm{F}} \right),$$

where $\eta_{1,i}, \eta_{0,i}, c_{i,j}$ are defined in A(4) in Assumption 1. Consequently, we have

$$\frac{1}{\beta_i}\|\Delta\boldsymbol{\Lambda}_i^{(k+1)}\|_{\mathrm{F}}^2 \leq \frac{2i}{\beta_i} \left( c_{i,i}^2 c_A^2 \left( \eta_{1,n-i+1}^2\|\Delta\mathbf{X}_{n-i+1}^{(k+1)}\|_{\mathrm{F}}^2 + \eta_{0,n-i+1}^2\|\Delta\mathbf{X}_{n-i+1}^{(k)}\|_{\mathrm{F}}^2 \right) \right)$$

$$+ \frac{2i}{\beta_i} c_A^2 \sum_{l=n-i+2}^{n} \left( \sum_{p=n-l+2}^{i} \bar{\beta}_{n-l+1} c_{p,i} d_A^2 + c_{n-l+1,i}\eta_{0,l} \right)^2 \|\Delta\mathbf{X}_l^{(k)}\|_{\mathrm{F}}^2$$

$$+ \frac{2i}{\beta_i} c_A^2 \sum_{l=n-i+2}^{n} \left( \sum_{p=n-l+2}^{i} \bar{\beta}_{n-l+1} c_{p,i} d_A^2 + c_{n-l+1,i}\eta_{1,l} \right)^2 \|\Delta\mathbf{X}_l^{(k+1)}\|_{\mathrm{F}}^2,$$

Summing both sides of the inequality over $i$ and we have

$$\sum_{i=1}^{m} \frac{1}{\beta_i}\|\Delta\boldsymbol{\Lambda}_i^{(k+1)}\|_{\mathrm{F}}^2$$

$$\leq \sum_{i=1}^{m} \frac{2i}{\beta_i} \left( c_{i,i} \left( \eta_{0,n-i+1}^2\|\Delta\mathbf{X}_{n-i+1}^{(k+1)}\|_{\mathrm{F}}^2 + \eta_{1,n-i+1}^2\|\Delta\mathbf{X}_{n-i+1}^{(k)}\|_{\mathrm{F}}^2 \right) \right)$$

$$+ \sum_{i=1}^{m} \frac{2i}{\beta_i} c_A^2 \sum_{l=n-i+2}^{n} \left( \sum_{p=n-l+2}^{i} \bar{\beta}_{n-l+1} c_{p,i} d_A^2 + c_{n-l+1,i}\eta_{1,l} \right)^2 \|\Delta\mathbf{X}_l^{(k)}\|_{\mathrm{F}}^2$$

$$+ \sum_{i=1}^{m} \frac{2i}{\beta_i} c_A^2 \sum_{l=n-i+2}^{n} \left( \sum_{p=n-l+2}^{i} \bar{\beta}_{n-l+1} c_{p,i} d_A^2 + c_{n-l+1,i}\eta_{0,l} \right)^2 \|\Delta\mathbf{X}_l^{(k+1)}\|_{\mathrm{F}}^2$$

$$\leq \sum_{l=n-m+1}^{n} \frac{2(n+1-l)}{\beta_{n+1-l}} c_{n-l+1,n-l+1} \left( \eta_{0,l}^2\|\Delta\mathbf{X}_l^{(k+1)}\|_{\mathrm{F}}^2 + \eta_{1,l}^2\|\Delta\mathbf{X}_l^{(k)}\|_{\mathrm{F}}^2 \right) \qquad (25)$$

$$+ \sum_{l=n-m+2}^{n} \sum_{i=n-l+2}^{m} \frac{2i}{\beta_i} c_A^2 \left( \sum_{p=n-l+2}^{i} \bar{\beta}_{n-l+1} c_{p,i} d_A^2 + c_{n-l+1,i}\eta_{1,l} \right)^2 \|\Delta\mathbf{X}_l^{(k)}\|_{\mathrm{F}}^2$$

$$+ \sum_{l=n-m+2}^{n} \sum_{i=n-l+2}^{m} \frac{2i}{\beta_i} c_A^2 \left( \sum_{p=n-l+2}^{i} \bar{\beta}_{n-l+1} c_{p,i} d_A^2 + c_{n-l+1,i}\eta_{0,l} \right)^2 \|\Delta\mathbf{X}_l^{(k+1)}\|_{\mathrm{F}}^2$$

Therefore, Inequality (25) can be rewritten as:

$$\sum_{i=1}^{m} \frac{1}{\beta_i}\|\Delta\boldsymbol{\Lambda}_i^{(k+1)}\|_{\mathrm{F}}^2 \leq \sum_{l=n-m+1}^{n} \theta_{0,l}\|\Delta\mathbf{X}_l^{(k)}\|_{\mathrm{F}}^2 + \theta_{1,l}\|\Delta\mathbf{X}_l^{(k+1)}\|_{\mathrm{F}}^2, \qquad (26)$$

where $\theta_{0,l}$ and $\theta_{1,l}$ are defined in (7). Finally, Inequality (21) follows from multiplying Inequality (26) by $r > 1$ and combining it with Lemma 4. $\qquad\square$

**Proof of Theorem 1**

*Proof.* According to (8), there exists $k_0$ such that $\mathcal{R}^{(k+1)} \leq \mathcal{R}^{(k_0)} \leq \mathcal{R}^{(0)}$ for all $k \geq 0$. Hence the following inequality holds:

$$
\sum_{i=1}^{n} f_i(\mathbf{X}_i^{(k+1)}) + \sum_{i=1}^{m} \frac{\beta_i}{2} \left\| \sum_{l=1}^{n-i+1} \mathcal{A}_{i,l}(\mathbf{X}_l^{(k+1)}) - \mathbf{b}_i + \frac{\mathbf{\Lambda}_i^{(k+1)}}{\beta_i} \right\|_{\mathrm{F}}^2
$$
$$
+ (r\theta_0 + a) \sum_{i=1}^{n} \|\Delta\mathbf{X}_i^{(k+1)}\|_{\mathrm{F}}^2 + a(\|\Delta\overline{\mathbf{\Lambda}}^{(k+1)}\|_{\mathrm{F}}^2) \leq \mathcal{R}^{(k_0)}. \tag{27}
$$

According to (24), we have $\mathcal{A}_{i,n-i+1}^{\mathrm{T}} \mathbf{\Lambda}_i^{(k+1)} = \mathbf{G}_i^{(k+1)}$ for $i = 1, \cdots, m$. If Assumption A(2) holds, by the update of $\mathbf{\Lambda}_i^{(k+1)}$, we have $\|\mathbf{\Lambda}_i^{(k+1)}\|_{\mathrm{F}} \leq c_A \|\mathbf{G}_i^{(k+1)}\|_{\mathrm{F}}$. According to (23), we have

$$
\|\mathbf{\Lambda}_i^{(k+1)}\|_{\mathrm{F}} \leq c_A \|\nabla f_{n-i+1}(\mathbf{X}_{n-i+1}^{(k+1)})\|_{\mathrm{F}} + c_A d_A \sum_{j=1}^{i-1} \|\mathbf{\Lambda}_j^{(k+1)}\|_{\mathrm{F}}
$$
$$
+ c_A \sum_{l=n-i+2}^{n} (\bar{\beta}_{n-l+1} d_A^2) \|\Delta\mathbf{X}_l^{(k+1)}\|_{\mathrm{F}} + c_A q_{n-i+1} \|\Delta\mathbf{X}_i^{(k+1)}\|_{\mathrm{F}}.
$$

It follows from Lemma 5 that for $i = 1, \cdots, m$, we have

$$
\|\mathbf{\Lambda}_i^{(k+1)}\|_{\mathrm{F}} \leq \sum_{p=1}^{i} c_{p,i} c_A \left( \|\nabla f_{n-p+1}(\mathbf{X}_{n-p+1}^{(k+1)})\|_{\mathrm{F}} + \sum_{l=n-p+2}^{n} \left( (\bar{\beta}_{n-l+1} d_A^2)\|\Delta\mathbf{X}_l^{(k+1)}\|_{\mathrm{F}} \right) + q_{n-i+1}\|\Delta\mathbf{X}_{n-p+1}^{(k+1)}\|_{\mathrm{F}} \right)
$$
$$
\leq \sum_{l=n-i+1}^{n} c_A c_{n-l+1,i}(\|\nabla f_l(\mathbf{X}_l^{(k+1)})\|_{\mathrm{F}} + q_{n-i+1}\|\Delta\mathbf{X}_{n-p+1}^{(k+1)}\|_{\mathrm{F}}) + c_A \sum_{l=n-i+2}^{n} \sum_{p=n-l+2}^{i} \left( c_{p,i} \bar{\beta}_{n-l+1}(d_A^2) \|\Delta\mathbf{X}_l^{(k+1)}\|_{\mathrm{F}} \right)
$$
$$
= \sum_{l=n-i+1}^{n} c_A c_{n-l+1,i}(\|\nabla f_l(\mathbf{X}_l^{(k+1)})\|_{\mathrm{F}} + q_{n-i+1}\|\Delta\mathbf{X}_{n-i+1}^{(k+1)}\|_{\mathrm{F}}) + c_A \sum_{l=n-i+2}^{n} \left( \sum_{p=n-l+2}^{i} c_{p,i} \bar{\beta}_{n-l+1}(d_A^2) + q_l \right) \|\Delta\mathbf{X}_l^{(k+1)}\|_{\mathrm{F}}
$$

Consequently, we have that for $i = 1, \cdots, m$:

$$
\frac{1}{\beta_i} \|\mathbf{\Lambda}_i^{(k+1)}\|_{\mathrm{F}}^2 \leq \frac{2i}{\beta_i} \sum_{l=n-i+1}^{n} \left( (c_{n-l+1,i} c_A)^2 \|\nabla f_{n-i+1}(\mathbf{X}_{n-i+1}^{(k+1)})\|_{\mathrm{F}}^2 \right) + \frac{2i}{\beta_i} \left( (c_{i,i} c_A \eta_{1,n-i+1})^2 \|\Delta\mathbf{X}_{n-i+1}^{(k+1)}\|_{\mathrm{F}}^2 \right)
$$
$$
+ \frac{2i}{\beta_i} c_A^2 \sum_{l=n-i+2}^{n} \left( \sum_{p=n-l+2}^{i} \bar{\beta}_{n-l+1} d_A^2 c_{p,i} + c_{n-l+1,i} \eta_{0,l} \right)^2 \|\Delta\mathbf{X}_l^{(k+1)}\|_{\mathrm{F}}^2.
$$

Summing both sides of the inequality over $i$ we have

$$\sum_{i=1}^{m} \frac{1}{\beta_i} \|\mathbf{\Lambda}_i^{(k+1)}\|_{\mathrm{F}}^2 \leq \sum_{i=1}^{m} \frac{2i}{\beta_i} \sum_{l=n-i+1}^{n} \left( c_{n-i+1,i}^2 \left( c_A^2 \|\nabla f_{n-i+1}(\mathbf{X}_{n-i+1}^{(k+1)})\|_{\mathrm{F}}^2 \right) \right) + \sum_{i=1}^{m} c_A^2 \frac{2i}{\beta_i} \left( c_{i,i}^2 \left( \eta_{0,n-i+1}^2 \|\Delta\mathbf{X}_{n-i+1}^{(k+1)}\|_{\mathrm{F}}^2 \right) \right)$$

$$+ \sum_{i=1}^{m} \frac{2i}{\beta_i} \sum_{l=n-i+2}^{n} \left( \sum_{p=n-l+2}^{i} \bar{\beta}_{n-l+1} c_A d_A c_{p,i} + c_{n-l+1,i}\eta_{0,l} \right)^2 \|\Delta\mathbf{X}_l^{(k+1)}\|_{\mathrm{F}}^2$$

$$= \sum_{l=n-m+1}^{n} \sum_{i=n-l+1}^{m} \frac{2(n+1-i)}{\beta_{n+1-i}} c_{n-l+1,i}^2 c_A^2 \eta_{0,l}^2 \|\nabla f_l(\mathbf{X}_l^{(k+1)})\|_{\mathrm{F}}^2 + \frac{2(n+1-l)}{\beta_{n+1-l}} c_{n-l+1,n-l+1}^2 \left( \eta_{0,l}^2 \|\Delta\mathbf{X}_l^{(k+1)}\|_{\mathrm{F}}^2 \right)$$

$$+ \sum_{l=n-m+2}^{n} \sum_{i=n-l+2}^{m} c_A^2 \frac{2i}{\beta_i} \left( \sum_{p=n-l+2}^{i} \bar{\beta}_{n-l+1} d_A^2 c_{p,i} + c_{n-l+1,i}\eta_{0,l} \right)^2 \|\Delta\mathbf{X}_l^{(k+1)}\|_{\mathrm{F}}^2.$$

Hence, it follows that

$$-\sum_{i=1}^{m} \frac{1}{2\beta_i} \|\mathbf{\Lambda}_i^{(k+1)}\|_{\mathrm{F}}^2 \geq -\left( \sum_{j=n-m+1}^{n} \left( \theta_{2,j}\|\Delta\mathbf{X}_j^{(k+1)}\|_{\mathrm{F}}^2 + \theta_{3,j}\|\nabla f_j(\mathbf{X}_j^{(k)})\|_{\mathrm{F}}^2 \right) \right). \tag{28}$$

where $\theta_{2,l} := \sum_{i=n-l+2}^{m} \frac{i}{\beta_i} c_A^2 \left( \sum_{p=n-l+2}^{i} \bar{\beta}_{n-l+1} c_{p,i} d_A^2 + c_{n-l+1,i}\eta_{1,l} \right)^2 + \frac{(n+1-l)}{\beta_{n+1-l}} c_{n-l+1,n-l+1}\eta_{0,l}^2, \theta_{3,l}$ is defined in A(4). Using Inequalities (27) and (28), we obtain

$$(1-\nu)\sum_{i=1}^{n} f_i(\mathbf{X}_i^{(k+1)}) + \sum_{i=1}^{m} \frac{\beta_i}{2} \left\| \sum_{l=1}^{n-i+1} \mathcal{A}_{i,l}(\mathbf{X}_l^{(k+1)}) - \mathbf{b}_i + \frac{\mathbf{\Lambda}_i^{(k+1)}}{\beta_i} \right\|_{\mathrm{F}}^2$$

$$+ a(\|\Delta\overline{\mathbf{X}}^{(k+1)}\|_{\mathrm{F}}^2 + \|\Delta\overline{\mathbf{\Lambda}}^{(k+1)}\|_{\mathrm{F}}^2) + \sum_{i=n-m+1}^{n} (r\theta_{1,i} - \theta_{2,i})\|\Delta\mathbf{X}_i^{(k+1)}\|_{\mathrm{F}}^2$$

$$\leq \mathcal{R}^{(0)} - \inf_{\mathbf{X}_i} \left\{ \nu f_i(\mathbf{X}_i) - \theta_{3,i}\|\nabla f_i(\mathbf{X}_i^{(k)})\|_{\mathrm{F}}^2 \right\}. \tag{29}$$

According to A(1), we set $\mathbf{W}_1 = \mathbf{X}_i^{(k)} - \delta\nabla f_i(\mathbf{X}_i^{(k)})$ and $\mathbf{W}_2 = \mathbf{X}_i^{(k)}$. It follows that $f(\mathbf{X}_i^{(k)} - \delta\nabla f(\mathbf{X}_i^{(k)})) \leq f(\mathbf{X}_i^{(k)}) - (\delta - \frac{L_{f,i}\delta^2}{2})\|\nabla f(\mathbf{X}_i^{(k)})\|_{\mathrm{F}}^2$. Since $f_i, i = n-m+1, \cdots, n$ is bounded from below, there exist $M$ and $\nu \in (0,1)$ such that

$$-M < \inf_{\mathbf{X}_i} \{ \nu f_i(\mathbf{X}_i) - (\delta - \frac{L_{f_i}\delta^2}{2})\|\nabla f_i(\mathbf{X}_i)\|_{\mathrm{F}}^2 \}, \quad i = n-m+1, \cdots, n. \tag{30}$$

Let $\delta = \frac{1}{L_{f_i}}$ and according to A(4), we have $\theta_{3,i} < \frac{1}{2L_{f_i}} = \delta - \frac{L_{f_i}\delta^2}{2}$. Since $r > 1$, according to the definition of $\theta_{1,i}, \theta_{2,i}$, it holds that $r\theta_{1,i} - \theta_{2,i} > 0$ for $i = n-m+1, \cdots, n$. It follows from (29) that

$$\sum_{i=1}^{n-m} f_i(\mathbf{X}_i^{(k+1)}) + \sum_{i=n-m+1}^{n} (1-\nu)f_i(\mathbf{X}_i^{(k+1)}) + \sum_{i=1}^{m} \frac{\beta_i}{2} \left\| \sum_{j=1}^{n-i+1} \mathcal{A}_{i,j}(\mathbf{X}_i^{(k+1)}) - \mathbf{b}_i + \frac{\mathbf{\Lambda}_i^{(k+1)}}{\beta_i} \right\|_{\mathrm{F}}^2$$

$$+ a(\|\Delta\overline{\mathbf{X}}^{(k+1)}\|_{\mathrm{F}}^2 + \|\Delta\overline{\mathbf{\Lambda}}^{(k+1)}\|_{\mathrm{F}}^2) < \mathcal{R}^{(0)} + mM. \tag{31}$$

Since $f_i, i = 1, \cdots, n-m$ are coercive, the sequences $\{\mathbf{X}_i^{(k)}\}_{k\geq0}, i = 1, \cdots, n-m$ are bounded and hence $\{\sum_{i=1}^{n-m} \mathcal{A}_i\mathbf{X}_i^{(k)}\}_{k\geq0}$ is bounded. According to (31), $\Delta\mathbf{\Lambda}_i^{(k+1)}$ is bounded. It follows from A(3) and the update of $\overline{\mathbf{\Lambda}}$ in Algorithm 1 that $\sum_{i=1}^{n-m} \mathcal{A}_i\mathbf{X}_i^{(k+1)} = [\frac{1}{\beta_1}\Delta\mathbf{\Lambda}_1^{(k+1)}; \cdots; \frac{1}{\beta_m}\Delta\mathbf{\Lambda}_{k+1,m}] - \sum_{i=n-m+1}^{n} \mathcal{A}_i\mathbf{X}_i^{(k+1)} + \mathbf{b}$. Since $\{\mathcal{A}\overline{\mathbf{X}}^{(k)}\}_{k\geq0}, i = n-m+1, \cdots, n$ are bounded and $\mathcal{A}_{n-i+1,i}^{\mathrm{T}}\mathcal{A}_{n-i+1,i}, i = n-m+1, \cdots, n$ has full rank, it follows

that $\{\mathbf{X}_i^{(k)}\}_{k\geq 0}, i = n-m+1,\cdots,n$ are bounded. From the fact that $\sum_{i=1}^{m} \frac{\beta_i}{2} \left\| \sum_{j=1}^{n-i+1} \mathcal{A}_{i,j}(\mathbf{X}_i^{(k+1)}) - \mathbf{b}_i + \frac{\mathbf{\Lambda}_i^{(k+1)}}{\beta_i} \right\|_{\mathrm{F}}^2$ is bounded, it follows that $\{\overline{\mathbf{\Lambda}}^{(k)}\}_{k\geq 0}$ is bounded. Consequently, $\{\overline{\mathbf{X}}^{(k)}\}_{k\geq 1}, \{\overline{\mathbf{\Lambda}}^{(k)}\}_{k\geq 1}$ are bounded. $\qquad\square$

Lemma 7 will be used in Lemma 2 to present the properties of the limiting point of $(\Delta\overline{\mathbf{X}}^{(k)}, \Delta\overline{\mathbf{\Lambda}}^{(k)})$.

**Lemma 7.** *Suppose Assumption 1 holds. It follows that*

$$\lim_{k\to\infty} \|\Delta\overline{\mathbf{X}}^{(k)}\|_{\mathrm{F}} = 0, \quad \lim_{k\to\infty} \|\Delta\overline{\mathbf{\Lambda}}^{(k)}\|_{\mathrm{F}} = 0. \tag{32}$$

*Proof.* It follows from (8) that

$$\sum_{k=1}^{K} (\|\overline{\mathbf{X}}^{(k)}\|_{\mathrm{F}}^2 + \|\overline{\mathbf{\Lambda}}^{(k)}\|_{\mathrm{F}}^2) \leq \frac{1}{a}(\mathcal{R}^{(1)} - \mathcal{R}^{(K)}) \leq \frac{1}{a}(\mathcal{R}^{(1)} - \inf_k \mathcal{R}^{(k)}).$$

This indicates that $\lim_{k\to\infty} \|\Delta\overline{\mathbf{X}}^{(k)}\|_{\mathrm{F}}^2 = 0, \lim_{k\to\infty} \|\Delta\overline{\mathbf{\Lambda}}^{(k)}\|_{\mathrm{F}}^2 = 0$. Hence the conclusion holds. $\qquad\square$

The bounded sequence in Theorem 1 guarantees the existence of $\{(\overline{\mathbf{X}}^{(k)}, \overline{\mathbf{\Lambda}}^{(k)})\}_{k\geq 0}$. Specifically, Lemma 2 presents the property of its limit point set.

**Proof of Lemma 2**

*Proof.* According to Theorem 1, $(\overline{\mathbf{X}}^{(k)}, \overline{\mathbf{\Lambda}}^{(k)})$ is a bounded sequence. Hence, the result $(i)$ follows from the Weierstrass theorem. The conclusions $(ii)$ and $(iii)$ follow from Lemmas 8 and 7. $\qquad\square$

We next show the following subgradient bound lemma that is used in Theorem 3 to prove the global convergence of Algorithm 1.

**Lemma 8** (Subgradient bound)**.** *Let* $\{(\overline{\mathbf{X}}^{(k)}, \overline{\mathbf{\Lambda}}^{(k)})\}_{k\geq 0}$ *be a sequence generated by Algorithm 1. It follows that* $\mathbf{d}^{(k+1)} = (\mathbf{d}_{\overline{\mathbf{X}}^{(k+1)}}, \mathbf{d}_{\overline{\mathbf{\Lambda}}^{(k+1)}}) \in \partial\mathcal{L}(\overline{\mathbf{X}}, \overline{\mathbf{\Lambda}})$ *where*

$$\mathbf{d}_{\mathbf{X}_i^{(k+1)}} := \sum_{j=1}^{\min\{n-i+1,m\}} \mathcal{A}_{j,i}^{\mathrm{T}} \Delta\mathbf{\Lambda}_j^{(k+1)} - (\mathcal{Q}_i^{(k)} \Delta\mathbf{X}_i^{(k+1)}) + \sum_{j=1}^{\min\{n-i+1,m\}} \beta_j \sum_{l=i+1}^{n-j+1} \mathcal{A}_{i,j}^{\mathrm{T}} \mathcal{A}_{i,l} \Delta\mathbf{X}_l^{(k+1)}, \ i = 1,\cdots,n,$$

$$\mathbf{d}_{\mathbf{\Lambda}_j^{(k+1)}} := \frac{1}{\beta_j} \Delta\mathbf{\Lambda}_i^{(k+1)}, \ j = 1,\cdots m. \tag{33}$$

*Furthermore, it holds that*

$$\|\mathbf{d}^{(k+1)}\|_{\mathrm{F}} \leq \pi(\|\Delta\overline{\mathbf{X}}^{(k+1)}\|_{\mathrm{F}} + \|\Delta\overline{\mathbf{\Lambda}}^{(k+1)}\|_{\mathrm{F}}),$$

*where* $\pi := \max_{i=1,\cdots,n,j=1,\cdots,m} \left\{ \beta_{\max} d_A^2 + q_j, m e_A + \frac{1}{\beta_i} \right\}$, *where* $e_A = \max_{i,j} \|\mathcal{A}_{i,j}\|, i = 1,\cdots,m, j = 1,\cdots,n-i+1$.

*Proof.* According to the optimality condition of $\mathbf{X}_i^{(k+1)}, i = 1,\cdots,n$, it follows that

$$-\sum_{j=1}^{\min\{n-i+1,m\}} \beta_j \mathcal{A}_{j,i}^{\mathrm{T}} \left( \sum_{l=1}^{i} \mathcal{A}_{j,l}(\mathbf{X}_l^{(k+1)}) + \sum_{l=i+1}^{n-j+1} \mathcal{A}_{j,l}(\mathbf{X}_l^{(k)}) - \mathbf{b}_j + \mathbf{\Lambda}_j^{(k)}/\beta_i \right) - \mathcal{Q}_i^{(k)}(\Delta\mathbf{X}_i^{(k+1)}) \in \partial f_i(\mathbf{X}_i^{(k+1)}).$$

It follows from the update of $\mathbf{\Lambda}_j^{(k+1)}$ that $\mathbf{d}_{\mathbf{X}_i^{(k+1)}} \in \partial_{\mathbf{X}_i}\mathcal{L}(\overline{\mathbf{X}}^{(k+1)}, \overline{\mathbf{\Lambda}}^{(k+1)})$. According to the update of $\mathbf{\Lambda}_i^{(k+1)}$, we have $\mathbf{d}_{\mathbf{\Lambda}_i^{(k+1)}} = \frac{1}{\beta_i}\Delta\mathbf{\Lambda}_i^{(k+1)} \in \partial_{\mathbf{\Lambda}_i}\mathcal{L}(\overline{\mathbf{X}}^{(k+1)}, \overline{\mathbf{\Lambda}}^{(k+1)})$. Since $\mathbf{d}^{(k+1)} = (\mathbf{d}_{\overline{\mathbf{X}}^{(k+1)}}, \mathbf{d}_{\overline{\mathbf{\Lambda}}^{(k+1)}})$, the proof is completed. $\qquad\square$

Using Theorem 1 and we have the following convergence result of $\mathcal{R}^{(k)}$.

**Lemma 9.** *Suppose Assumption 1 holds. The sequence $\{\mathcal{R}^{(k)}\}_{k \geq 1}$ is bounded from below and converges.*

*Proof.* It follows from Theorem 1 that $\{(\overline{\mathbf{X}}^{(k)}, \overline{\mathbf{\Lambda}}^{(k)})\}_{k \geq 1}$ is bounded. Hence, we have $\left\langle \overline{\mathbf{\Lambda}}^{(k)}, \mathcal{A}(\overline{\mathbf{X}}^{(k)}) - \mathbf{b} \right\rangle$ and $\| \sum_{i=1}^{n} \mathcal{A}_i(\mathbf{X}_i^{(k)}) - \mathbf{b} \|_F^2$ are bounded. Since $f_i, i = 1, \cdots, n - m$ are coercive and $\{\overline{\mathbf{X}}^{(k)}\}_{k \geq 0}$ is bounded, then $\{f_i(\mathbf{X}_i^{(k)})\}$ is bounded. Hence $\mathcal{R}^{(k)}$ is bounded from below due to $f_i, i = n - m + 1, \cdots, n$ is bounded from below. According to (8), $\{\mathcal{R}^{(k)}\}_{k \geq 1}$ is monotonically decreasing which indicates that $\{\mathcal{R}^{(k)}\}_{k \geq 1}$ is bounded from below and converges. $\qquad \square$

### A.4. Proof of Lemma 2

It follows from the convexity of $f_i, i = 1, \cdots, n$ that $\mu_{f_i} \geq 0$. If $f_i$ is strongly convex, $\mu_{f_i} > 0$. It follows from the optimality condition of $\mathbf{X}_i^{(k+1)}, i = 1, \cdots, n$ that

$$
- \sum_{j=1}^{\min\{m, n-i+1\}} \mathcal{A}_{j,i}^{\mathrm{T}} \left( \beta_j \left( \sum_{l=1}^{i} \mathcal{A}_{j,l} \mathbf{X}_l^{(k+1)} + \sum_{l=i+1}^{n-j+1} \mathcal{A}_{j,l} \mathbf{X}_l^{(k)} + \mathbf{b}_j \right) + \mathbf{\Lambda}_j^{(k)} \right) - \mathcal{Q}_i(\mathbf{X}_i^{(k+1)} - \mathbf{X}_i^{(k)}) \in \partial f_i(\mathbf{X}_i^{(k+1)}).
$$
(34)

By combining the update of $\mathbf{\Lambda}_i^{(k+1)}$, the following inequality holds.

$$
- \sum_{j=1}^{\min\{m, n-i+1\}} \mathcal{A}_{j,i}^{\mathrm{T}} \left( \beta_j \sum_{l=i+1}^{n-j+1} \mathcal{A}_{j,l} (\mathbf{X}_l^{(k)} - \mathbf{X}_l^{(k+1)}) + \mathbf{\Lambda}_j^{(k+1)} \right) - \mathcal{Q}_i(\mathbf{X}_i^{(k+1)} - \mathbf{X}_i^{(k)}) \in \partial f_i(\mathbf{X}_i^{(k+1)}).
$$
(35)

In our analysis, for $w_i \in \mathbb{R}^{m \times n}, i = 1, \cdots, 4$, the following identity is used:

$$
(w_1 - w_2)^{\top} (w_3 - w_4) = \frac{1}{2} \left( \|w_1 - w_4\|_F^2 - \|w_1 - w_3\|_F^2 \right) + \frac{1}{2} \left( \|w_2 - w_3\|_F^2 - \|w_2 - w_4\|_F^2 \right).
$$
(36)

We first prove the following lemma to present an upper bound for $\|\mathbf{X}_1^{(k)} - \mathbf{X}_{*,1}\|_F^2$.

**Lemma 10** (Bound for $\|\mathbf{X}_1^{(k+1)} - \mathbf{X}_{*,1}\|_F^2$). *Suppose $\mathcal{A}_1$ is injective, we have*

$$
\|\mathbf{X}_1^{(k+1)} - \mathbf{X}_{*,1}\|_F^2 \leq 2c_{A,1} \sum_{j=1}^{m} u_{1,j,k} + 2c_{A,1} \sum_{j=1}^{m} \| \sum_{j=2}^{n-i+1} \mathcal{A}_{i,j} (\mathbf{X}_1^{(k)} - \mathbf{X}_{*,1}) \|_F^2,
$$
(37)

*where $c_{A,1}$ denotes the inverse of the smallest positive eigenvalue of $\mathcal{A}_1^{\mathrm{T}} \mathcal{A}_1$.*

*Proof.* Since $\mathcal{A}_1$ is injective, we have

$$
\|\mathbf{X}_1^{(k)} - \mathbf{X}_{*,1}\|_F^2 \leq c_{A,1} \|\mathcal{A}_1(\mathbf{X}_1^{(k+1)} - \mathbf{X}_{*,1})\|_F^2 = c_{A,1} \sum_{i=1}^{m} \|\mathcal{A}_{1,i}(\mathbf{X}_1^{(k+1)} - \mathbf{X}_{*,1})\|_F^2
$$

$$
= c_{A,1} \sum_{i=1}^{m} \|(\mathcal{A}_{i,1} \mathbf{X}_1^{(k+1)} + \sum_{j=2}^{n-i+1} \mathcal{A}_{i,j} \mathbf{X}_1^{(k)} - \mathbf{b}_i) - \sum_{j=2}^{n-i+1} \mathcal{A}_{i,j} (\mathbf{X}_1^{(k)} - \mathbf{X}_{*,1}) \|_F^2
$$

$$
\leq 2c_{A,1} \sum_{i=1}^{m} \|\mathcal{A}_{i,1} \mathbf{X}_1^{(k+1)} + \sum_{j=2}^{n-i+1} \mathcal{A}_{i,j} \mathbf{X}_1^{(k)} - \mathbf{b}_i \|_F^2 + 2c_{A,1} \| \sum_{j=2}^{n-i+1} \mathcal{A}_{i,j} (\mathbf{X}_1^{(k)} - \mathbf{X}_{*,1}) \|_F^2
$$

$$
= 2c_{A,1} \sum_{j=1}^{m} u_{1,j,k} + 2c_{A,1} \sum_{i=1}^{m} \| \sum_{j=2}^{n-i+1} \mathcal{A}_{i,j} (\mathbf{X}_1^{(k)} - \mathbf{X}_{*,1}) \|_F^2.
$$

The proof is completed. $\qquad \square$

**Proof of Lemma 2**

*Proof.* According to the optimality condition of $\mathbf{X}_i^{(k+1)}$, the update of $\mathbf{\Lambda}_i^{(k+1)}$, $i = 1, \cdots, n$ and (34), the following inequality holds:

$$
(\mathbf{X}_i^{(k+1)} - \mathbf{X}_{\mathrm{op},i})^{\mathrm{T}} \left[ \sum_{j=1}^{\min\{n-i+1,m\}} \mathcal{A}_{j,i}^{\mathrm{T}} \left( \mathbf{\Lambda}_j^{(k+1)} - \mathbf{\Lambda}_{*,j} - \beta_j \sum_{l=i+1}^{n-j+1} \mathcal{A}_{j,l}(\mathbf{X}_l^{(k)} - \mathbf{X}_l^{(k+1)}) \right) - \mathcal{Q}_i(\mathbf{X}_i^{(k+1)} - \mathbf{X}_i^{(k)}) \right]
$$
$$
\geq \kappa_{f_i} \|\mathbf{X}_i^{(k+1)} - \mathbf{X}_{\mathrm{op},i}\|_{\mathrm{F}}^2.
\tag{38}
$$

From the update of $\mathbf{\Lambda}_j^{(k+1)}$, $j = 1, \cdots, m$, we can obtain that:

$$
\sum_{i=1}^n \sum_{j=1}^{\min\{n-i+1,m\}} \mathcal{A}_{i,j}(\mathbf{X}_j^{(k+1)} - \mathbf{X}_{\mathrm{op},j}) = \sum_{i=1}^m \frac{1}{\beta_i}(\mathbf{\Lambda}_i^{(k)} - \mathbf{\Lambda}_i^{(k+1)}).
\tag{39}
$$

Summing over $i = 1, \cdots, n$ and it follows that

$$
\sum_{i=1}^m \frac{1}{\beta_i}(\mathbf{\Lambda}_i^{(k)} - \mathbf{\Lambda}_i^{(k+1)})^{\mathrm{T}}(\mathbf{\Lambda}_i^{(k+1)} - \mathbf{\Lambda}_{*,i}) + \sum_{i=1}^{n-1}(\mathbf{X}_{\mathrm{op},i} - \mathbf{X}_i^{(k+1)})^{\mathrm{T}} \sum_{j=1}^{\min\{n-i+1,m\}} \beta_j \mathcal{A}_{j,i}^{\mathrm{T}} \left[ \sum_{l=i+1}^{n-j+1} \mathcal{A}_{j,l}(\mathbf{X}_l^{(k)} - \mathbf{X}_l^{(k+1)}) \right]
$$
$$
+ \sum_{i=1}^n (\mathbf{X}_i^{(k+1)} - \mathbf{X}_{\mathrm{op},i})^{\mathrm{T}} \mathcal{Q}_i((\mathbf{X}_i^{(k)}) - (\mathbf{X}_i^{(k+1)})) \geq \sum_{i=1}^n \kappa_{f_i} \|\mathbf{X}_i^{(k+1)} - \mathbf{X}_{\mathrm{op},i}\|_{\mathrm{F}}^2.
\tag{40}
$$

Defining $s_{i,j,k} := \| \sum_{l=1}^i \mathcal{A}_{j,l}\mathbf{X}_{\mathrm{op},l} + \sum_{l=i+1}^{n-j+1} \mathcal{A}_{j,l}\mathbf{X}_l^{(k)} - \mathbf{b}_j \|_{\mathrm{F}}^2$, we have that

$$
\sum_{i=1}^n (\mathbf{X}_{\mathrm{op},i} - \mathbf{X}_i^{(k+1)})^{\mathrm{T}} \sum_{j=1}^{\min\{n-i+1,m\}} \beta_j \mathcal{A}_{j,i}^{\mathrm{T}} \left[ \sum_{l=i+1}^{n-j+1} \mathcal{A}_{j,l}(\mathbf{X}_l^{(k)} - \mathbf{X}_l^{(k+1)}) \right]
$$
$$
= \sum_{j=1}^m \beta_j \sum_{i=1}^{n-j+1} \left[ \left( \sum_{l=1}^i \mathcal{A}_{j,l}\mathbf{X}_{\mathrm{op},l} - \mathbf{b}_j \right) - \left( \sum_{l=1}^{i-1} \mathcal{A}_{j,l}\mathbf{X}_{\mathrm{op},l} + \mathcal{A}_{j,i}\mathbf{X}_i^{(k+1)} - \mathbf{b}_j \right) \right]^{\mathrm{T}} \left[ -\sum_{l=i+1}^{n-j+1} \mathcal{A}_{j,l}\mathbf{X}_l^{(k+1)} + \sum_{l=i+1}^{n-j+1} \mathcal{A}_{j,l}\mathbf{X}_l^{(k)} \right]
$$
$$
= \sum_{j=1}^m \beta_j \sum_{i=1}^{n-j+1} \left[ \frac{1}{2}(s_{i,j,k} - s_{i,j,k+1}) + \frac{1}{2}(s_{i-1,j,k+1} - u_{i,j,k}) \right]
$$
$$
\leq \frac{1}{2} \sum_{j=1}^m \beta_j \sum_{i=1}^{n-j+1}(s_{i,j,k} - s_{i,j,k+1}) + \frac{1}{2} \sum_{j=1}^m \sum_{i=1}^{n-j+1} s_{i-1,j,k+1} - \frac{1}{2} \sum_{j=1}^m \beta_j u_{1,j,k}
$$
$$
= \frac{1}{2} \sum_{j=1}^m \beta_j \sum_{i=1}^{n-j+1}(s_{i,j,k} - s_{i,j,k+1}) + \frac{1}{2} \sum_{i=1}^m \frac{1}{\beta_i}\|\mathbf{\Lambda}_i^{(k+1)} - \mathbf{\Lambda}_i^{(k)}\|_{\mathrm{F}}^2 + \frac{1}{2} \sum_{j=1}^m \sum_{i=2}^{n-j+1} \beta_j s_{i-1,j,k+1} - \frac{1}{2} \sum_{j=1}^m \beta_j u_{1,j,k}, \tag{41}
$$

where the second equality follows from (36) and the last follows from the update of $\mathbf{\Lambda}_i$.

$$
\sum_{i=1}^n (\mathbf{X}_i^{(k+1)} - \mathbf{X}_{\mathrm{op},i})^{\mathrm{T}} \mathcal{Q}_i(\mathbf{X}_i^{(k)} - \mathbf{X}_i^{(k+1)}) = -\sum_{i=1}^n \frac{1}{2}\|\mathbf{X}_i^{(k)} - \mathbf{X}_i^{(k+1)}\|_{\mathcal{Q}_i}^2 + \frac{1}{2}\|\mathbf{X}_i^{(k)} - \mathbf{X}_{\mathrm{op},i}\|_{\mathcal{Q}_i}^2 - \frac{1}{2}\|\mathbf{X}_i^{(k+1)} - \mathbf{X}_{\mathrm{op},i}\|_{\mathcal{Q}_i}^2.
\tag{42}
$$

It follows from (40) and (41) that

$$
\sum_{j=1}^m \sum_{i=1}^{n-j+1} \frac{\beta_j}{2}(s_{i,j,k} - s_{i,j,k+1}) + \sum_{i=1}^m \frac{1}{\beta_i}(\mathbf{\Lambda}_i^{(k)} - \mathbf{\Lambda}_i^{(k+1)})^{\mathrm{T}}(\mathbf{\Lambda}_i^{(k+1)} - \mathbf{\Lambda}_{*,i}) + \sum_{i=1}^m \frac{1}{2\beta_i}\|\mathbf{\Lambda}_i^{(k+1)} - \mathbf{\Lambda}_i^{(k)}\|_{\mathrm{F}}^2
$$
$$
+ \sum_{j=1}^m \frac{\beta_j}{2} \sum_{i=2}^{n-j+1} s_{i-1,j,k+1} \geq \sum_{i=1}^n \kappa_{f_i} \|\mathbf{X}_i^{(k+1)} - \mathbf{X}_{\mathrm{op},i}\|_{\mathrm{F}}^2 + \sum_{j=1}^m \frac{\beta_j}{2} u_{1,j,k}.
$$

Using (9) and we obtain

$$s_{i-1,j,k+1} = \left\| \sum_{l=i}^{n-j+1} \mathcal{A}_{j,l}(\mathbf{X}_j^{(k+1)} - \mathbf{X}_{\mathrm{op},j}) \right\|_{\mathrm{F}}^2 \leq (n-i-j+2) \sum_{l=i}^{n-j+1} e_A^2 \|\mathbf{X}_l^{(k+1)} - \mathbf{X}_{\mathrm{op},l}\|_{\mathrm{F}}^2,$$

Therefore, it follows that

$$\sum_{j=1}^m \frac{\beta_j}{2} \sum_{i=1}^{n-j+1} s_{i-1,j,k+1} \leq \beta_{\max} \sum_{j=1}^m \sum_{i=2}^{n-j+1} \left( (n-i-j+2) \sum_{l=i}^{n-j+1} e_A^2 \|\mathbf{X}_l^{(k+1)} - \mathbf{X}_{\mathrm{op},l}\|_{\mathrm{F}}^2 \right) \tag{43}$$

$$= \beta_{\max} \sum_{j=1}^m \sum_{i=2}^{n-j+1} \left( e_A^2 \|\mathbf{X}_i^{(k+1)} - \mathbf{X}_{\mathrm{op},i}\|_{\mathrm{F}}^2 \left[ (n-j+2)(i-1) - \frac{i(i+1)}{2} + 1 \right] \right),$$

where $\beta_{\max} = \max_{i=1,\cdots,m} \beta_i$. Switch the summation order of $i$, $j$ and we obtain

$$\sum_{j=1}^{\min\{m,n-i+1\}} \left[ (n-j+2)(i-1) - \frac{i(i+1)}{2} + 1 \right] = -\frac{i(i+1)}{2} \cdot \min(m, n-i+1) + \min(m, n-i+1)$$

$$+ (i-1) \left[ \min(m, n-i+1)(n+1) - \frac{\min(m,n-i+1)(\min(m,n-i+1)-1)}{2} \right].$$

Consequently, it holds that $\sum_{j=1}^m \frac{\beta_j}{2} \sum_{i=1}^{n-j+1} s_{i-1,j,k+1} \leq \sum_{i=1}^n a_i \|\mathbf{X}_i^{(k)} - \mathbf{X}_{\mathrm{op},i}\|_{\mathrm{F}}^2$. By combining (41), (43) and using the identity

$$\sum_{i=1}^m \frac{1}{\beta_i} (\mathbf{\Lambda}_i^{(k)} - \mathbf{\Lambda}_i^{(k+1)})^{\mathrm{T}} (\mathbf{\Lambda}_i^{(k+1)} - \mathbf{\Lambda}_{*,i}) + \sum_{i=1}^m \frac{1}{2\beta_i} \|\mathbf{\Lambda}_i^{(k+1)} - \mathbf{\Lambda}_i^{(k)}\|_{\mathrm{F}}^2 = \sum_{i=1}^m \frac{1}{2\beta_i} (\|\mathbf{\Lambda}_{*,i} - \mathbf{\Lambda}_i^{(k)}\|_{\mathrm{F}}^2 - \|\mathbf{\Lambda}_{*,i} - \mathbf{\Lambda}_i^{(k+1)}\|_{\mathrm{F}}^2),$$

we have

$$r^{(k)} - r^{(k+1)} \geq \sum_{i=2}^n \left[ \left( \kappa_{f_i} - \frac{\beta_{\max} a_i}{2} \right) \|\mathbf{X}_i^{(k+1)} - \mathbf{X}_{\mathrm{op},i}\|_{\mathrm{F}}^2 \right] + \sum_{j=1}^m \frac{\beta_j}{2} u_{1,j,k} + \frac{1}{2} \sum_{i=1}^n \|\mathbf{X}_i^{(k+1)} - \mathbf{X}_i^{(k)}\|_{\mathcal{Q}_i}^2.$$

The proof is completed. □

### A.5. Proof of Theorem 4

*Proof.* According to the assumptions, $f_2, \ldots, f_n$ are strongly convex. Denote the right-hand side of inequality (10) by $\eta^{(k)}$. It follows from (10) that $\eta^{(k)} \geq 0$ and $\sum_{k=0}^\infty \eta^{(k)} < +\infty$, which further implies that $\eta^{(k)} \to 0$. Hence, for any $(\overline{\mathbf{X}}_{\mathrm{op}}, \overline{\mathbf{\Lambda}}_{\mathrm{op}}) \in \Omega_*$, we have

$$\mathbf{X}_i^{(k)} - \mathbf{X}_{\mathrm{op},i} \to 0 \quad \text{for } i = 2, \ldots, n, \quad \text{and} \quad \mathcal{A}_{i,1}\mathbf{X}_1^{(k+1)} + \sum_{j=2}^{n-i+1} \mathcal{A}_{i,j}\mathbf{X}_j^{(k)} - \mathbf{b}_i \to 0, i = 1, \cdots, m,$$

which also implies that $\mathcal{A}_1 \mathbf{X}_1^{(k)} - \mathcal{A}_1 \mathbf{X}_{\mathrm{op},1} \to 0$. In Scenario 2 and 4, since $f_1$ is strongly convex, $\mu_{f_1} > 0$ and (10) implies that $\mathbf{X}_1^{(k)} - \mathbf{X}_{\mathrm{op},1} \to 0$. In Scenario 1 and 3, it is assumed that $\mathcal{A}_1$ is of injective. It follows from $\mathcal{A}_1 \mathbf{X}_1^{(k)} - \mathcal{A}_1 \mathbf{X}_{\mathrm{op},1} \to 0$ that $\mathbf{X}_1^{(k)} - \mathbf{X}_{\mathrm{op},1} \to 0$. Furthermore, when (12) holds, it follows from Lemma 2 that

$$\sum_{j=1}^m \frac{1}{2\beta_i} \sum_{i=j}^{n-j+1} \left\| \sum_{l=i+1}^{n-j+1} \mathcal{A}_{j,l}(\mathbf{X}_j^{(k)} - \mathbf{X}_{\mathrm{op},j}) \right\|_{\mathrm{F}}^2 + \sum_{j=1}^m \frac{1}{\beta_j} \|\mathbf{\Lambda}_{\mathrm{op},i} - \mathbf{\Lambda}_i^{(k)}\|_{\mathrm{F}}^2 + \frac{1}{2} \sum_{i=1}^n \|\mathbf{X}_i^{(k)} - \mathbf{X}_{\mathrm{op},i}\|_{\mathcal{Q}_i}^2$$

is nonincreasing and upper bounded. Hence we have that $\|\mathbf{\Lambda}_{\mathrm{op},i} - \mathbf{\Lambda}_i^{(k)}\|, i = 1, \cdots, m$ converges and $\{\mathbf{\Lambda}_i^{(k)}\}$ is bounded. Therefore, $\{\mathbf{\Lambda}_i^{(k)}\}$ has a converging subsequence $\{\mathbf{\Lambda}_{k_j,i}\}$. Let $\mathbf{\Lambda}_{*,i} = \lim_{j\to\infty}\{\mathbf{\Lambda}_{k_j,i}\}$. By passing the limit in (35), it holds that $\sum_{j=1}^{n-i+1}\mathcal{A}_{i,j}^{\mathrm{T}}\mathbf{\Lambda}_{*,j} = \nabla f_i(\mathbf{X}_{*,i})$ for $i = 1,\dots,n$. Hence we have $(\overline{\mathbf{X}}_*, \mathbf{\Lambda}_{*,1}, \cdots, \mathbf{\Lambda}_{*,m}) \in \Omega^*$ and we can set $\overline{\mathbf{\Lambda}}_* = (\mathbf{\Lambda}_{*,1}, \cdots, \mathbf{\Lambda}_{*,m})$. Since $\|\mathbf{\Lambda}_{*,i} - \mathbf{\Lambda}_i^{(k)}\|$ converges and $\mathbf{\Lambda}_{k_j,i} \to \mathbf{\Lambda}_{*,i}$, we conclude that $\overline{\mathbf{\Lambda}}^{(k)} \to \overline{\mathbf{\Lambda}}_*$. Hence the proof is completed. $\qquad\square$

### A.6. Proof of Theorem 6

The following lemma is used to bound $\sum_{i=1}^m \frac{1}{\beta_i}\|\mathbf{\Lambda}_i^{(k+1)} - \mathbf{\Lambda}_{*,i}\|_{\mathrm{F}}^2$ in different scenarios.

**Lemma 11** (bound for $\mathbf{\Lambda}_i^{(k+1)}$)**.** *Suppose that Assumptions 2 and 3 hold. Let $(\overline{\mathbf{X}}_*, \overline{\mathbf{\Lambda}}_*) \in \Omega_*$ be the convergence point in Theorem 4. In Scenarios 1 and 2, where $f_i$ has Lipschitz continuous gradients and A(2) holds, we have*

$$
\begin{aligned}
\sum_{i=1}^m \frac{1}{\beta_i}\|\mathbf{\Lambda}_i^{(k+1)} - \mathbf{\Lambda}_{*,i}\|_{\mathrm{F}}^2 &\leq \sum_{i=n-m+1}^n B_i\|\mathbf{X}_i^{(k+1)} - \mathbf{X}_{*,i}\|_{\mathrm{F}}^2 + \sum_{i=n-m+1}^n C_i\|\mathbf{X}_i^{(k+1)} - \mathbf{X}_i^{(k)}\|_{\mathrm{F}}^2 \\
&+ \left(\sum_{j=1}^m \sum_{i=1}^{n-j+1} D_j\left(\left(\left\|\sum_{l=i+1}^{n-j+1}\mathcal{A}_{j,l}(\mathbf{X}_l^{(k)} - \mathbf{X}_{*,l})\right\|_{\mathrm{F}}^2 + \left\|\sum_{l=i+1}^{n-j+1}\mathcal{A}_{j,l}(\mathbf{X}_l^{(k+1)} - \mathbf{X}_{*,l})\right\|_{\mathrm{F}}^2\right)\right)\right),
\end{aligned}
\tag{44}
$$

*where $B_j = \sum_{i=n-j+1}^m \frac{4iB_{j,i}^2}{\beta_j}, C_i = \sum_{i=n-j+1}^m \frac{4iC_{j,i}^2}{\beta_j}, D_j = \max_i \frac{4iD_{j,i}}{\beta_i}, B_{j,i} = c_A L_{f_j} c_{n-j+1,i}, C_{j,i} = c_A c_{n-j+1,i} q_j, D_{j,i} = \sum_{p=j}^i c_A e_A \bar{\beta}_i c_{p,i}$. If $f_i, i = 1, \cdots, n$, all have Lipschitz continuous gradients, it follows that*

$$
\begin{aligned}
\sum_{i=1}^m \frac{1}{\beta_i}\|\mathbf{\Lambda}_i^{(k+1)} - \mathbf{\Lambda}_{*,i}\|_{\mathrm{F}}^2 &\leq \sum_{i=n-m+1}^n \tilde{B}_i\|\mathbf{X}_i^{(k+1)} - \mathbf{X}_{*,i}\|_{\mathrm{F}}^2 + \sum_{i=n-m+1}^n \tilde{C}_i\|\mathbf{X}_i^{(k+1)} - \mathbf{X}_i^{(k)}\|_{\mathrm{F}}^2 \\
&+ \left(\sum_{j=1}^m \sum_{i=1}^{n-j+1} \tilde{D}_j\left(\left(\left\|\sum_{l=i+1}^{n-j+1}\mathcal{A}_{j,l}(\mathbf{X}_l^{(k)} - \mathbf{X}_{*,l})\right\|_{\mathrm{F}}^2 + \left\|\sum_{l=i+1}^{n-j+1}\mathcal{A}_{j,l}(\mathbf{X}_l^{(k+1)} - \mathbf{X}_{*,l})\right\|_{\mathrm{F}}^2\right)\right)\right),
\end{aligned}
\tag{45}
$$

*where for $i = 1, \cdots, n, j = 1, \cdots, m$.*

$$
\tilde{B}_i = \frac{3c_{A,1}^2 L_{f_i}^2}{\beta_{\min}}, \tilde{C}_i = \frac{3\beta_{\max}^2 c_{A,1}^2 q_i}{\beta_{\min}}, \tilde{D}_j = \frac{3m\beta_{\max}^2 c_{A,1} e_A}{\beta_{\min}}.
\tag{46}
$$

*Proof.* For Scenarios 1,2 where A(2) holds, it follows from $\mathbf{\Lambda}_i^{(k+1)} - \mathbf{\Lambda}_{*,i} \in \mathrm{Im}(\mathcal{A}_{i,n-i+1}), i = 1, \cdots, m$ that

$$
\|\mathbf{\Lambda}_i^{(k+1)} - \mathbf{\Lambda}_{*,i}\| \leq c_A\|\mathcal{A}_{i,n-i+1}^{\mathrm{T}}(\mathbf{\Lambda}_i^{(k+1)} - \mathbf{\Lambda}_{*,i})\|, \quad i = 1, \cdots, m.
\tag{47}
$$

Using the optimality conditions (35) and the Lipschitz continuity of $\nabla f_i, i = n - m + 1, \dots, n$, we have

$$
\begin{aligned}
&\nabla f_i(\mathbf{X}_i^{(k+1)}) - \sum_{j=1}^{n-i+1}\mathcal{A}_{j,i}^{\mathrm{T}}\left(\beta_j \sum_{l=i+1}^{n-j+1}\mathcal{A}_{j,l}(\mathbf{X}_l^{(k)} - \mathbf{X}_l^{(k+1)}) + \mathbf{\Lambda}_j^{(k+1)}\right) + \mathcal{Q}_i(\mathbf{X}_i^{(k+1)} - \mathbf{X}_i^{(k)}) = 0, \\
&\sum_{j=1}^{n-i+1}\mathcal{A}_{j,i}^{\mathrm{T}}\mathbf{\Lambda}_{*,i} = \nabla f_i(\mathbf{X}_{*,i}).
\end{aligned}
\tag{48}
$$

Together with Assumptions 2 implies that for $i = n - m + 1, \cdots, n$, we have

$$
\begin{aligned}
\|\mathbf{\Lambda}_{n-i+1}^{(k+1)} - \mathbf{\Lambda}_{*,n-i+1}\|_{\mathrm{F}} &\leq c_A\|\mathcal{A}_{n-i+1,i}^{\mathrm{T}}(\mathbf{\Lambda}_{n-i+1}^{(k+1)} - \mathbf{\Lambda}_{*,n-i+1})\|_{\mathrm{F}} \\
&\leq c_A\bar{\beta}_{n-i+1}\left\|\sum_{j=1}^{n-i+1}\mathcal{A}_{j,i}^{\mathrm{T}}\left(\sum_{l=i+1}^{n-j+1}\mathcal{A}_{j,l}(\mathbf{X}_j^{(k)} - \mathbf{X}_{*,j})\right)\right\|_{\mathrm{F}} + c_A\bar{\beta}_{n-i+1}\left\|\sum_{j=1}^{n-i+1}\mathcal{A}_{j,i}^{\mathrm{T}}\left(\sum_{l=i+1}^{n-j+1}\mathcal{A}_{j,l}(\mathbf{X}_j^{(k+1)} - \mathbf{X}_{*,j})\right)\right\|_{\mathrm{F}} \\
&+ q_i c_A\|\mathbf{X}_i^{(k+1)} - \mathbf{X}_i^{(k)}\|_{\mathrm{F}} + L_{f_i} c_A\|\mathbf{X}_i^{(k+1)} - \mathbf{X}_{*,i}\|_{\mathrm{F}} + c_A d_A \sum_{l=1}^{n-i}\|\mathbf{\Lambda}_l^{(k+1)} - \mathbf{\Lambda}_{*,l}\|_{\mathrm{F}}),
\end{aligned}
$$

Consequently, it follows that for $i = 1, \cdots, m$ we have:

$$\|\mathbf{\Lambda}_i^{(k+1)} - \mathbf{\Lambda}_{*,i}\|_{\mathrm{F}} \leq c_A e_A \bar{\beta}_i \sum_{j=1}^{i} \left\| \left( \sum_{l=n-i+2}^{n-j+1} \mathcal{A}_{j,l}(\mathbf{X}_j^{(k)} - \mathbf{X}_{*,j}) \right) \right\|_{\mathrm{F}} + c_A e_A \bar{\beta}_i \sum_{j=1}^{i} \left\| \left( \sum_{l=n-i+2}^{n-j+1} \mathcal{A}_{j,l}(\mathbf{X}_j^{(k+1)} - \mathbf{X}_{*,j}) \right) \right\|_{\mathrm{F}}$$

$$+ q_{n-i+1} c_A \|\mathbf{X}_{n-i+1}^{(k+1)} - \mathbf{X}_{n-i+1}^{(k)}\|_{\mathrm{F}} + c_A L_{f_{n-i+1}} \|\mathbf{X}_{n-i+1}^{(k+1)} - \mathbf{X}_{*,n-i+1}\|_{\mathrm{F}} + \sum_{j=1}^{i-1} c_A d_A \|\mathbf{\Lambda}_j^{(k+1)} - \mathbf{\Lambda}_{*,j}\|).$$

According to Lemma 5, for $i = 1, \cdots, m$

$$\|\mathbf{\Lambda}_i^{(k+1)} - \mathbf{\Lambda}_{*,i}\|_{\mathrm{F}} \leq \sum_{p=1}^{i} c_{p,i} \left( c_A e_A \bar{\beta}_i \sum_{j=1}^{p} \left( \left\| \sum_{l=n-i+2}^{n-j+1} \left( \mathcal{A}_{j,l}(\mathbf{X}_l^{(k+1)} - \mathbf{X}_{*,l}) \right) \right\|_{\mathrm{F}} + \left\| \sum_{l=n-i+2}^{n-j+1} \left( \mathcal{A}_{j,l}(\mathbf{X}_l^{(k)} - \mathbf{X}_{*,l}) \right) \right\|_{\mathrm{F}} \right) \right.$$

$$\left. + c_A q_{n-p+1} \|\mathbf{X}_{n-p+1}^{(k+1)} - \mathbf{X}_{n-p+1}^{(k)}\|_{\mathrm{F}} + c_A L_{f_{n-p+1}} \|\mathbf{X}_{n-p+1}^{(k+1)} - \mathbf{X}_{*,n-p+1}\|_{\mathrm{F}} \right)$$

$$\leq \sum_{j=1}^{i} \sum_{p=j}^{i} c_{p,i} c_A e_A \bar{\beta}_i \left( \left\| \sum_{l=n-i+2}^{n-j+1} \left( \mathcal{A}_{j,l}(\mathbf{X}_l^{(k+1)} - \mathbf{X}_{*,l}) \right) \right\|_{\mathrm{F}} + \left\| \sum_{l=n-i+2}^{n-j+1} \left( \mathcal{A}_{j,l}(\mathbf{X}_l^{(k)} - \mathbf{X}_{*,l}) \right) \right\|_{\mathrm{F}} \right)$$

$$+ \sum_{j=n-i+1}^{n} c_A c_{n-j+1,i} q_j \|\mathbf{X}_j^{(k+1)} - \mathbf{X}_j^{(k)}\|_{\mathrm{F}} + \sum_{j=n-i+1}^{n} c_A c_{n-j+1,i} L_{f_j} \|\mathbf{X}_j^{(k+1)} - \mathbf{X}_{*,j}\|_{\mathrm{F}}$$

$$\leq \sum_{j=1}^{i} D_{j,i} \left( \left\| \sum_{l=n-i+2}^{n-j+1} \left( \mathcal{A}_{j,l}(\mathbf{X}_l^{(k+1)} - \mathbf{X}_{*,l}) \right) \right\|_{\mathrm{F}} + \left\| \sum_{l=n-i+2}^{n-j+1} \left( \mathcal{A}_{j,l}(\mathbf{X}_l^{(k)} - \mathbf{X}_{*,l}) \right) \right\|_{\mathrm{F}} \right)$$

$$+ \sum_{j=n-i+1}^{n} C_{j,i} \|\mathbf{X}_j^{(k+1)} - \mathbf{X}_j^{(k)}\|_{\mathrm{F}} + \sum_{j=n-i+1}^{n} B_{j,i} \|\mathbf{X}_j^{(k+1)} - \mathbf{X}_{*,j}\|_{\mathrm{F}},$$

where $B_{j,i} = c_A L_{f_i} c_{n-j+1,i}, C_{j,i} = c_A c_{n-j+1,i} q_j, D_{j,i} = \sum_{p=j}^{i} c_A e_A \bar{\beta}_i c_{p,i}$. Hence we have that

$$\sum_{i=1}^{m} \frac{1}{\beta_i} \|\mathbf{\Lambda}_i^{(k)} - \mathbf{\Lambda}_{*,i}\|_{\mathrm{F}}^2 \leq \sum_{i=1}^{m} \left( \sum_{j=1}^{i} \frac{4i D_{j,i}}{\beta_i} \left( \left\| \sum_{l=n-i+2}^{n-j+1} \left( \mathcal{A}_{j,l}(\mathbf{X}_l^{(k+1)} - \mathbf{X}_{*,l}) \right) \right\|_{\mathrm{F}}^2 + \left\| \sum_{l=n-i+2}^{n-j+1} \left( \mathcal{A}_{j,l}(\mathbf{X}_l^{(k)} - \mathbf{X}_{*,l}) \right) \right\|_{\mathrm{F}}^2 \right) \right.$$

$$\left. + \sum_{j=n-i+1}^{n} \frac{4i C_{j,i}}{\beta_i} \|\mathbf{X}_j^{(k+1)} - \mathbf{X}_j^{(k)}\|_{\mathrm{F}}^2 + \sum_{j=n-i+1}^{n} \frac{4i B_{j,i}}{\beta_i} \|\mathbf{X}_j^{(k+1)} - \mathbf{X}_{*,j}\|_{\mathrm{F}}^2 \right)$$

$$= \sum_{j=n-m+1}^{n} \sum_{i=n-j+1}^{m} \frac{4i B_{j,i}^2}{\beta_i} \|\mathbf{X}_j^{(k+1)} - \mathbf{X}_{*,j}\|_{\mathrm{F}}^2$$

$$+ \sum_{j=1}^{m} \sum_{i=1}^{n-j+1} D_j \left( \left\| \sum_{l=i+1}^{n-j+1} \mathcal{A}_{j,l}(\mathbf{X}_l^{(k)} - \mathbf{X}_{*,l}) \right\|_{\mathrm{F}}^2 + \left\| \sum_{l=i+1}^{n-j+1} \mathcal{A}_{j,l}(\mathbf{X}_l^{(k+1)} - \mathbf{X}_{*,l}) \right\|_{\mathrm{F}}^2 \right)$$

$$+ \sum_{j=n-m+1}^{n} \sum_{i=n-j+1}^{m} \frac{4i C_{j,i}^2}{\beta_i} \|\mathbf{X}_j^{(k+1)} - \mathbf{X}_j^{(k)}\|_{\mathrm{F}}^2$$

$$= \sum_{j=n-m+1}^{n} B_j \|\mathbf{X}_j^{(k+1)} - \mathbf{X}_{*,j}\|_{\mathrm{F}}^2 + \sum_{j=1}^{m} D_j \sum_{i=1}^{n-j+1} \left( \left\| \sum_{l=i+1}^{n-j+1} \mathcal{A}_{j,l}(\mathbf{X}_l^{(k)} - \mathbf{X}_{*,l}) \right\|_{\mathrm{F}}^2 + \left\| \sum_{l=i+1}^{n-j+1} \mathcal{A}_{j,l}(\mathbf{X}_l^{(k+1)} - \mathbf{X}_{*,l}) \right\|_{\mathrm{F}}^2 \right)$$

$$+ \sum_{j=n-m+1}^{n} C_j \|\mathbf{X}_j^{(k+1)} - \mathbf{X}_j^{(k)}\|_{\mathrm{F}}^2,$$

where the first inequality follows from the Cauchy-Schwarz inequality.

Consider the case where $f_i, i = 1, \cdots, n$ are Lipschitz continuously differentiable. It follows from $\overline{\boldsymbol{\Lambda}}^{(k+1)} - \overline{\boldsymbol{\Lambda}}_* \subseteq \mathrm{Im}\mathcal{A}$ and (48) that

$$
\sum_{i=1}^{m} \frac{1}{\beta_i} \|\boldsymbol{\Lambda}_i^{(k+1)} - \boldsymbol{\Lambda}_{*,i}\|_{\mathrm{F}}^2 \leq \frac{c_{A,1}}{\beta_{\min}} \left\| \mathcal{A}^{\mathrm{T}}(\boldsymbol{\Lambda}^{(k+1)} - \boldsymbol{\Lambda}_*) \right\|_{\mathrm{F}}^2
$$

$$
\leq \frac{3}{\beta_{\min}} \sum_{i=1}^{n} \left\| \sum_{j=1}^{\min\{n-i+1,m\}} \mathcal{A}_{j,i}^{\mathrm{T}} \left( \sum_{l=i+1}^{n-j+1} \beta_j \mathcal{A}_{j,l}(\mathbf{X}_l^{(k)} - \mathbf{X}_l^{(k+1)}) \right) \right\|_{\mathrm{F}}^2
$$

$$
+ \sum_{i=1}^{n} \frac{3c_{A,1}L_{f_i}^2}{\beta_{\min}} \|\mathbf{X}_i^{(k+1)} - \mathbf{X}_{k,*}\|_{\mathrm{F}}^2 + \sum_{i=1}^{n} \frac{3c_{A,1}}{\beta_{\min}} \|\mathbf{X}_i^{(k)} - \mathbf{X}_{*,i}\|_{\mathcal{Q}_i}^2
$$

$$
\leq \frac{3m\beta_{\max}^2 c_{A,1}e_A}{\beta_{\min}} \left( \sum_{i=1}^{n} \sum_{j=1}^{\min\{n-i+1,m\}} \left( \left\| \sum_{l=i+1}^{n-j+1} \mathcal{A}_{i,j}(\mathbf{X}_j^{(k)} - \mathbf{X}_{*,j}) \right\|_{\mathrm{F}}^2 + \left\| \sum_{l=i+1}^{n-j+1} \mathcal{A}_{i,j}(\mathbf{X}_j^{(k+1)} - \mathbf{X}_{*,j}) \right\|_{\mathrm{F}}^2 \right) \right)
$$

$$
+ \frac{3}{\beta_{\min}} \sum_{i=1}^{n} c_{A,1}L_{f_i}^2 \|\mathbf{X}_i^{(k+1)} - \mathbf{X}_{k,*}\|_{\mathrm{F}}^2 + \sum_{i=1}^{n} \frac{3c_{A,1}}{\beta_{\min}} \|\mathbf{X}_i^{(k)} - \mathbf{X}_{*,i}\|_{\mathcal{Q}_i}^2.
$$

Consequently, (44) holds with $\tilde{B}_i = \frac{3c_{A,1}L_{f_i}^2}{\beta_{\min}}, \tilde{C}_i = \frac{3c_{A,1}}{\beta_{\min}}, \tilde{D}_i = \frac{3m\beta_{\max}^2 c_{A,1}e_A}{\beta_{\min}}$. $\qquad\square$

Lemma 11 is used to bound $\sum_{i=1}^{m} \frac{1}{\beta_i} \|\boldsymbol{\Lambda}_i^{(k+1)} - \boldsymbol{\Lambda}_{*,i}\|_{\mathrm{F}}^2$ under different scenarios. Specifically, (44) is used in Scenarios 1 and 2, whereas (46) is used in Scenarios 3 and 4.

**Proof of Theorem 5**

*Proof.* According to (13), we have that $\tilde{r}^{(k)} - \tilde{r}^{(k+1)} \geq C$, where $C = \sum_{i=2}^{n} \left[ (\kappa_{f_i} - \beta_{\max}a_i) \|\mathbf{X}_i^{(k+1)} - \mathbf{X}_{*,i}\|_{\mathrm{F}}^2 \right] + \sum_{i=1}^{m} \frac{\beta_i}{2} u_{1,i,k} + \sum_{j=1}^{m} \frac{\beta_j}{2} \sum_{i=1}^{n-j+1} \|\sum_{l=i+1}^{n-j+1} \mathcal{A}_{j,l}\mathbf{X}_l^{(k)} - \mathcal{A}_{j,l}\mathbf{X}_{*,l}\|_{\mathrm{F}}^2 + \frac{1}{2} \sum_{i=1}^{n} \|\mathbf{X}_i^{(k+1)} - \mathbf{X}_i^{(k)}\|_{\mathcal{Q}_i}^2$. We need to prove that $C > \delta_1 \tilde{r}^{(k+1)}$. According to the proof of Lemma 2, we have the following inequality:

$$
\sum_{j=1}^{m} \frac{\beta_j}{2} \sum_{i=1}^{n-j+1} \left\| \sum_{l=i+1}^{n-j+1} \mathcal{A}_{j,l}\mathbf{X}_l^{(k+1)} - \mathcal{A}_{j,l}\mathbf{X}_{*,l} \right\|_{\mathrm{F}}^2 \leq \sum_{i=2}^{n} \frac{\beta_{\max}a_i}{2} \|\mathbf{X}_i^{(k+1)} - \mathbf{X}_{*,i}\|_{\mathrm{F}}^2. \tag{49}
$$

We first consider Scenario 1. In cases 1 and 2, where $\mathcal{Q}_i = 0, i = 1, \cdots, n - m$, we only need to bound $\sum_{i=1}^{m} \frac{1}{\beta_i} \|\boldsymbol{\Lambda}_i^{(k)} - \boldsymbol{\Lambda}_{*,i}\|_{\mathrm{F}}^2$ and $\frac{1}{2} \sum_{i=n-m+1}^{n} \|\mathbf{X}_i^{(k)} - \mathbf{X}_{*,i}\|_{\mathcal{Q}_i}^2$. According to the strong convexity of $f_i, i = n - m + 1, \cdots, n$ and Lemma 11, we have

$$
\sum_{i=1}^{m} \frac{1}{\beta_i} \|\boldsymbol{\Lambda}_i^{(k+1)} - \boldsymbol{\Lambda}_{*,i}\|_{\mathrm{F}}^2 \leq \left( \sum_{i=n-m+1}^{n} B_i \|\mathbf{X}_i^{(k+1)} - \mathbf{X}_{*,i}\|_{\mathrm{F}}^2 \right) + \sum_{i=n-m+1}^{n} C_i \|\mathbf{X}_i^{(k+1)} - \mathbf{X}_i^{(k)}\|_{\mathrm{F}}^2
$$

$$
+ \left( \sum_{j=1}^{m} \sum_{i=1}^{n-j+1} D_j \left( \left( \left\| \sum_{l=i+1}^{n-j+1} \mathcal{A}_{j,l}(\mathbf{X}_l^{(k)} - \mathbf{X}_{*,l}) \right\|_{\mathrm{F}}^2 + \left\| \sum_{l=i+1}^{n-j+1} \mathcal{A}_{j,l}(\mathbf{X}_l^{(k+1)} - \mathbf{X}_{*,l}) \right\|_{\mathrm{F}}^2 \right) \right) \right).
$$

Consequently, we have

$$
\tilde{r}^{(k+1)} \leq \sum_{i=2}^{n} \beta_{\max}a_i \|\mathbf{X}_i^{(k+1)} - \mathbf{X}_{*,i}\|_{\mathrm{F}}^2 + \left( \sum_{i=n-m+1}^{n} B_i \|\mathbf{X}_i^{(k+1)} - \mathbf{X}_{*,i}\|_{\mathrm{F}}^2 \right) + \sum_{i=n-m+1}^{n} C_i \|\mathbf{X}_i^{(k+1)} - \mathbf{X}_i^{(k)}\|_{\mathrm{F}}^2
$$

$$
+ \left( \sum_{j=1}^{m} \sum_{i=1}^{n-j+1} D_j \left( \left( \left\| \sum_{l=i+1}^{n-j+1} \mathcal{A}_{j,l}(\mathbf{X}_l^{(k)} - \mathbf{X}_{*,l}) \right\|_{\mathrm{F}}^2 + \left\| \sum_{l=i+1}^{n-j+1} \mathcal{A}_{j,l}(\mathbf{X}_l^{(k+1)} - \mathbf{X}_{*,l}) \right\|_{\mathrm{F}}^2 \right) \right) \right)
$$

$$
+ \frac{1}{2} \sum_{i=n-m+1}^{n} \|\mathbf{X}_i^{(k+1)} - \mathbf{X}_{*,i}\|_{\mathcal{Q}_i}^2.
$$

It follows from (43) and (49) that (14) holds with $\delta = \min\{\delta_{1,1}, \delta_{1,2}, \delta_{1,3}\}$, where $\delta_{1,1} = \min_{i=\min\{n-m+1,2\},\cdots,n} \frac{\kappa_{f_i} - \beta_{\max} a_i}{(\beta_{\max} + D_{\max})a_i + B_i + q_i/2}$, $\delta_{1,2} = \min_{i=n-m+1,\cdots,n} \frac{q_i^-}{C_i}$, $\delta_{1,3}, \min_{j=1,\cdots,m} \frac{\beta_j}{2D_j}$, $D_{\max} = \max_{j=1,\cdots,m} D_j$. We note that in Case 1, $q_i^- = C_i = 0$ for $i = n - m + 1, \cdots, n$, hence $\delta_{1,2} = \infty$.

In Cases 3 and 4, due to the strong convexity of $f_i, i = 2, \cdots, n$, (37) and Lemma 11, we obtain

$$\tilde{r}^{(k+1)} \le \sum_{i=2}^{n} a_i \|\mathbf{X}_i^{(k+1)} - \mathbf{X}_{*,i}\|_{\mathrm{F}}^2 + \left( \sum_{i=n-m+1}^{n} B_i \|\mathbf{X}_i^{(k+1)} - \mathbf{X}_{*,i}\|_{\mathrm{F}}^2 \right) + \sum_{i=n-m+1}^{n} C_i \|\mathbf{X}_i^{(k+1)} - \mathbf{X}_i^{(k)}\|_{\mathrm{F}}^2$$
$$+ \left( \sum_{j=1}^{m} \sum_{i=1}^{n-j+1} D_j \left( \left( \left\| \sum_{l=i+1}^{n-j+1} \mathcal{A}_{j,l}(\mathbf{X}_l^{(k)} - \mathbf{X}_{*,l}) \right\|_{\mathrm{F}}^2 + \left\| \sum_{l=i+1}^{n-j+1} \mathcal{A}_{j,l}(\mathbf{X}_l^{(k+1)} - \mathbf{X}_{*,l}) \right\|_{\mathrm{F}}^2 \right) \right) \right)$$
$$+ \frac{1}{2} \sum_{i=2}^{n} \|\mathbf{X}_i^{(k+1)} - \mathbf{X}_{*,i}\|_{\mathcal{Q}_i}^2 + 2c_{A,1} q_1^+ (\sum_{j=1}^{m} u_{1,j,k} + \sum_{i=1}^{m} \| \sum_{j=2}^{n-i+1} \mathcal{A}_{i,j}(\mathbf{X}_1^{(k)} - \mathbf{X}_{*,1})\|_{\mathrm{F}}^2).$$

Then the linear convergence result (14) holds where $\delta = \min\{\delta_{1,4}, \delta_{1,5}, \delta_{1,6}\}$, where $\delta_{1,4} = \min_{i=1,\cdots,n} \frac{q_i^-}{C_i}, \delta_{1,5} = \min_{i=\min\{n-m+1,2\},\cdots,n} \frac{\kappa_{f_i} - \beta_{\max} a_i}{(\beta_{\max} + D_{\max})a_i + B_i + q_i/2}, \delta_{1,6} = \min_{j=1,\cdots,m} \{ \frac{\beta_{\min}}{4c_{A,1} q_1^+}, \frac{\beta_j}{2D_j + 4c_{A,1} q_1^+} \}.$

We next consider Scenario 2. Due to the strong convexity of $f_i, i = 1, \cdots, n$, the linear convergence result (14) holds. According to Lemma 10, the following inequality holds:

$$\tilde{r}^{(k+1)} \le \sum_{i=2}^{n} a_i \|\mathbf{X}_i^{(k+1)} - \mathbf{X}_{*,i}\|_{\mathrm{F}}^2 + \left( \sum_{i=n-m+1}^{n} B_i \|\mathbf{X}_i^{(k+1)} - \mathbf{X}_{*,i}\|_{\mathrm{F}}^2 \right) + \sum_{i=n-m+1}^{n} C_i \|\mathbf{X}_i^{(k+1)} - \mathbf{X}_i^{(k)}\|_{\mathrm{F}}^2$$
$$+ \left( \sum_{j=1}^{m} \sum_{i=1}^{n-j+1} D_j \left( \left( \left\| \sum_{l=i+1}^{n-j+1} \mathcal{A}_{j,l}(\mathbf{X}_l^{(k)} - \mathbf{X}_{*,l}) \right\|_{\mathrm{F}}^2 + \left\| \sum_{l=i+1}^{n-j+1} \mathcal{A}_{j,l}(\mathbf{X}_l^{(k+1)} - \mathbf{X}_{*,l}) \right\|_{\mathrm{F}}^2 \right) \right) \right)$$
$$+ \frac{1}{2} \sum_{i=1}^{n} \|\mathbf{X}_i^{(k+1)} - \mathbf{X}_{*,i}\|_{\mathcal{Q}_i}^2.$$

The linear convergence result (14) holds with $\delta = \min\{\delta_{2,1}, \delta_{2,2}, \delta_{2,3}\}$, where $\delta_{2,1} = \min_{i=1,\cdots,n} \frac{q_i^-}{\beta_{\max} C_i}, \delta_{2,2} = \min_{i=1,\cdots,n} \frac{\kappa_{f_i} - \beta_{\max} a_i}{(\beta_{\max} + D_{\max})a_i + B_i + q_i}, \delta_{2,3} = \min_{j=1,\cdots,m} \{ \frac{\beta_j}{2D_j} \}.$

For Scenario 3, we have

$$\tilde{r}^{(k+1)} \le \sum_{i=2}^{n} a_i \|\mathbf{X}_i^{(k+1)} - \mathbf{X}_{*,i}\|_{\mathrm{F}}^2 + \left( \sum_{i=n-m+1}^{n} \tilde{B}_i \|\mathbf{X}_i^{(k+1)} - \mathbf{X}_{*,i}\|_{\mathrm{F}}^2 \right) + \sum_{i=n-m+1}^{n} c_i \|\mathbf{X}_i^{(k+1)} - \mathbf{X}_i^{(k)}\|_{\mathrm{F}}^2$$
$$+ \left( \sum_{j=1}^{m} \sum_{i=1}^{n-j+1} \tilde{D}_j \left( \left( \left\| \sum_{l=i+1}^{n-j+1} \mathcal{A}_{j,l}(\mathbf{X}_l^{(k)} - \mathbf{X}_{*,l}) \right\|_{\mathrm{F}}^2 + \left\| \sum_{l=i+1}^{n-j+1} \mathcal{A}_{j,l}(\mathbf{X}_l^{(k+1)} - \mathbf{X}_{*,l}) \right\|_{\mathrm{F}}^2 \right) \right) \right)$$
$$+ \frac{1}{2} \sum_{i=2}^{n} \|\mathbf{X}_i^{(k+1)} - \mathbf{X}_{*,i}\|_{\mathcal{Q}_i}^2 + 2c_A q_1^+ (\sum_{j=1}^{m} u_{1,j,k} + \sum_{i=1}^{m} \| \sum_{j=2}^{n-i+1} \mathcal{A}_{i,j}(\mathbf{X}_1^{(k)} - \mathbf{X}_{*,1})\|_{\mathrm{F}}^2).$$

Consequently, the Q-linear convergence result (14) holds with $\delta = \min\{\delta_{3,1}, \delta_{3,2}, \delta_{3,3}\}$, where $\delta_{3,1} = \min_{i=2,\cdots,n} \frac{q_i^-}{\beta_{\max} c_i}, \delta_{3,2} = \min_{i=2,\cdots,n} \frac{\kappa_{f_i} - \beta_{\max} a_i}{(\beta_{\max} + \tilde{D}_{\max})a_i + \tilde{B}_i + q_i}, \delta_{3,3} = \min_{j=1,\cdots,m} \{ \frac{\beta_{\min}}{4c_{A,1} q_1^+}, \frac{\beta_j}{2\tilde{D}_j} \}.$

For Scenario 4, the following inequality holds:

$$\tilde{r}^{(k+1)} \leq \sum_{i=2}^{n} a_i \|\mathbf{X}_i^{(k+1)} - \mathbf{X}_{*,i}\|_{\mathrm{F}}^2 + \left( \sum_{i=n-m+1}^{n} \tilde{B}_i \|\mathbf{X}_i^{(k+1)} - \mathbf{X}_{*,i}\|_{\mathrm{F}}^2 \right) + \sum_{i=n-m+1}^{n} \tilde{C}_i \|\mathbf{X}_i^{(k+1)} - \mathbf{X}_i^{(k)}\|_{\mathrm{F}}^2$$

$$+ \left( \sum_{j=1}^{m} \sum_{i=1}^{n-j+1} \tilde{D}_j \left( \left\| \sum_{l=i+1}^{n-j+1} \mathcal{A}_{j,l}(\mathbf{X}_l^{(k)} - \mathbf{X}_{*,l}) \right\|_{\mathrm{F}}^2 + \left\| \sum_{l=i+1}^{n-j+1} \mathcal{A}_{j,l}(\mathbf{X}_l^{(k+1)} - \mathbf{X}_{*,l}) \right\|_{\mathrm{F}}^2 \right) \right)$$

$$+ \frac{1}{2} \sum_{i=1}^{n} \|\mathbf{X}_i^{(k+1)} - \mathbf{X}_{*,i}\|_{\mathcal{Q}_i}^2.$$

Consequently, it follows that (14) holds with $\delta = \min\{\delta_{4,1}, \delta_{4,2}, \delta_{4,3}\}$, where $\delta_{4,1} = \min_{i=1,\cdots,n} \frac{q_i^-}{\beta_{\max} \tilde{C}_i}, \delta_{4,2} = \min_{i=1,\cdots,n} \frac{\kappa_{f_i} - \beta_{\max} a_i}{(\beta_{\max} + \tilde{D}_{\max}) a_i + \tilde{B}_i + q_i}, \delta_{4,3} = \min_{j=1,\cdots,m} \left\{ \frac{\beta_j}{2\tilde{D}_j} \right\}.$

$\square$

*Proof.* Note that under all four scenarios, we have shown that the sequence

$$\left( \sum_{j=2}^{n} \mathcal{A}_j \mathbf{X}_j^{(k)}, \sum_{j=3}^{n} \mathcal{A}_j \mathbf{X}_j^{(k)}, \ldots, \sum_{j=n}^{n} \mathcal{A}_j \mathbf{X}_j^{(k)}, \overline{\mathbf{\Lambda}}^{(k)} \right)$$

converges Q-linearly. It follows that $\overline{\mathbf{\Lambda}}^{(k)}$ and $\sum_{j=i+1}^{n} \mathcal{A}_j \mathbf{X}_j^{(k)}, i = 1, \ldots, n-1$ converge R-linearly. Since any part of a Q-linear convergent quantity converges R-linearly, which also implies that $\mathcal{A}_2 \mathbf{X}_2^{(k)}, \ldots, \mathcal{A}_n \mathbf{X}_n^{(k)}$ converge R-linearly. It now follows from (39) that $\mathcal{A}_1 \mathbf{X}_1^{(k)}$ converges R-linearly. Consider cases 1 and 3, where $\mathcal{Q}_n$ is zero. By setting $i = n$ in (38), one obtains

$$(\mathbf{X}_n^{(k+1)} - \mathbf{X}_{*,n})^\top \mathcal{A}_n^\top (\mathbf{\Lambda}_1^{(k+1)} - \mathbf{\Lambda}_{*,1}) \geq \kappa_{f_n} \|\mathbf{X}_n^{(k+1)} - \mathbf{X}_{*,n}\|_{\mathrm{F}}^2,$$

which implies that $\|\mathbf{X}_n^{(k+1)} - \mathbf{X}_{*,n}\| \|\mathcal{A}_n\| \|\mathbf{\Lambda}_1^{(k+1)} - \mathbf{\Lambda}_{*,1}\| \geq \kappa_{f_n} \|\mathbf{X}_n^{(k+1)} - \mathbf{X}_{*,n}\|_{\mathrm{F}}^2$, i.e., $\|\mathbf{X}_n^{(k+1)} - \mathbf{X}_{*,n}\| \leq \frac{\|\mathcal{A}_n\|}{\kappa_{f_n}} \|\mathbf{\Lambda}_1^{(k+1)} - \mathbf{\Lambda}_{*,1}\|$. The R-linear convergence of $\mathbf{X}_n^{(k)}$ follows from the fact that $\mathbf{\Lambda}_1^{(k)}$ converges R-linearly. $\square$

### A.7. Additional numerical experiment

The tables report the main pADMM and classical ADMM numerical values, while the figures additionally compare linearized ADMM and semiproximal ADMM. The leukemia dataset consists of 72 observations on 7129 genes measured with DNA microarrays, i.e., $n = 72, p = 7129$. The Alzheimer's disease MRI dataset is a 2D MRI image dataset, which is labeled with moderate, mild, very mild, and no dementia. We take the first 50 images in each tag and compress the original image size to $70 \times 70$. The results of the compared algorithms under nonconvex and strongly convex settings are shown in Tables 5, 6, 7, and 8. It is shown that pADMM requires fewer iterations and time when obtaining the same level of accuracy.

|  | pADMM-MCP | ADMM-MCP | L-ADMM-MCP | Semi-ADMM-MCP | pADMM-SCAD | ADMM-SCAD | L-ADMM-SCAD | Semi-ADMM-SCAD |
|---|---|---|---|---|---|---|---|---|
| $L_1$ | 4.52e-07 | 4.36e-06 | 6.54e-06 | 1.25e-06 | 4.54e-07 | 4.18e-06 | 6.27e-06 | 1.24e-06 |
| $L_2$ | 1.20e-02 | 1.98e-02 | 2.97e-02 | 6.19e-02 | 1.22e-02 | 1.99e-02 | 2.99e-02 | 6.24e-02 |
| Time(s) | 19.64 | 41.86 | 25.56 | 45.64 | 20.23 | 41.72 | 33.32 | 43.46 |
| Iteration | 61 | 68 | 67 | 67 | 58 | 65 | 65 | 68 |

*Table 5.* Numerical results of generalized elastic net regression for leukemia data.

|  | pADMM-MCP | ADMM-MCP | L-ADMM-MCP | Semi-ADMM-MCP | pADMM-SCAD | ADMM-SCAD | L-ADMM-SCAD | Semi-ADMM-SCAD |
|---|---|---|---|---|---|---|---|---|
| $L_1$ | 7.73e-07 | 1.17e-05 | 1.16e-06 | 2.08e-06 | 7.24e-07 | 5.31e-05 | 1.09e-06 | 3.38e-06 |
| $L_2$ | 8.33e-03 | 1.24e-02 | 1.25e-02 | 4.69e-02 | 8.25e-03 | 2.89e-02 | 1.24e-02 | 6.20e-02 |
| Time(s) | 16.77 | 95.84 | 56.30 | 66.40 | 16.87 | 50.46 | 57.23 | 67.35 |
| Iteration | 117 | 127 | 121 | 130 | 119 | 100 | 122 | 132 |

*Table 6.* Numerical results of generalized elastic net regression for the Alzheimer's disease MRI dataset.

| Algorithm | pADMM-BA1 | ADMM-BA1 | L-ADMM-BA1 | Semi-ADMM-BA1 | pADMM-BA2 | ADMM-BA2 | L-ADMM-BA2 | Semi-ADMM-BA2 |
|---|---|---|---|---|---|---|---|---|
| Residual error | 9.78e-07 | 9.90e-07 | 1.21e-06 | 1.06e-06 | 9.34e-07 | 4.46e-07 | 1.18e-06 | 1.01e-06 |
| Relative error | 8.29e-09 | 7.22e-09 | 1.34e-08 | 9.12e-09 | 1.75e-11 | 1.72e-11 | 2.81e-11 | 2.03e-11 |
| Time(s) | 11.6 | 12.56 | 13.84 | 12.43 | 164.89 | 223.10 | 268.71 | 236.58 |
| Iteration | 371 | 381 | 444 | 398 | 583 | 600 | 711 | 646 |

*Table 7.* Numerical results of problem (16) on synthetic data.

| Algorithm | pADMM-tripart1 | ADMM-tripart1 | L-ADMM-tripart1 | Semi-ADMM-tripart1 | pADMM-tripart2 | ADMM-tripart2 | L-ADMM-tripart2 | Semi-ADMM-tripart2 |
|---|---|---|---|---|---|---|---|---|
| Residual error | 9.90e-07 | 9.94e-07 | 2.49e-06 | 9.84e-07 | 9.33e-07 | 9.37e-07 | 2.34e-06 | 9.28e-07 |
| Relative error | 1.50e-11 | 1.51e-11 | 3.78e-11 | 1.49e-11 | 2.23e-12 | 2.73e-12 | 6.83e-12 | 2.70e-12 |
| Time(s) | 308.36 | 395.47 | 355.74 | 462.54 | 734.40 | 568.55 | 799.35 | 807.84 |
| Iteration | 1992 | 1892 | 2002 | 2482 | 2173 | 2232 | 2126 | 2273 |

*Table 8.* Numerical results of problem (16) on the NMCNF dataset. The algorithm suffix indicates the dataset employed.

