# OpenReview forum: "A proximal ADMM for multiblock problems with block anti-upper triangular constraints"
_ICML.cc/2026/Conference — ICML 2026 regular_

### Official Review · Reviewer_ZSZP · 2026-02-19

**Soundness:** 3
**Presentation:** 3
**Significance:** 2
**Originality:** 2
**Overall Recommendation:** 4
**Confidence:** 3

**Summary:**

This paper presents a convergence analysis of the proximal Alternating Direction Method of Multipliers (ADMM) for problems with block anti-upper triangular constraints. It breaks the strong assumptions such as full column rank of the entire constraint matrix—many problems with block anti-upper triangular structures do not satisfy these requirements. The authors introduce a new assumption and use it to establish global convergence of the proximal ADMM for nonconvex problems and linear convergence rate in the strongly convex setting. This work extends the theoretical understanding of multi-block ADMM to more general cases involving block anti-upper triangular constraints.

**Compliance With Llm Reviewing Policy:**

Affirmed.

**Final Justification:**

I raised my original score. The author solves some concerns but the experiment performance still needs to be improved.

**Key Questions For Authors:**

1. What the difference between Assumption A(2) in Page 4 and the assumption in the essay "Global Convergence of ADMM in Nonconvex Nonsmooth Optimization"? Is it a special scenario of the assumption in the latter essay?
2. What is the impact of hyperparameters on experimental results? There is no discussion in the article.
3. Adopting a relative difference, such as (f_i - f_{i-1}) / f_{i-1}, as the stopping criterion would be more appropriate.
4. In non-convex settings, PADMM appears to achieve higher solution accuracy for the error metric selected in the paper. Is there an interpretable explanation for this?
5. Is there any counterexample for this problem?

**Limitations:**

1. There are still some strict assumptions for the objective function like Lipschitz differentiable. Can this assumption be overcame?
2. It seems that the advantages in low-dimension setting is not significant.

**Strengths And Weaknesses:**

Strengths:
The algorithmic formulation is well-articulated, and the numerical experiments are clearly described.

Weaknesses:
The proof is incomplete, and the appendix materials are not integrated into the main text. While the focus is on the theoretical contribution, the manuscript fails to highlight where the key challenges or difficulties in the proofs lie.

---

> ### Author Rebuttal · Authors · 2026-03-30
>
> # On weakness
>
> We would like to clarify that the complete proofs are all in appendix and the **appendix is included in the supplementary materials**. We will integrate the appendix into the main text in the revised version.
>
> We highlighted the key challenges and the main difficulties along with the statement of lemmas and theorems, and also in the proofs, which are distributed in Section 2 and the Appendix.
> Specifically, for the **nonconvex** part, we weaken the classical **range assumption** and establish a new convergence analysis for problems with **block anti-upper triangular constraints** where the range assumption fails but A(2) still holds. Therefore, we have to modify some theoretical results. For example, Lemma 4 in Appendix ensures that
> $
> \sum_{i=1}^m \frac{1}{\beta_i} ||\Delta \Lambda_i^{(k+1)} ||_F^2
> $
> can be bounded by
>
> $
> ||\Delta X_i^{(k)}|| \quad \text{and} \quad ||\Delta X_i^{(k+1)}||, \quad i=n-m+1,\dots,n,
> $
>
> which is a generalization of the existing results. The proofs rely on Assumption A(2) and are therefore novel.
>
> For the **strongly convex** part, we analyze four scenarios, each with four cases. The main technical challenge is to control
>
> $
> \sum_{i=1}^m \frac{1}{\beta_i} ||\Lambda_i^{(k+1)} - \Lambda_{*,i}||_F^2
> $
> under these different settings. In particular, this term must be handled uniformly across all sixteen cases, which requires a delicate case-by-case analysis. This is precisely where the newly introduced Assumption (A2) plays a crucial role. Compared with existing works, they only consider the case \(m=1\).
>
>
> # On questions
>
> 1.   Assumption A(2) is a **weaker version** of the range assumption in the work you cite. If the range assumption holds, then A(2) is automatically satisfied. The four examples discussed in our paper are all practical applications where the classical range assumption fails, but A(2) still holds. For more discussions, we refer to the rebuttal to Reviewer LYmF.
>
> 2.  We tested pADMM under different choices of the parameter \(q\), under both strongly convex and nonconvex settings.  \(q\) is a key parameter, and Assumption A(4) is imposed on \(q\); in particular, \(q\) must be sufficiently large for A(4) to hold. Our results are shown in Figures 5-8 at [link](https://anonymous.4open.science/r/pADMM-7987).
> For both nonconvex and strongly convex experiments, a larger \(q\) generally accelerates convergence without affecting final accuracy.
>
> 3. For relative differences of the form
> $
> \frac{|f_i - f_{i-1}|}{|f_{i-1}|},
> $
> such a criterion may not be suitable in practice. If an algorithm converges slowly, $f_i$ may be approximately equal to $f_{i-1}$ while $X_i$ is far from the optimal $X_*$. Two algorithms may then report solutions of very different accuracies, leading to unfair comparisons.
> We believe a suitable stopping criterion is related to first-order optimality:
> - **Nonconvex case:** Theorems 3 and 5 indicate that when $L_1^k$ and $L_2^k$ are small, the gradient tends to zero (see Lemma 8 in Appendix). These are suitable indicators.
> - **Strongly convex case:** Since the optimal solution is known or approximated accurately, the distance $||X^k - X_*||$ is a natural and reliable stopping criterion.
>
> 4.   In nonconvex problems, the proximal term stabilizes each block update by preventing iterates from moving too far from the previous step. It can also alleviate nonconvexity of subproblems and improve numerical properties, so that subproblems that may be ill-conditioned in vanilla ADMM while become convex or more tractable after adding the proximal term. This results in a more stable and robust iterative trajectory with less oscillation and smaller residual-based errors.
>
>  5
> Consider the following example where the objective does not have a Lipschitz gradient:
> $$
> \min_{X_1,X_2} \ -||X_1||_1 + ||X_2||_1 \quad \text{s.t. } X_1 = X_2
> $$
> Apply pADMM to this problem with penalty parameter $\beta>0$ and proximal parameters $q_1 \ge 0$, $q_2 \ge 0$. Assume $q_1 < \frac{\beta}{2}$. With initial point
> $
> (X_1^{(0)}, X_2^{(0)}, \Lambda^{(0)}) = ( -\frac{2}{\beta + 4q_1}, 0, -\frac{\beta}{\beta + 4q_1} ),
> $
> the generated sequence satisfies
> $$
> (X_1^{(2k+1)}, X_2^{(2k+1)}, \Lambda^{(2k+1)}) = ( \frac{2}{\beta + 4q_1}, 0, \frac{\beta}{\beta + 4q_1} )
> $$
> $$
> (X_1^{(2k)}, X_2^{(2k)}, \Lambda^{(2k)}) = ( -\frac{2}{\beta + 4q_1}, 0, -\frac{\beta}{\beta + 4q_1} ).
> $$
> It follows that pADMM diverges. Thus, Lipschitz continuity of the gradient is essential for convergence in nonconvex ADMM.
>
> # On Limitations
>
> 1.  The Lipschitz gradient condition cannot be relaxed. It is essential for controlling $
> \sum_{i=1}^m \frac{1}{\beta_i} ||\Delta \Lambda_i^{(k+1)} ||_F^2
> $, a key step in establishing sufficient decrease and convergence. See point 5 in questions for the counterexample.
>
> 2. Low-dimensional subproblems can be solved very quickly, so absolute runtime gaps are small. Nevertheless, appendix tables show both runtime and iteration counts remain favorable.

---

> > ### Author Rebuttal · Reviewer_ZSZP · 2026-04-01
> >
> > I appreciate your responses and have no further questions at this time.

---

> > > ### Author Response · Authors · 2026-04-01
> > >
> > > Thank you very much for your careful reading and feedback. Since your concerns appear to have been fully addressed and the other two reviewers have given positive evaluations, we would be very grateful if you could consider raising your score. If you have any further questions or issues, please feel free to let us know at any time.

---

### Official Review · Reviewer_PLkU · 2026-03-12

**Soundness:** 3
**Presentation:** 4
**Significance:** 2
**Originality:** 3
**Overall Recommendation:** 4
**Confidence:** 2

**Summary:**

The paper establishes convergence properties of a proximal Alternating Direction Method of Multipliers (ADMM) method. Novel to their approach is, that they consider block anti-upper triangular constraints to the problem. They argue, that they are the first to consider this particular type of constraints. They present an ADMM algorithm and provide assumptions under which the algorithm converges. A key point is, that the provided assumptions are weaker than standard assumptions in the literature for commonly used constraints. These assumptions mainly concern the range of the operators that are used in the constraints. Examples are provided, of problems that fall under their setup. In particular, for some of these problems, they claim that the standard assumptions to obtain convergence results, do not hold. This shows that their approach yields convergence results, that were not previously established. For the convergence analysis, they make a distinction for the objective function. In the general non-convex case, they provide global convergence as discussed. In the case that the functions are strongly convex, they additionally provide rates of convergence.

**Compliance With Llm Reviewing Policy:**

Affirmed.

**Key Questions For Authors:**

No.

**Limitations:**

yes

**Strengths And Weaknesses:**

The paper is of technical nature and as that seems convincing. It contains the main steps that are used to establish convergence, while the detailed proofs are kept in the supplementary material. The general, high-level approach to proving convergence seems to use established methodology and the authors discuss the difficulties that arise for their specific setup and how consequently their proof technique differs. For the problems, where experiments were presented, convergence can be observed and the proposed algorithm seems to behave well. The convergence result still appears to rely on a Kurdyka-Łojasiewicz property that is assumed, where I am not sure whether it is properly justified if it holds. The result in question is cited from a the literature regarding the same algorithm with stronger assumptions on the constraints, so it might be an appropriate assumption. Generally the paper is well structured. The aim is clearly formulated as well as the main steps of the proof. While it is clearly stated how they differ from the existing literature, they refer to the range assumption used in existing works and explicitly stating it would have helped understanding their contribution better. The contribution advances the theoretical background of a particular optimization method, making its scope rather domain specific. Although the use of the ADMM also in the field of Machine Learning is highlighted and it is shown that existing methods lack convergence properties, I am not sure of the general significance for Machine Learning. The provided approach seems to be original, since it is the first work that leverages this specific constraint set and although proofs are based on existing proofs, they explain how they have adapted them to work in their setup.

---

> ### Author Rebuttal · Authors · 2026-03-30
>
> We thank the reviewer for the careful reading, the positive assessment of our work, and the comments that will help improve this paper. Below we respond to the main points one by one.
>
> ### 1. For  "The convergence result still appears to rely on a Kurdyka-Łojasiewicz property that is assumed, where I am not sure whether it is properly justified if it holds."
>
> Thanks for the comments. In the nonconvex case, our convergence result is fundamentally based on the Kurdyka–Łojasiewicz (KL) property, which plays a critical role in nonconvex optimization.
>
> It has been shown in [1] that a semialgebraic function has the KL property and that all the objective functions in Examples (2) to (6) and in numerical experiments are semialgebraic functions. Therefore, they satisfy the KL property. We will add a paragraph to clarify this.
>
> ### 2.  For  "While it is clearly stated how they differ from the existing literature, they refer to the range assumption used in existing works, and explicitly stating it would have helped understanding their contribution better."
>
> Thank you for the comments to improve the readability. The existing range assumption has been stated in the paragraph before Section 1.1. In the revision, we will make the existing range assumption more explicit. For more discussions about Assumption A(2), we refer to the rebuttal to Reviewer LYmF.
>
> ###  3. For "Although the use of the ADMM also in the field of Machine Learning is highlighted and it is shown that existing methods lack convergence properties, I am not sure of the general significance for Machine Learning."
>
> Our goal is to advance the theoretical foundation of proximal ADMM for an important structured class of problems. The class of problems studied in this paper is practically meaningful.   These problems are encountered in various machine learning applications, such as the consensus problem (example 1), image processing (example 3), and robust regression (example 4).
> Our contribution is the first to provide a theoretical analysis for this structured class of problems under the weaker Assumption A(2), thereby extending the scope of proximal ADMM beyond the existing range-assumption-based theory.
>
> [1] Attouch H, Bolte J, Redont P, et al. Proximal alternating minimization and projection methods for nonconvex problems: An approach based on the Kurdyka-Łojasiewicz inequality[J]. Mathematics of Operations Research, 2010, 35(2): 438-457.

---

> > ### Author Rebuttal · Reviewer_PLkU · 2026-04-04
> >
> > My questions were all answered by the authors.

---

### Official Review · Reviewer_LYmF · 2026-03-13

**Soundness:** 3
**Presentation:** 2
**Significance:** 3
**Originality:** 2
**Overall Recommendation:** 4
**Confidence:** 3

**Summary:**

This paper proposed a novel assumption for proximal ADMM for multiblock problems with the block anti-upper triangular constraints.This novel assumption proposed extends the theoretical foundation of algorithms for problems with block anti-upper triangular constraints.For nonconvex  problems,authors give the global convergence analysis under the proposed assumption without the classic “range assumption”.Moreover,in the strongly convex setting,this paper also prove the global convergence of the proximal ADMM and establish the linear convergence under four different scenarios.

**Compliance With Llm Reviewing Policy:**

Affirmed.

**Key Questions For Authors:**

Although this paper provided a rigorous theoretical explanation,I still have doubts about the following aspects:
1. Authors mentioned multiple times in the article that detailed theoretical analysis can be found in the appendix (such as line 61 mentioning Table 3,line 181 mentioning that all theorems and lemma proofs can be found in the appendix, etc.). In fact, I did not see the appendix.
2. This paper claims A(2) is weaker than the classical range assumption,Could the authors provide a simple example where A(2) holds but the classical range assumption fails?
3. The experiments only compare with classical ADMM,designing extensive experiments including ADMM variants would strengthen the empirical evaluation.

**Strengths And Weaknesses:**

Strengths:
1.This novel assumption proposed extends the theoretical foundation of algorithms for problems with block anti-upper triangular constraints.

2. For nonconvex  problems,authors give the global convergence analysis under the proposed assumption without the classic “range assumption”.Moreover,in the strongly convex setting,this paper also prove the global convergence of the proximal ADMM and establish the linear convergence under four different scenarios.

Weaknesses:
1. Authors mentioned multiple times in the article that detailed theoretical analysis can be found in the appendix (such as line 61 mentioning Table 3,line 181 mentioning that all theorems and lemma proofs can be found in the appendix, etc.). In fact, I did not see the appendix.
2. This paper claims A(2) is weaker than the classical range assumption,Could the authors provide a simple example where A(2) holds but the classical range assumption fails?
3. The experiments only compare with classical ADMM,designing extensive experiments including ADMM variants would strengthen the empirical evaluation.

---

> ### Author Rebuttal · Authors · 2026-03-30
>
> We thank the reviewer for the careful reading and positive assessment of our work. Below we respond to the main points one by one.
>
> ---
>
> ## 1. About the appendix and supplementary materials
>
> The appendix and the code files are included in the **supplementary materials**. In particular, it contains:
>
> - The comparison with related work for the **nonconvex** setting in **Table 3**,
> - The comparison with related work for the **strongly convex** setting in **Table 4**,
> - All detailed proofs of the theorems and lemmas.
>
> ---
>
> ## 2. About whether Assumption A(2) is weaker than the classical range assumption
>
> Thank you for this question. Consider the robust regression problem:
>
> $$
> \min_{\mathbf{X}_1, \mathbf{X}_2, \mathbf{X}_3} \ g(\mathbf{X}_1) + \alpha_1 \Big( \alpha_2 P(\mathbf{X}_2) + \frac{1}{2} (1-\alpha_2) ||\mathbf{X}_3||_F^2 \Big)
> $$
>
> $$
> \text{s.t. } \mathcal{H}(\mathbf{X}_1) = \mathbf{X}_2, \quad \mathcal{H}(\mathbf{X}_1) = \mathbf{X}_3.
> $$
> When we combine these constraint matrices into one overall matrix, the constraint of the above problem is formulated as:
>
>
> $$
> [-\mathcal{I} \quad 0 \quad \mathcal{H} \\ ; \quad 0 \quad -\mathcal{I} \quad \mathcal{H}]
> [\mathbf{X}_3 \quad \mathbf{X}_2 \quad \mathbf{X}_1]^T
> = [0 \quad 0]^T
> $$
>
>
> This clearly shows that:
>
> - For $ \mathbf{X}_ 1 $, both parts of the constraint are $ \mathcal{H} $, i.e.,  $ \mathcal{A}_ {1, 3} = \mathcal{A}_ {2, 3} = \mathcal{H} $
>
> - For $ \mathbf{X}_ 2 $, the constraint consists of $ \mathcal{I} $ and 0, i.e.,  $ \mathcal{A}_ {1, 2} = 0 $ and $ \mathcal{A}_ {2, 2} = -\mathcal{I} $
>
> - For $ \mathbf{X}_ 3 $, the constraint consists of 0 and $ \mathcal{I} $, i.e.,  $ \mathcal{A}_ {1, 1} = -\mathcal{I} $ and $ \mathcal{A}_ {2, 3} = 0 $
>
>
> As you can see, when $\mathcal{H}$ is invertible, **Assumption A(2)** holds because it satisfies
>
>
>
> $$
> \text{range}(\mathcal {A}_ {1,1}) \subseteq \text{range}(\mathcal{A}_ {1,3}) \quad \text{and} \quad
> \text{range}(\mathcal{A}_ {2,2}) \subseteq \text{range}(\mathcal{A}_ {2,3}),
> $$
>
> but the **range assumption** does not hold.
>
> In fact, the examples 1 to 4 presented in our paper already illustrate this point. They all involve linear constraints with block anti-upper triangular structure under which the **classical range assumption fails**, while **Assumption A(2) still holds**. Hence, these examples demonstrate that Assumption A(2) is not only weaker than the classical range assumption, but also practically meaningful and applicable to important problem classes.
>
> ---
>
> ## 3. About the empirical comparisons with ADMM variants
>
> We compared the proposed ADMM method with the semiproximal ADMM [1] and the linearized ADMM method [2] using the nonconvex robust regression problem and the block angular problem discussed in Section 3.1. Since the classic range assumption does not hold for this problem, the semiproximal ADMM and the linearized ADMM method are not theoretically guaranteed to converge globally.
>
> The codes can also be found at [https://anonymous.4open.science/r/pADMM-7987](https://anonymous.4open.science/r/pADMM-7987). The results are reported in Figures 1-4 at the same link. Specifically, Figures 1 and 2 are for nonconvex robust regression, while Figures 3 and 4 are for the block angular problem in the strongly convex setting.
> It is shown in the figures that pADMM outperforms the other comparative algorithms, demonstrating its superior convergence performance and robustness. We will add these comparisons to the revised version of this paper.
>
> [1] Yashtini M. Convergence and rate analysis of a proximal linearized ADMM for nonconvex nonsmooth optimization[J]. Journal of Global Optimization, 2022, 84(4): 913-939.
>
> [2] Lam X Y, Sun D, Toh K C. Semi-proximal augmented Lagrangian-based decomposition methods for primal block-angular convex composite quadratic conic programming problems[J]. INFORMS Journal on Optimization, 2021, 3(3): 254-277.

---

> > ### Author Rebuttal · Reviewer_LYmF · 2026-04-04
> >
> > The authors have already solved my problems.

---

### Decision · Program_Chairs · 2026-04-30

**Decision:**

Accept (regular)

**Comment:**

This paper establishes the global convergence and rate analysis of the proximal Alternating Direction Method of Multipliers (ADMM) for multiblock problems featuring block anti-upper triangular constraints. A key contribution is the introduction of a novel assumption, $A(2)$, which relaxes the classical range assumption that typically fails for these specific problem structures. Under this new assumption, the authors successfully prove global convergence in non-convex settings and establish linear convergence in strongly convex settings.

During the initial review phase, all three reviewers (LYmF, PLKU, ZSZP) recognized the technical soundness of the paper and assigned a score of "Weak Accept". They commended the paper for extending the theoretical foundation of multi-block ADMM algorithms.

However, the reviewers raised several valid points for clarification:

Reviewer LYmF requested a concrete example demonstrating where Assumption $A(2)$ holds but the classical range assumption fails, pointed out the need for empirical comparisons with other ADMM variants, and noted that the appendix appeared to be missing.

Reviewer PLKU requested more explicit statements regarding the existing range assumption and asked for clarification on the paper's general significance for Machine Learning.

Reviewer ZSZP raised technical questions regarding hyperparameter impacts, alternative stopping criteria, and potential counterexamples, alongside a procedural note that the single-column formatting of algorithms potentially violated conference guidelines.

Justification for Acceptance:
The paper makes a solid and rigorously proven theoretical contribution to optimization methods. It specifically addresses a practical gap in solving problems with block anti-upper triangular constraints where standard assumptions are insufficient. The initial scores were uniformly positive, and the authors engaged in an exemplary rebuttal process that successfully cleared all residual doubts. Notably, following the authors' responses, all three reviewers explicitly marked their concerns as "Fully resolved". Given the technical soundness of the work , the value of the theoretical extension , and the unanimous consensus among reviewers post-rebuttal, I confidently recommend this paper for acceptance.